# Analysis of extracellular mRNA in human urine reveals splice variant biomarkers of muscular dystrophies

Layal Antoury[1,2], Ningyan Hu[1,2], Leonora Balaj[1,2], Sudeshna Das [1,2], Sofia Georghiou[2,3], Basil Darras[2,3], Tim Clark [1,2], Xandra O. Breakefield[1,2,4] & Thurman M. Wheeler[1,2]

Urine contains extracellular RNA (exRNA) markers of urogenital cancers. However, the capacity of genetic material in urine to identify systemic diseases is unknown. Here we describe exRNA splice products in human urine as a source of biomarkers for the two most common forms of muscular dystrophies, myotonic dystrophy (DM) and Duchenne muscular dystrophy (DMD). Using a training set, RT-PCR, droplet digital PCR, and principal component regression, we identify ten transcripts that are spliced differently in urine exRNA from patients with DM type 1 (DM1) as compared to unaffected or disease controls, form a composite biomarker, and develop a predictive model that is 100% accurate in our independent validation set. Urine also contains mutation-specific *DMD* mRNAs that confirm exon-skipping activity of the antisense oligonucleotide drug eteplirsen. Our results establish that urine mRNA splice variants can be used to monitor systemic diseases with minimal or no clinical effect on the urinary tract.

[1] Department of Neurology, Massachusetts General Hospital, Boston, MA, USA. [2] Harvard Medical School, Boston, MA, USA. [3] Department of Neurology, Boston Children's Hospital, Boston, MA, USA. [4] Department of Radiology, Massachusetts General Hospital, Boston, MA, USA. Correspondence and requests for materials should be addressed to T.M.W. (email: twheeler1@mgh.harvard.edu)

Pre-mRNA splicing occurs when introns are removed to generate a protein-coding message, while alternative splicing involves inclusion or exclusion of certain exons to code for different protein isoforms from the same gene. These splice variants are a fundamental process of nature that increase biodiversity, mainly in eukaryotes. Mis-regulation of pre-mRNA alternative splicing is found in a number of neurologic and neuromuscular diseases[1]. For example, in myotonic dystrophy type 1 (DM1), an expanded CUG repeat (CUG$^{exp}$) in the 3′ UTR of the *DM Protein Kinase* (*DMPK*) transcript disrupts splicing regulator proteins in the muscleblind-like (MBNL) family, causing abnormal splicing of a number of pre-mRNAs[2,3]. In DM1 patients, pre-mRNA splicing outcomes in muscle biopsies are used as biomarkers of disease severity[4], while in DM1 mice they also have served as sensitive indicators of therapeutic drug activity[5,6]. Less invasive biomarkers to assess disease state and response to therapy in DM1 are currently unavailable, and optimal outcome measures of therapeutic success remain undefined. As a result, a recent clinical trial of an antisense oligonucleotide (ASO) drug for DM1 required participants' consent to multiple muscle biopsies to monitor splicing outcomes in response to therapy and was restricted to adult patients[7]. This experimental drug for DM1 is designed to induce knockdown of *DMPK*-CUG$^{exp}$ transcripts through the RNase H pathway, thereby rescuing muscle cells from the pathogenic effects of splicing mis-regulation[6,8].

Extracellular vesicles (EVs) include exosomes, microparticles/microvesicles (MVs), and other membrane-encased particles released and taken up by cells as a form of extracellular communication[9]. EVs in serum and urine contain mRNA and non-coding RNAs, including microRNA (miRNA), collectively termed extracellular RNAs (exRNAs), which are released from different tissues and can serve as genetic biomarkers of cancers and other disease states[10–12]. Mutations, deletions, translocations, and transcriptome variations also have been shown extensively in EVs, especially for cancers[13–15]. Differentiated skeletal muscle cells in culture release EVs[16,17] and a handful of miRNA biomarkers and several protein signatures have been identified in serum of muscular dystrophy (MD) patients[18]. However, the capacity of muscle-derived exRNA in urine to serve as biomarkers for MDs seems unlikely given that they would be released into the blood circulation and would be unable to pass through the glomerular filtration system of the kidney[19]. Here we examine whether RNA splice products in human blood or urine could meet sensitivity and specificity criteria as robust biomarkers of disease activity and/or therapeutic target engagement for MDs.

## Results

### Characterization of exRNA in biofluids from DM1 and unaffected (UA) control subjects.
To examine the possibility of detecting biomarkers of MDs in human biofluids, we analyzed exRNA microarray and raw sequencing data from two previous studies and found that more than 30 transcripts previously reported as "splicing biomarkers" in DM1 muscle biopsy tissue could be detected in control human serum and urine (Supplementary Tables 1, 2)[20,21]. To determine whether splice variants of these transcripts also are present in human biofluids, we collected urine, with or without blood, samples from 40 subjects with DM1 and 29 unaffected (UA) controls (Supplementary Tables 3, 4). First, we removed cells from biofluid samples by low-speed centrifugation followed by passage of the supernatant through a 0.8 μm filter. Using particle-tracking analysis to characterize extracellular vesicles in cell-free biofluids, we found that particle content was greater in serum than urine, and showed no difference in number or size between DM1 and

controls (Supplementary Fig. 1). Conversely, mean particle size was greater, and spanned a larger range, in urine than serum in both DM1 and UA controls (Supplementary Fig. 1). The mean particle size of ~75 nm that we observed in serum is in the range of exosomes (30–100 nm), while the mean urine particle size of 130–140 nm is in the range of microparticles (100–1000 nm)[22]. Optical density curves of exRNA isolated from particles, and the 260/280 nm absorbance ratios, appeared similar in DM1 and UA controls in both urine and serum (Supplementary Fig. 1). exRNA quality and size distribution also appeared similar in DM1 and UA controls in both urine and serum by capillary gel electrophoresis (Supplementary Fig. 1).

To estimate exRNA concentrations obtained from urine and serum samples, we used two separate methods. Measurements by microvolume spectrophotometry appeared similar in DM1 and UA urine and serum samples (Supplementary Fig. 2). By contrast, using capillary gel electrophoresis, mean exRNA concentrations in DM1 urine samples appeared significantly higher than in UA urine samples, and also significantly higher in DM1 urine than in DM1 or UA serum samples (Supplementary Fig. 2). Capillary gel electrophoresis measurements of RNA concentration were lower in urine and serum of both groups than measurements of identical samples using microvolume spectrophotometry, being about 1/3 lower in DM1 urine samples, and an order of magnitude lower in UA urine, and in DM1 and UA serum samples (Supplementary Fig. 2). Similarly, microvolume spectrophotometry estimations of the exRNA mass recovered per milliliter of biofluid (exRNA yield) were significantly higher in serum than in urine for both DM1 and UA groups, and showed no difference between DM1 and UA groups in either urine or serum samples, while capillary gel electrophoresis measurements indicated that exRNA recovery tended to be higher in DM1 urine samples than in UA (Supplementary Fig. 2).

Due to the disagreement between measurements using these two methods, we correlated exRNA concentrations determined by microvolume spectrophotometry or capillary gel electrophoresis with droplet digital PCR (ddPCR) measurements of a reference gene *General Transcription Factor 2B* (*GTF2B*)[6], measured by copies of *GTF2B* transcripts per microliter of cDNA that was made using the same exRNA sample measured by each method. Microvolume spectrophotometry estimations of exRNA concentration showed no correlation with *GTF2B* values, while capillary gel electrophoresis estimations showed good correlation (Supplementary Fig. 2).

### Quantitative gene expression in biofluids from DM1 and UA controls.
Urine contains RNA from two broad categories: (1) exRNA released by cells lining the urinary tract, including the kidney, bladder, and urothelial (transitional epithelial) cells that line the inside of the ureters and proximal urethra[22,23], and (2) RNA contained inside of cells normally present in the urine, including small numbers of erythrocytes (0–2/high power field [HPF]) and leukocytes (0–2/HPF), exfoliated renal tubular cells, urothelial cells, and squamous epithelial cells, and recently described "urine stem cells"[24–27]. Using droplet digital PCR (ddPCR), we analyzed *DMPK* gene expression in exRNA from urine and serum, and in total RNA from urine cells. *DMPK* content, as measured by copies per microliter of input cDNA, appeared similar between DM1 and UA subjects in both sources of urine RNA (Fig. 1a). However, in urine exRNA, expression of the reference gene *GTF2B*[6] tended to be higher in DM1 as compared to UA subjects, and normalization of *DMPK* expression to *GTF2B* in urine exRNA revealed *DMPK* levels in DM1 patients about half that in the UA group

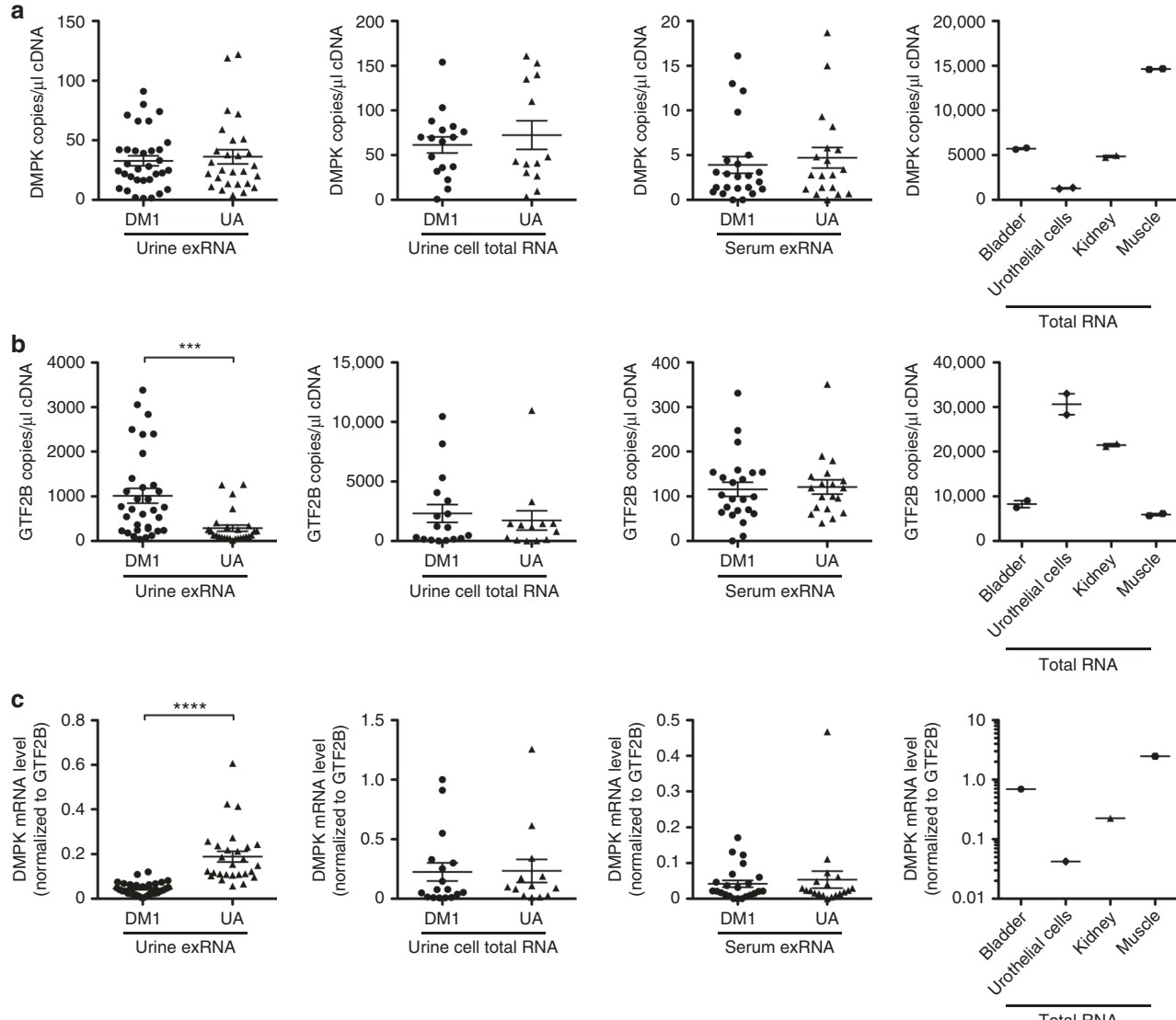

Fig. 1 *DMPK* gene expression in human urine and serum. We used droplet digital PCR (ddPCR) to examine gene expression in extracellular RNA (exRNA) from urine (N = 33 DM1 patients and 27 UA control subjects without muscular dystrophy), total RNA from urine cells (N = 17 DM1 and 13 UA controls), serum exRNA (N = 23 DM1 and 19 UA controls), and commercially available total RNA from human bladder tissue, urothelial cells, kidney tissue, and skeletal muscle tissue. **a** Expression of *DMPK* mRNA, and **b** reference gene *GTF2B* mRNA, as measured by RNA copy number per microliter of input cDNA. **c** *DMPK* expression normalized to *GTF2B*. Individual data points represent the mean of duplicate assays for each sample. Error bars indicate mean ± s.e.m. ****$P < 0.0001$; ***$P = 0.0002$ ($t$ test)

(Fig. 1b, c). By contrast, *GTF2B* levels in urine cell total RNA showed no difference between DM1 and UA groups, and normalized DMPK expression was similar in these groups (Fig. 1b, c). In serum, *DMPK* and *GTF2B* were expressed at similar levels in DM1 and UA subjects, and in both groups were present at levels nearly an order of magnitude lower than in urine (Fig. 1a–c).

Normalized *DMPK* expression in urine exRNA of UA subjects appeared most similar to that in total RNA from normal human kidney tissue and urothelial cells, and approximately threefold lower than in bladder, and an order of magnitude lower than in skeletal muscle (Fig. 1c). Using a second quantitative PCR method, Taqman qPCR, *DMPK* mRNA expression normalized to *GTF2B*, or to a second reference gene, *Glyceraldehyde 3-phosphate dehydrogenase* (*GAPDH*), also appeared significantly lower in urine exRNA of DM1 patients than UA subjects, and appeared similar in serum exRNA from both groups (Supplementary Fig. 3).

**Alternative mRNA splice variants in biofluid exRNA and total RNA from urine cells**. Next we analyzed alternative splicing patterns in urine exRNA of DM1, UA controls, and 15 individuals with mutations in the dystrophin (*DMD*, in italics) gene as muscular dystrophy controls (N = 12 Duchenne muscular dystrophy [DMD, no italics], 3 non-ambulatory Becker muscular dystrophy [BMD]; Supplementary Tables 3, 4), focusing on transcripts previously reported as biomarkers of DM1 disease severity in muscle biopsies (Supplementary Table 5)[4]. We examined 33 alternative splice events and found 10 that are different in urine exRNA of DM1 patients as compared to subjects with *DMD* gene mutations or UA individuals (Fig. 2), while the remaining 23 splice events appeared similar in DM1 and UA groups (Supplementary Fig. 4). Longitudinal analysis found that splicing patterns in urine exRNA appeared similar in two or more consecutive samples collected from the same individual over a 6–26-month period (Fig. 3).

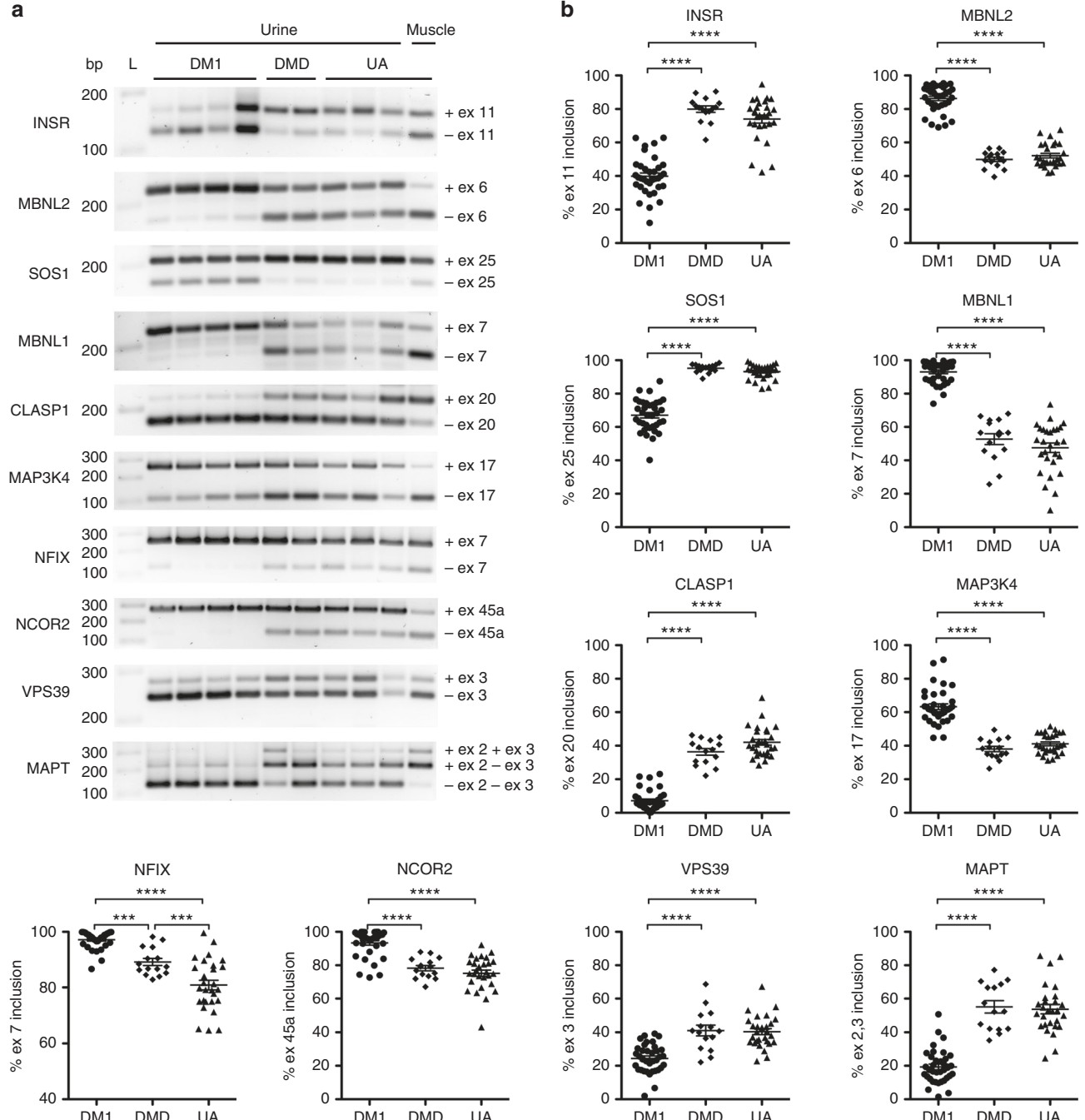

**Fig. 2** Alternative splicing of extracellular mRNA from human urine. We isolated urine exRNA from 36 DM1, 15 DMD/BMD controls (DMD), and 28 unaffected (UA) subjects, and examined alternative splicing by RT-PCR and gel electrophoresis[4]. Commercially available total RNA from skeletal muscle tissue served as a control. **a** Representative gel images showing alterative splicing of *INSR* exon 11, *MBNL2* exon 6, *SOS1* exon 25, *MBNL1* exon 7, *CLASP1* exon 20, *MAP3K4* exon 17, *NFIX* exon 7, *NCOR2* exon 45a, *VPS39* exon 3, and *MAPT* exons 2 and 3. PCR cycle number was 36 (*INSR*, *MBNL2*, *SOS1*, *CLASP1*, *MAP3K4*, *NFIX*, *NCOR2*, and *VPS39*) or 37 (*MBNL1* and *MAPT*). Control muscle cDNA was diluted 1:100 and amplified in the same PCR reaction as urine samples. "L" = DNA ladder. "bp" = base pairs. **b** Individual data points represent quantification of splicing in urine exRNA samples of all individuals examined. Error bars indicate mean ± s.e.m. ****P < 0.0001; *** = mean difference 8.0, 95% CI of difference 3.2–12.7 (*NFIX*, DM1 vs. DMD/BMD) and mean difference 8.3, 95% CI of difference 3.5–13.2 (*NFIX*, DMD/BMD vs. UA); one-way ANOVA

In contrast to urine exRNA, splicing in total RNA from urine cells was different between DM1 and the UA group for only two transcripts, *MBNL2* and *MAPT*, and the difference between means of this splice event was smaller than in exRNA (Supplementary Fig. 5). A third transcript, *MAP3K4*, showed a trend for a difference between the DM1 and UA groups in urine cells, but failed to reach statistical significance (Supplementary Fig. 5). Examination of these splice products in serum exRNA failed to show a difference between the DM1 and UA control groups for any of the transcripts examined (Supplementary Fig. 6).

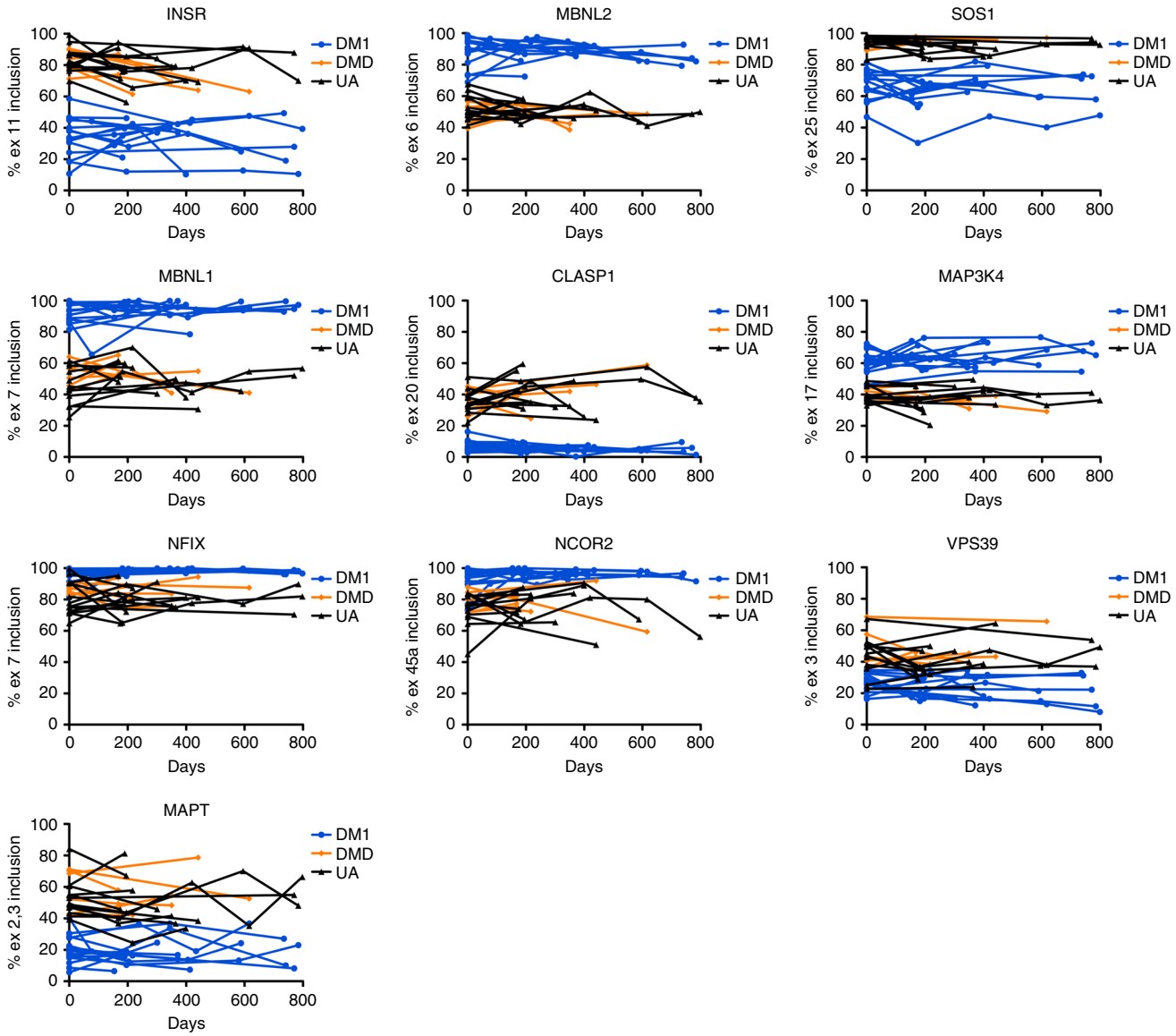

**Fig. 3** Longitudinal analysis of exRNA alternative splicing in human urine. We collected two or more urine specimens from 15 DM1, 5 DMD/BMD, and 13 UA subjects over the course of several months and examined exRNA splicing outcomes by RT-PCR. Note that due to low collection volume, low RNA recovery, and/or exhaustion of cDNA samples from testing 33 overall splice events and 7 transcripts by ddPCR, longitudinal data for *NCOR2* are available from only 14 DM1 subjects, *CLASP1* and *NCOR2* from only 12 UA subjects, and several transcripts are missing 2-year data points for one or two DM1 and one or two UA subjects

**Composite splicing biomarker and predictive modeling for DM1**. To increase statistical power, several individual biomarkers can be combined to reach a single interpretive readout known as a composite biomarker[28]. Using R statistical software[29], we performed principal component analysis[30,31] of the ten transcripts spliced differently in urine exRNA of DM1 patients to generate a composite splicing biomarker score for each subject, revealing a wide separation between DM1 and DMD/BMD or UA individuals (Fig. 4a). The first principal component, PC1, is the variable that best separates the data points into two groups, DM1 and non-DM. The ten transcripts contribute fairly evenly to the separation along PC1 (Supplementary Table 6).

To generate a predictive model of exon inclusion for DM1, we pooled data sets from the first 45 subjects that were enrolled ($N = 23$ DM1, 22 UA), randomly assigned 75%, irrespective of genotype, to a training set, and the remaining 25% to an independent validation set[32]. Using our training set with principal component regression[33] and a threshold of 0.5, such

that values greater than threshold were DM1 and below threshold UA, the model produced zero false positives and false negatives in a fivefold cross validation test (Fig. 4b). The predictive model was 100% accurate in distinguishing DM1 patients from UA subjects in the validation set ($N = 6$ DM1, 5 UA), and in the next 19 consecutive subjects that were enrolled after its implementation ($N = 13$ DM1, 6 UA), for a total of 30 evaluated using the model (Fig. 4c). The root mean square error of prediction (RMSEP) demonstrates that PC2 through PC10 contribute little to the model, and therefore, only PC1 was used to generate the model (Fig. 4d). The regression coefficients, calculated as a weighted sum, indicate the relative contribution of each individual transcript to the model, with *MBNL2* most responsible for prediction, followed by *CLASP1*, *SOS1*, and *MBNL1* (Fig. 4e).

**Validation of alternative splicing in urine exRNA by ddPCR**. A consideration in the evaluation of a biomarker is the accuracy of

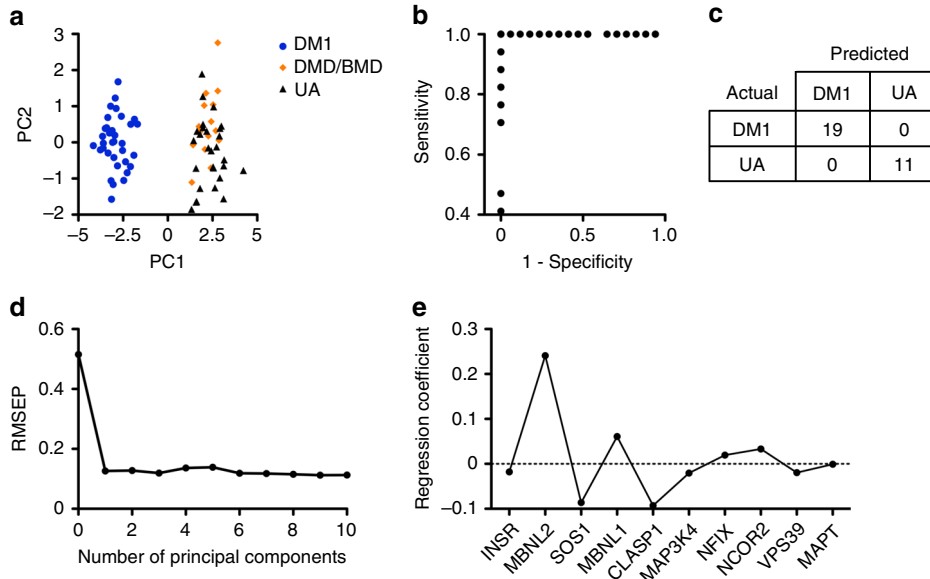

**Fig. 4** Principal component analysis and predictive modeling of urine splicing outcomes. Using principal component regression, a linear combination of ten urine exRNA transcripts that show differential splicing in DM1 subjects (*INSR, MBNL2, SOS1, MBNL1, CLASP1, MAP3K4, NFIX, NCOR2, VPS39*, and *MAPT*) was used to develop a composite biomarker and predictive model of DM1. **a** Principal component (PC) score that served as a composite splicing biomarker for each subject ($N = 36$ DM1, 15 DMD/BMD, 28 UA). The first and second principal components are plotted. **b** We combined PC splicing data from the first 45 consecutive DM1 ($N = 23$) and UA ($N = 22$) subjects enrolled in the study, then randomly assigned 75%, irrespective of genotype, to a training set, and the remaining 25% to an independent validation set[32]. Using the training set and a threshold of 0.5 (see Methods), we developed a predictive model that produced zero false positives and false negatives in a fivefold cross validation test. The receiver operating characteristic (ROC) curve is shown. **c** The predictive model accurately distinguished DM1 from UA in all 11 subjects of the validation set used to test the model ($N = 6$ DM1, 5 UA), and in the next 19 consecutive subjects that were enrolled in the study after model implementation ($N = 13$ DM1, 6 UA), for a total of $N = 30$ that were evaluated using the model. **d** Root mean square error of prediction (RMSEP) as a function of the number of principal components. **e** Regression coefficients, calculated as a weighted sum, which demonstrate the relative contribution to the model of each of the ten transcripts that were used to generate the model

the measuring technique. Analysis of splicing by conventional RT-PCR using the same set of primers to identify the exon inclusion and exon exclusion isoforms may result in preferential amplification of the shorter exon exclusion product, resulting in an overestimation of its levels[34]. ddPCR is an alternative approach that enables absolute quantification of each splice product in terms of copy number per microliter cDNA with high precision, sensitivity, and reproducibility[35]. To examine alternative splicing in urine exRNA, we designed separate primer probe sets for *INSR* exon 11 inclusion and exclusion that have amplicon sizes of 93 and 89 nucleotides, respectively (Fig. 5a). The copy number per microliter of transcripts containing exon 11 appeared similar in DM1 and UA groups, and was lower in both groups than in the DMD/BMD group, while the concentration of the exon 11 exclusion isoform was higher in DM1 than the control groups (Fig. 5b, c). *INSR* exon 11 inclusion was significantly different in the DM1 vs. control groups (Fig. 5d), and showed a strong correlation with RT-PCR measurement ($r = 0.88$, $P < 0.0001$; Fig. 5e). We also evaluated alternative splicing in urine exRNA of four additional transcripts, *MBNL2, MBNL1, CLASP1*, and *MAP3K4*, by ddPCR, and found that inclusion percentages were significantly different in DM1 vs. UA subjects, and values for all four demonstrated a linear relationship to RT-PCR (Fig. 5f, g).

A second form of myotonic dystrophy, DM type 2, (DM2) is caused by a CCTG repeat expansion in the *CNBP* gene[36], and generally features a milder phenotype than DM1. In most populations, DM2 appears to be less common than DM1, although prevalence of DM2 may be underestimated due to the possibility that milder symptoms typical of DM2 lead to under-diagnosis[37]. Muscle biopsies from DM2 patients feature

mis-regulated alternative splicing of many of the same genes as in DM1 muscle biopsies[4]. To determine whether DM2 also affects splicing in urine, we examined alternative splicing by ddPCR in urine exRNA from four individuals with DM2, and compared them to our DM1 and UA groups. Inclusion of *MBNL2* exon 6, *MBNL1* exon 7, and *CLASP1* exon 20 in DM2 urine exRNA samples was significantly different than in the UA and DM1 groups, showing an intermediate change, while the splicing pattern of *INSR* and *MAP3K4* appeared similar in the DM2 and UA groups (Fig. 5d, f).

The sum total of each transcript examined by ddPCR, calculated as the quantity of inclusion plus exclusion splice products, was significantly higher in the DM1 group as compared to the UA group (Fig. 6a–e). *MAP3K4* was the only transcript that was spliced differently in urine exRNA samples based on biologic sex in both the DM1 and UA groups, with inclusion of exon 17 being higher in DM1 males than females, and also higher in UA males than females (Fig. 7a and Supplementary Fig. 7). Inclusion of *INSR* exon 11 was higher in DM1 females than in DM1 males, although showed no difference between males and females in the non-DM group (Supplementary Fig. 7). Exon inclusion showed significant correlation with CTG repeat length in DM1 males for *MAP3K4* ($r = 0.62$, $P = 0.005$), but no correlation in DM1 females (Fig. 7b), while splicing of *INSR, MBNL2, MBNL1*, and *CLASP1* showed weak or no correlation with CTG repeat length in either group (Supplementary Fig. 7).

ddPCR analysis of splicing in urine cells confirmed an increase of *MBNL2* exon 6 inclusion in DM1 vs. UA samples that was evident by RT-PCR, and revealed statistically significant differences in splicing of *INSR, MBNL1, CLASP1*, and *MAP3K4* between the DM1 and UA groups that were non-significant by

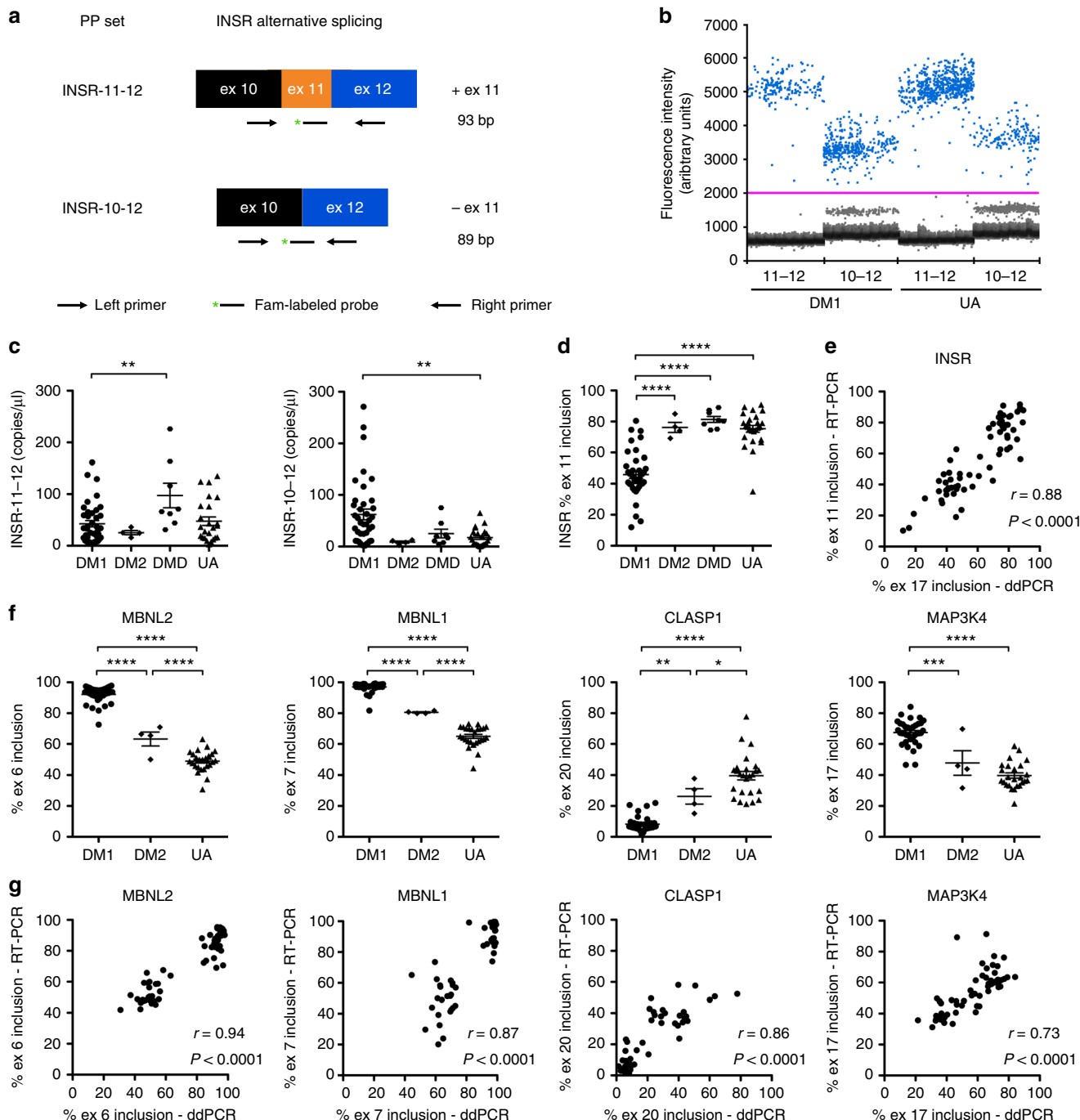

**Fig. 5** Droplet digital PCR (ddPCR) analysis of alternative splicing in human urine exRNA. We used ddPCR to examine splicing of *INSR* exon 11 in urine exRNA samples from DM1 (*N* = 37), DM2 (*N* = 4), DMD/BMD (*N* = 8), and UA controls (*N* = 25). **a** Assay design using separate primer probe (PP) sets to identify exon 11 inclusion (INSR-11-12; amplicon size 93 base pairs, "bp") and exon 11 exclusion (INSR-10-12; amplicon size 89 bp). **b** Representative droplet populations in urine exRNA samples from DM1 and UA subjects (*N* = 4 each) using each assay (INSR-11-12 and INSR-10-12). **c** Quantification of *INSR* splice products that include exon 11 (left) and exclude exon 11 (right) as copies per microliter cDNA. Individual data points represent the mean of duplicate assays for each sample. Error bars indicate mean ± s.e.m. **P = 0.0015 (DM1 vs. UA); *P = 0.01 (DM1 vs. DMD). **d** Calculation of *INSR* exon 11 inclusion using data in **c**. Error bars indicate mean ± s.e.m. ****P < 0.0001 (DM1 vs. DMD/BMD) and (DM1 vs. UA) groups. **e** *INSR* exon 11 inclusion measured by ddPCR (*x*-axis) vs. RT-PCR (*y*-axis) in all subjects. The correlation coefficient *r* and *P* value are shown. **f** Exon inclusion percentages for *MBNL2* exon 6, *MBNL1* exon 7, *CLASP1* exon 20, and *MAP3K4* exon 17. Error bars indicate mean ± s.e.m. ****P < 0.0001 (all four transcripts, DM1 vs. UA), (*MBNL2* and *MBNL1*, DM1 vs. DM2), and, (*MBNL1*, DM2 vs. UA); *** mean difference 14.2, 95% CI of difference 6.2–22.1 (*MBNL2*, DM1 vs. DM2), and mean difference 20.1, 95% CI of difference 9.0–31.2, (*MAP3K4*, DM1 vs. DM2); ** mean difference 17.9, 95% CI of difference 5.5–30.3, (*CLASP1*, DM1 vs. DM2); * mean difference 13.2, 95% CI of difference 0.4–26.4, (*CLASP1*, DM2 vs. UA); one-way ANOVA. **g** Exon inclusion of *MBNL2*, *MBNL1*, *CLASP1*, and *MAP3K4* measured by ddPCR (*x*-axis) vs. RT-PCR (*y*-axis) in all subjects. The correlation coefficient *r* and *P* values are shown

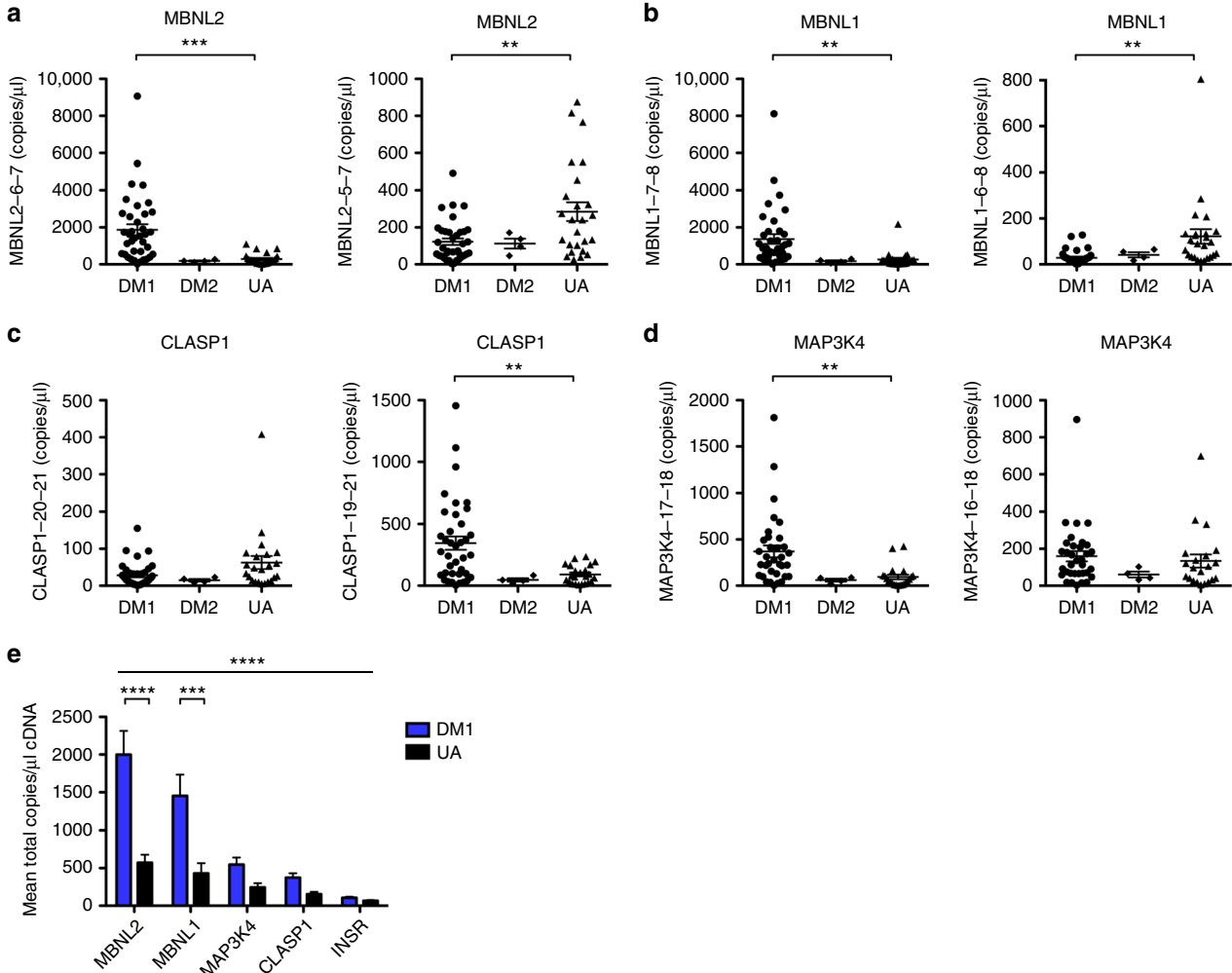

**Fig. 6** Droplet digital PCR (ddPCR) analysis of alternative splicing in human urine exRNA. We used ddPCR to examine alternative splicing of *MBNL2* exon 6, *MBNL1* exon 7, *CLASP1* exon 20, and *MAP3K4* exon 17 in urine exRNA samples from DM1 ($N = 34$–37), DM2 ($N = 4$), and UA controls ($N = 24$ or 25). Individual data points represent the mean of duplicate assays for each sample. **a** Quantification of *MBNL2* splice products that include (MBNL2-6-7; left) and exclude exon 6 (MBNL2-5-7; right) in urine exRNA as transcript copies/μl cDNA. Error bars indicate mean ± s.e.m. ****$P < 0.0001$ (exon 6 inclusion, DM1 vs. UA); **$P = 0.005$ (exon 6 exclusion, DM1 vs. UA); one-way ANOVA. **b** Quantification of *MBNL1* splice products that include (MBNL1-7-8; left) and exclude exon 7 (MBNL1-6-8; right) in urine exRNA as transcript copies/μl cDNA. Error bars indicate mean ± s.e.m. **$P = 0.003$ (exon 7 inclusion, DM1 vs. UA), and $P = 0.005$ (exon 7 exclusion, DM1 vs. UA); one-way ANOVA. **c** Quantification of *CLASP1* splice products that include (CLASP1-20-21; left) and exclude exon 20 (CLASP1-19-21; right) in urine exRNA as transcript copies/μl cDNA. Error bars indicate mean ± s.e.m. ***$P = 0.0009$ (exon 20 exclusion, DM1 vs. UA); one-way ANOVA. **d** Quantification of *MAP3K4* splice products that include (MAP3K4-17-18; left) and exclude exon 17 (MAP3K4-16-18; right) in urine exRNA as transcript copies/μl cDNA. Error bars indicate mean ± s.e.m. **$P = 0.0025$ (exon 17 inclusion, DM1 vs. UA); one-way ANOVA. **e** Mean total copies/μl cDNA (exon inclusion + exclusion values from **a** to **d** and Fig. 5) for each transcript in urine exRNA. Error bars indicate mean ± s.e.m. ****$P < 0.0001$ (all five transcripts combined, DM1 vs. UA) and (*MBNL2*, DM1 vs. UA); ***$P < 0.001$ (*MBNL1*, DM1 vs. UA); two-way ANOVA

RT-PCR (Fig. 8). ddPCR analysis of splicing in urine cells showed good correlation with splicing in urine exRNA for *MBNL2* ($r = 0.66$, $P < 0.0001$), but weak correlation with exRNA for *INSR*, *MBNL1*, *CLASP1*, and *MAP3K4* (Fig. 8). In serum exRNA, ddPCR also confirmed a similar splicing pattern of *MBNL2* exon 6 and *MBNL1* exon 7 in the DM1 and UA groups, and suggested a small increased exon inclusion for both transcripts in DM2 subjects as compared to UA controls, although inclusion percentages for all three groups were substantially lower than in urine exRNA (Supplementary Fig. 8).

**Splicing outcomes related to symptom onset and clinical measures of severity**. DM1 can be classified into groups based on age of symptom onset: classical adult-onset, a less common

childhood or juvenile-onset, and a rare and severe congenital form. Splicing patterns in muscle biopsies of congenital, childhood, or juvenile-onset DM1 patients are poorly characterized, due in large part to the need for general anesthesia to perform biopsies in children, combined with the wide availability of genetic testing in the United States and many other countries, rendering biopsies unnecessary for diagnosis. Of the 39 DM1 urine exRNA samples that we analyzed by ddPCR, $N = 4$ were congenital, $N = 9$ childhood or juvenile-onset, $N = 23$ adult-onset, and $N = 3$ were from asymptomatic individuals that never sought medical care for DM1 until after they were found to have a CTG repeat expansion by genetic testing that was performed as a result of a younger, symptomatic family member having been diagnosed. In urine exRNA, the splicing pattern in samples from asymptomatic individuals was significantly different from both

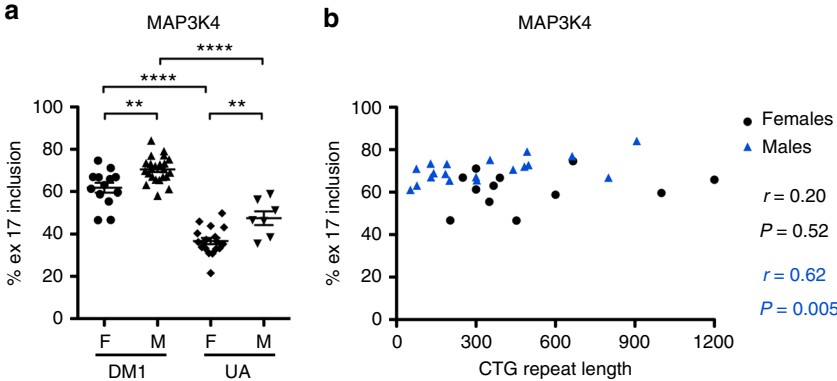

**Fig. 7** Alternative spicing patterns in urine exRNA samples vs. biologic sex and CTG repeat length. **a** MAP3K4 exon 17 inclusion in females vs. males with DM1 and UA groups. Error bars indicate mean ± s.e.m. ****$P < 0.0001$ DM1 females vs. UA females and DM1 males vs. UA males; ** mean difference 8.7, 95% CI of difference 2.0–15.3 (DM1 females vs. DM1 males) and mean difference 10.9, 95% CI of difference 2.3–19.4 (UA females vs. UA males). **b** MAP3K4 exon 17 inclusion vs. DMPK gene CTG repeat length in females (black) and males (blue) with DM1. The correlation coefficient r and P values for each are shown

the symptomatic and UA groups, appearing intermediate in degree (Fig. 9a). Using values from INSR, MBNL2, MBNL1, CLASP1, and MAP3K4 and principal component analysis, we created a ddPCR-based composite splicing biomarker score for each specimen, demonstrating a wide separation between the symptomatic DM1 and UA groups, and an intermediate position of the asymptomatic group (Fig. 9b and Supplementary Table 7).

DM1 typically affects distal muscles, such as those in the hands, forearms, feet, and lower legs, more severely than proximal muscles, such as those in the shoulders or hips. To determine whether urine exRNA splicing patterns could be related to clinical measures of muscle function, we examined ddPCR splicing in DM1 patients who were able to walk at least five steps on their heels while maintaining ankle dorsiflexion, a functional measure of ankle strength ($N = 23$; "+"), and those who were unable to do so ($N = 16$; "−"). Splicing of INSR and MAP3K4 in samples from DM1 individuals who had preserved ankle strength showed intermediate changes between DM1 individuals who had ankle weakness and UA controls, while splicing patterns of MBNL2, MBNL1, and CLASP1 appeared similar in DM1 individuals regardless of ankle strength (Fig. 9c, d).

Next we examined splicing based on the presence or absence of hand grip myotonia, which is delayed muscle relaxation due to repetitive action potentials that results from mis-splicing of CLCN1, the muscle chloride channel[38], often described by patients as difficulty releasing grip from a handshake or a door knob, and is the symptom that most often leads to diagnosis of DM. In DM1 and DM2 individuals without clinical myotonia, splicing of MBNL2, MBNL1, CLASP1, and MAP3K4 showed changes intermediate between those with myotonia and UA subjects (Supplementary Fig. 9).

DM1 patients have increased risk for cardiac arrhythmia and sudden death[39]. Splicing in urine exRNA from DM1 individuals with cardiac arrhythmia, defined here as documented atrial fibrillation/flutter in the medical record, use of a pacemaker, and/or use of an implantable cardioverter-defibrillator, showed no significant difference as compared to samples from DM1 individuals with no known cardiac arrhythmia (Supplementary Fig. 9).

In urine cells, splicing patterns of individual transcripts showed no significant difference based on age of symptom onset, heel walking, myotonia, or cardiac arrhythmia, although ability to heel walk showed a non-significant trend for a difference of INSR,

MBNL1, and MAP3K4 splicing (Supplementary Figs. 9, 10). Similarly, the ddPCR-based composite biomarker scores for urine cells showed overlap between groups in all four categories (Fig. 9b, d and Supplementary Fig. 9).

**Gene expression and alternative splicing patterns in the human urinary tract.** In previous studies, exRNAs in urine have been used as biomarkers of prostate cancer, bladder cancer, and kidney transplant rejection, suggesting that cells lining the urinary tract are the primary contributors to the urine exRNA pool[12,40,41]. To determine whether skeletal muscle tissue may contribute to the exRNA pool, we examined expression of skeletal actin, ACTA1, by ddPCR. In human urine exRNA and urine cell total RNA, ACTA1 levels were low or undetectable in DM1 and UA subjects, and normalized values appeared similar to those in human kidney and bladder tissue, and urothelial cells, but several orders of magnitude lower than in skeletal muscle (Supplementary Fig. 11).

To determine whether the urinary tract may be a primary contributor of the exRNA alternative splice variants in urine, we examined splicing in commercially available total RNA from human kidney and bladder tissues, and urothelial cells. Exon inclusion/exclusion percentages in urine exRNA of UAs appeared more similar to normal kidney and urothelial cells than in normal skeletal muscle or bladder (Supplementary Fig. 12).

**Alternative splicing in the urinary tract of DM1 mouse models.** Nine of the ten transcripts that showed differential alternative splicing in DM1 urine exRNA are regulated by muscleblind-like 1 (MBNL1) protein in muscle[4,32], and loss of MBNL1 protein function in muscle explains >80% of splicing pathology due to CUG$^{exp}$ RNA expression[42]. To determine whether MBNL1 protein also regulates these splice events in the urinary tract, we examined alternative splicing in bladder and kidney tissues of the Mbnl1 knockout mouse model of DM1[2]. Splice patterns in kidney and bladder tissues of Mbnl1 knockout mice appeared significantly different than in wild-type mice, consistent with splicing alterations in muscle of these mice (Supplementary Fig. 13). In contrast, the human skeletal actin-long repeat (HSA$^{LR}$) transgenic mouse model of DM1, which, by design, restricts pathogenic effects of CUG$^{exp}$ RNA to skeletal muscle[43], shows the expected normal splicing pattern in kidney and bladder tissues (Supplementary Fig. 13).

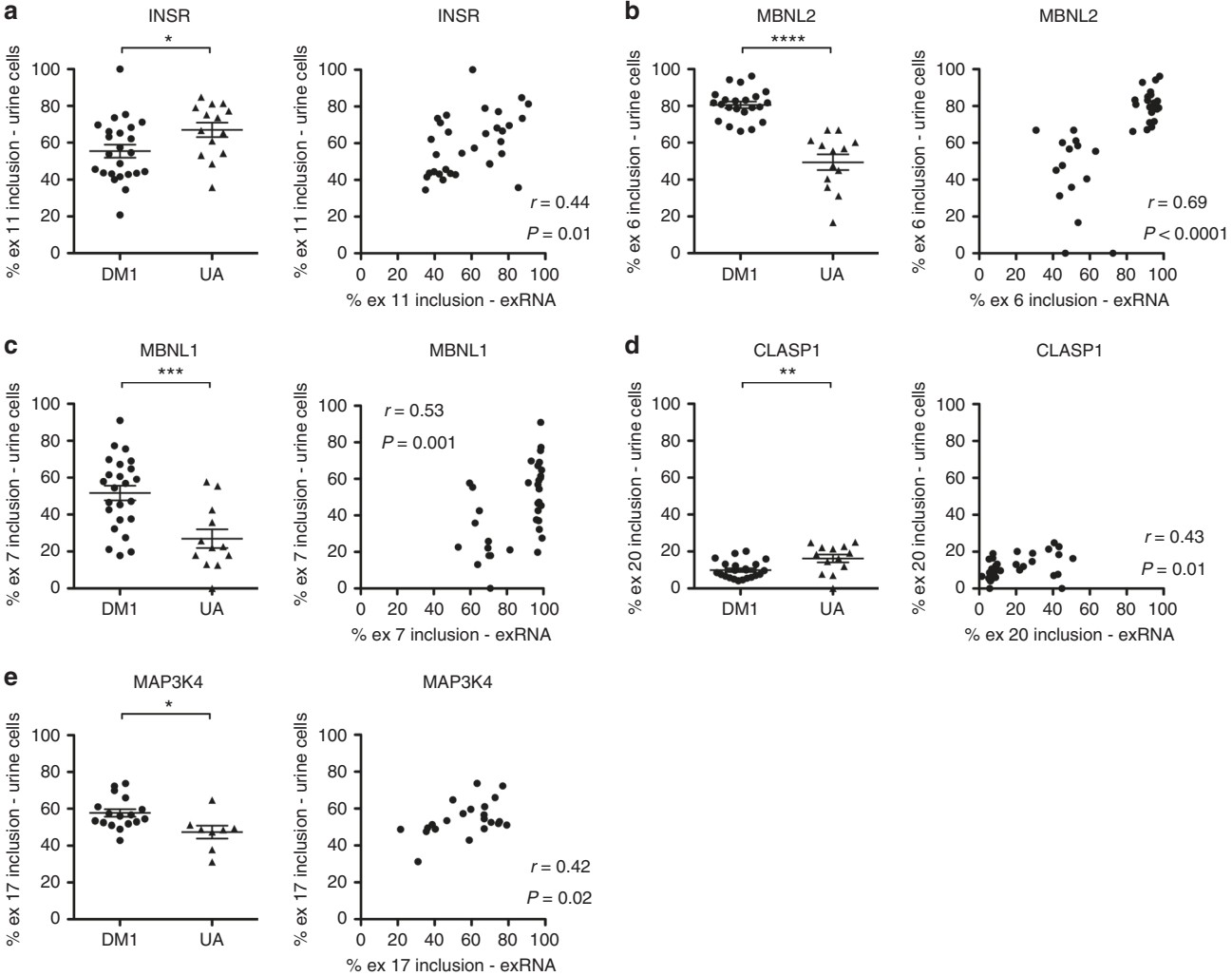

**Fig. 8** ddPCR analysis of alternative splicing in urine cell total RNA and correlation with splicing in urine exRNA. We used ddPCR to determine the alternative splicing pattern in urine cell RNA of DM1 ($N = 22$–24) and UA ($N = 12$–14) subjects and correlated with splicing patterns in exRNA obtained from the same sample. **a** Left, *INSR* exon 11 inclusion in urine cell total RNA. *$P = 0.04$; *t* test. Right, correlation of *INSR* exon 11 inclusion in urine cells with exon 11 inclusion in urine exRNA. The correlation coefficient *r* and *P* values are shown. **b** Left, *MBNL2* exon 6 inclusion in urine cell total RNA. Note that values for two DM1 and one UA specimen were undetectable. ****$P < 0.0001$; *t* test. Right, correlation of *MBNL2* exon 6 inclusion in urine cells with exon 6 inclusion in urine exRNA. The correlation coefficient *r* and *P* values are shown. **c** Left, *MBNL1* exon 7 inclusion in urine cell total RNA. Note that values for one DM1 and two UA specimens were undetectable. ***$P = 0.0008$; *t* test. Right, correlation of *MBNL1* exon 7 inclusion in urine cells with exon 7 inclusion in urine exRNA. The correlation coefficient *r* and *P* values are shown. **d** Left, *CLASP1* exon 20 inclusion in urine cell total RNA. Note that values for two DM1 and one UA specimen were undetectable. **$P = 0.005$; *t* test. Right, correlation of *CLASP1* exon 20 inclusion in urine cells with exon 20 inclusion in urine exRNA. The correlation coefficient *r* and *P* values are shown. **e** Left, *MAP3K4* exon 17 inclusion in urine cell total RNA. Note that, due to exhaustion of supply, only $N = 17$ DM1 and $N = 8$ UA specimens were available for analysis. *$P = 0.01$; *t* test. Right, correlation of *MAP3K4* exon 17 inclusion in urine cells with exon 17 inclusion in urine exRNA. The correlation coefficient *r* and *P* values are shown. All error bars indicate mean ± s.e.m.

**Activity of therapeutic ASOs in mouse muscle and urinary tract tissues**. Of the ten transcripts that we identify here as being spliced differently in urine exRNA of DM1 patients, five of them, *Mbnl1*, *Mbnl2*, *Mapt*, *Nfix*, and *Vps39*, previously were reported as sensitive indicators of therapeutic ASO drug effects in skeletal muscle of HSA[LR] mice[4,6]. Here we show that mis-splicing in muscle of three of the five remaining transcripts, *Map3k4*, *Clasp1*, and *Ncor2*, also is corrected by ASO treatment (Supplementary Fig. 14). In muscle tissue of HSA[LR] and *Mbnl1* knockout mice, splicing of *Sos1* is normal, while splicing of *Insr* in both models changes in the opposite direction of DM1 patients[4].

Next we examined the activity in mouse urinary tract tissues of a therapeutic ASO designed to treat DM1 by inducing RNase H cleavage and subsequent knockdown of *DMPK* transcripts[8]. The *DMPK* target sequence of this ASO is identical in humans,

cynomolgus monkeys, and mice. We treated FVB wild-type mice with either saline or 25 mg/kg twice weekly for 4 weeks (eight total doses), and harvested tissues 1 week after the final dose. Using ddPCR, we found that *Dmpk* levels were reduced by ~40% in bladder, ~50% in kidney, and ~75% in muscle tissues (Supplementary Fig. 15), consistent with the level of knockdown in kidney and muscle of monkeys in the prior study[8].

**Personalized DMD deletion transcripts and confirmation of ASO exon skipping activity in human urine**. ASOs also are being evaluated therapeutically for DMD, typically for individuals with frame-shifting deletions of the *DMD* gene. The treatment strategy involves direct modification of dystrophin pre-mRNA splicing by inducing removal, or skipping, of a target exon to restore the open reading frame and produce an internally

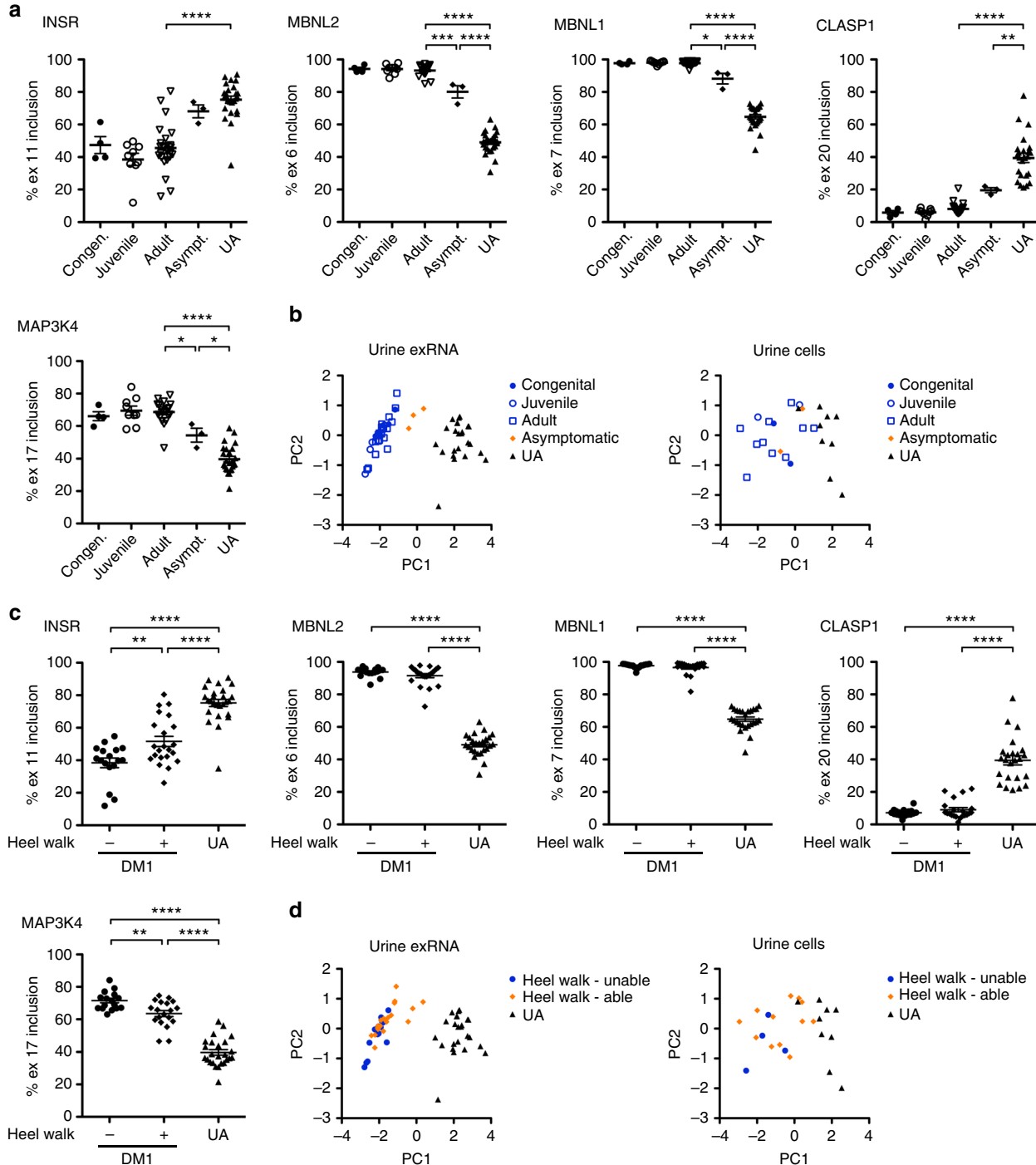

**Fig. 9** Urine exRNA alternative splicing and composite biomarker scores as a function of symptoms. **a** ddPCR quantification of *INSR*, *MBNL2*, *MBNL1*, *CLASP1*, and *MAP3K4* alternative splicing in urine exRNA of congenital (*N* = 4 or 5), juvenile-onset (*N* = 6), adult-onset (*N* = 22 to 24), and asymptomatic adult (*N* = 3) DM1 patients, and in adult UA controls (*N* = 25 or 26). ****$P < 0.0001$ (all five transcripts, adult-onset vs. UA) and (*MBNL2* and *MBNL1*, asymptomatic vs. UA,); *** mean difference 13.1, 95% CI of difference 4.3–22.0 (*MBNL2*, adult-onset vs. asymptomatic); ** mean difference 19.9, 95% CI of difference 3.5–36.3 (*CLASP1*, asymptomatic vs. UA); * mean difference 9.6, 95% CI of difference 1.5–17.6 (*MBNL1*, adult-onset vs. asymptomatic), mean difference 14.2, 95% CI of difference 0.01–28.4 (*MAP3K4*, adult-onset vs. asymptomatic), and mean difference 14.7, 95% CI of difference 0.6–28.7 (*MAP3K4*, asymptomatic vs. UA); one-way ANOVA. **b** Composite biomarker scores generated by principal component analysis of ddPCR splicing measurements in urine exRNA (left) and urine cells (right) displayed as a function of symptom onset. See Supplementary Fig. 9 for splicing quantification of each transcript in urine cells. **c** ddPCR quantification of *INSR*, *MBNL2*, *MBNL1*, *CLASP1*, and *MAP3K4* alternative splicing in urine exRNA of DM1 patients who were able to walk at least five steps on their heels while maintaining ankle dorsiflexion (a functional measure of ankle strength) ("+"; *N* = 19–22), and DM1 patients who were unable ("−"; *N* = 17). **** $P < 0.0001$ (all five transcripts, DM1-Unable vs. UA, and DM1-Able vs. UA); ** mean difference 13.2, 95% CI of difference 3.1–23.2 (*INSR*, DM1-Unable vs. DM1-Able), and mean difference 8.0, 95% CI of difference 1.7–14.4 (*MAP3K4*, DM1-Unable vs. DM1-Able); one-way ANOVA. **d** Composite biomarker scores for urine exRNA (left) and urine cells (right) displayed as a function of the ability to heel walk. All error bars indicate mean ± s.e.m.

truncated, partially functional dystrophin protein[44]. Detection of ASO drug activity in DMD patients involves multiple muscle biopsies to examine whether the target exon has been removed from the mRNA, and quantification of dystrophin protein production by immunofluorescence and western blotting[44]. To determine whether *DMD* deletion transcripts could be identified in biofluid, we examined urine exRNA from six therapy-naive subjects with DMD and found patient-specific deletion transcripts that correspond to the specific gene deletion in all six (Supplementary Fig. 16).

To examine the possibility that biofluid exRNA may be a viable pharmacodynamic biomarker to monitor therapeutic drug target engagement in DMD patients, we examined urine RNA from an individual with an exon 45–50 deletion in the *DMD* gene who is being treated with eteplirsen, the exon-skipping ASO that was granted accelerated approval by the U.S. Food and Drug Administration (FDA) in September 2016[45]. This individual is non-ambulatory and began receiving the ASO in December 2014 as a participant in an open-label safety and tolerability clinical trial of eteplirsen in advanced DMD. Due to accelerated approval of eteplirsen, he has continued receiving weekly infusions of the drug after the clinical trial primary completion date of April 2017. He has never had a muscle biopsy, and pharmacological activity of eteplirsen in this individual had never been established. In lieu of a muscle biopsy, we collected a urine "liquid biopsy" to determine whether splice products in urine mRNA could confirm ASO target engagement. Qualitative end point RT-PCR analysis of urine exRNA and total RNA in urine cells using primers spanning exon 51, the therapeutic target of eteplirsen, demonstrated predominance of a small PCR product (Fig. 10a). Sequencing of this PCR product identified a transition from DMD exon 44 directly to exon 52, confirming absence of exon 51 in both urine exRNA and total RNA from urine cells (Fig. 10b). To quantify the degree of exon 51 inclusion in these samples, we used ddPCR and previously published primer probe sets specific for the skipped (exons 44–52) and unskipped (exons 51–52) products[35], finding that ~90% of *DMD* transcripts in urine exRNA and 98% in urine cells are missing exon 51 (Fig. 10c, d).

Next we examined urine RNA from an individual with DMD due to a deletion of exon 52 in the *DMD* gene, S8, who also is non-ambulatory and had been treated with eteplirsen for more than 3 years at the time of sample collection. RT-PCR using primers spanning the exon 52 deletion identified two bands in urine exRNA and urine cell total RNA that sequencing confirmed were the exon 51 inclusion and exon 51 exclusion (skipped) splice products (Fig. 10e). Using ddPCR and primer probe sets specific for the skipped (exons 50–53) and unskipped (exons 51–53) products[35], exon 51 inclusion was ~26% in both exRNA and urine cells (Fig. 10f–h).

Most of the 79 *DMD* exons, including exon 51, are constitutively spliced[35,46]. To determine whether inclusion of exon 51 could be reduced by a previously unknown alternative splicing event that is unique to urine exRNA and/or urine cells, we used RT-PCR and three separate sets of primers to examine exon 51 splicing in DM1 and UA urine samples, and found no evidence of a splice product that excludes exon 51, while sequencing of PCR products confirmed the presence of exon 51 (Supplementary Figure 16).

## Discussion

Our results demonstrate that mRNA splicing patterns in urine present a rich source of personalized biomarkers with applications to a number of genetic diseases. For DM1, we find ten alternative splice variants in urine exRNA that combine to serve as a robust composite biomarker of DM1 disease activity that is far more powerful than any single splice event. This is significant because mis-regulated alternative splicing outcomes in muscle tissue also serve as sensitive indicators of disease activity in DM1 patients[4] and therapeutic response in DM1 mice[5,6]. Indeed, splice products in muscle biopsies, an invasive procedure, were used in a recent clinical trial as measures of ASO activity in DM1 patients[7]. We propose that our ddPCR-based splicing biomarkers in urine exRNA have the potential to identify drug target engagement in upcoming clinical trials for DM1, initially as a pre-treatment baseline measurement, followed by monitoring during the course of treatment, with the expectation that pharmacological activity should be detected as a shift in the composite splicing pattern.

Because DM1 is primarily a disease of skeletal muscle, heart, and the central nervous system (CNS), and these tissues release EVs, it is counter-intuitive that exRNA reflecting the characteristic mis-regulated splicing events in DM1 would appear in urine rather than in blood, as exRNA has not been shown to pass from the blood through the proximal tubules of the kidney[22]. This suggests that the source of exRNA in these biofluids is likely to be different, and that the primary source in serum is unlikely to be muscle tissue. The fact that the presence of *DMPK* transcripts is an order of magnitude lower in serum than in urine exRNA suggests that the cells contributing to the serum exRNA pool may be primarily those that are unaffected in DM1 patients due to low expression of *DMPK*, thereby explaining the similarity of serum splicing patterns in DM1 and UA subjects. The ~50% lower expression of *DMPK* in urine exRNA of DM1 vs. UA subjects may be due to retention of mutant $CUG^{exp}$ transcripts in the nucleus, thereby preventing their release into the cytoplasm and incorporation into EVs[47].

For biomarker development, an advantage of exRNA over total RNA from urine cells includes greater stability of exRNA due to encasement within membrane-bound vesicles that serve as protection of contents against RNases. In fact, a previous study found that RNA recovery was maintained even after storage of urine for up to 30 days at room temperature[48], suggesting that urine specimens could be analyzed in a central location for exRNA biomarkers to achieve optimal assay reliability. By contrast, due to limited viability of urine cells in voided urine specimens, efficient recovery of RNA from urine cells would require each site or laboratory that collects urine specimens to isolate RNA immediately, introducing variability into the assay. In this study, we also found an important difference of splice patterns in the urine exRNA pool, as compared to those in total RNA from urine cells, indicating that the source of these two RNA populations is distinct, but also suggesting that urine cells have the potential to serve as a second biomarker source complementary to urine exRNA.

For DMD, the urine splice products are more than traditional biomarkers: they also can serve as personalized genetic markers designed specifically for each individual patient and enable the possibility to monitor splice-shifting activity of ASOs. Our finding of ASO-dependent exon-skipping activity in urine exRNA and urine cell total RNA provides the first non-invasive measurement of eteplirsen target engagement, and suggests that our urine biomarkers have the potential to facilitate development of novel ASOs that can outperform eteplirsen or target new exons in DMD. At this point, it is unlikely that an RNA-based assay will eliminate the need for muscle biopsies, as dystrophin protein measurement was used as a surrogate marker of drug effect that led to the accelerated approval of eteplirsen by the U.S. Food and Drug Administration[45]. However, ddPCR-based monitoring of skipped/unskipped *DMD* splice products in urine during the course of treatment may complement splicing analysis of muscle biopsies[35] as newer and better splice-shifting drugs are developed. Long term, it will be important to correlate these early indicators

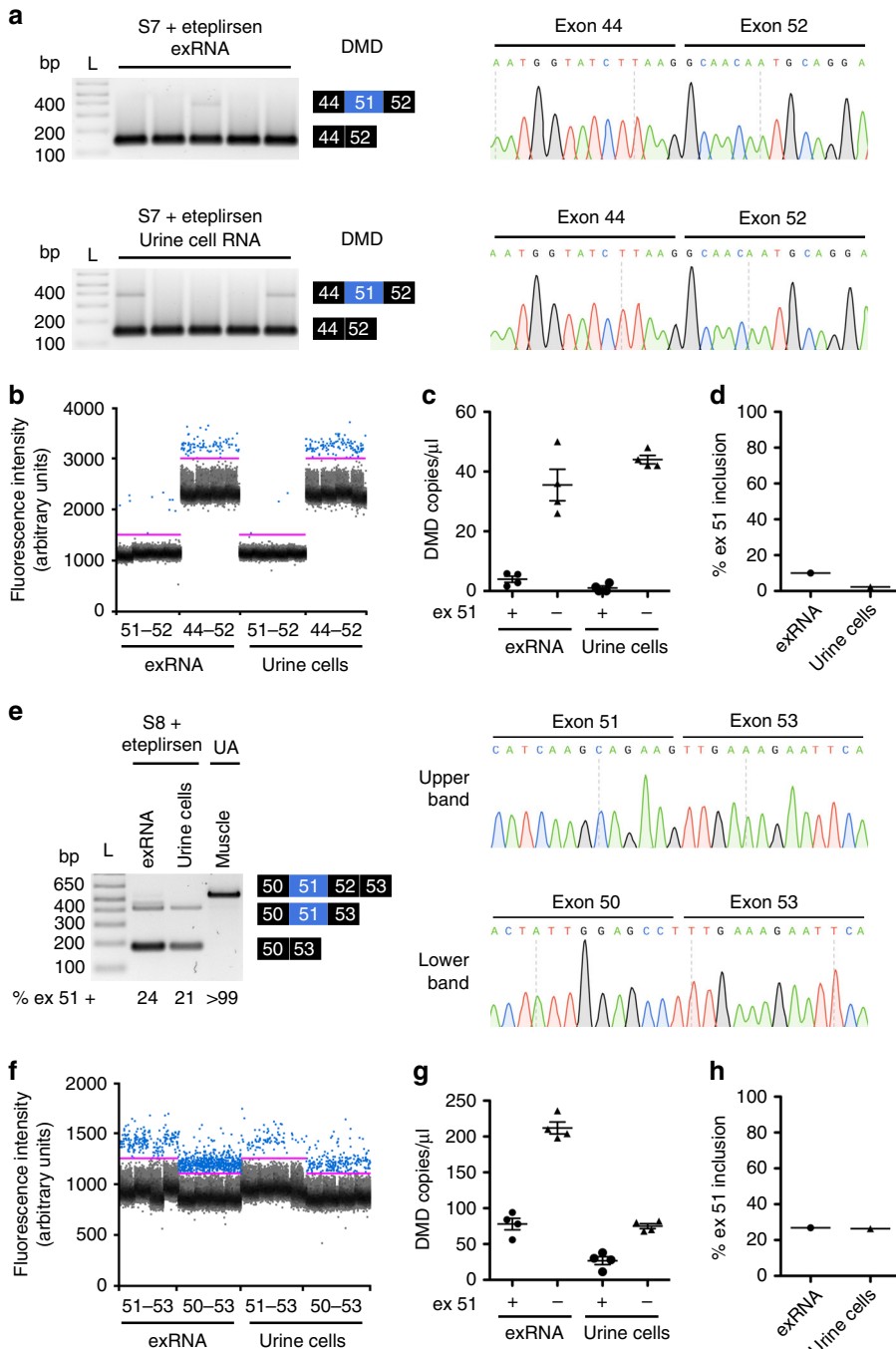

**Fig. 10** Pharmacologic exon skipping activity of eteplirsen in human urine. Using RT-PCR and ddPCR, we examined urine exRNA and urine cell RNA in two non-ambulatory individuals with DMD who have been treated with 30 mg/kg eteplirsen weekly for ~3 years. **a** Left, using RT-PCR and primers targeting exons 44 and 52, two bands are identified in urine exRNA and urine cell total RNA ($N = 5$ replicates each) from a non-ambulatory 17 year-old DMD patient (S7) with an exon 45–50 deletion. Right, sequencing of the lower-band PCR product. **b** ddPCR analysis of urine exRNA and urine cell total RNA ($N = 4$ replicates each) from S7 using separate probe sets that are specific for DMD transcripts with exon 51 included (51–52) or skipped (44–52)[35]. **c** Individual data points indicate quantification of *DMD* splice products in urine exRNA and urine cell total RNA ($N = 4$ replicates each) that have exon 51 included (+) or skipped (−) as copies per microliter of cDNA. **d** Calculation of percent exon 51 inclusion in urine exRNA and urine cells using the data in **c**. **e** Left, using RT-PCR and primers targeting exons 49 and 53, two bands are identified in urine exRNA and urine cell total RNA from a non-ambulatory 19 year-old DMD patient (S8) with an exon 52 deletion. UA muscle tissue RNA served as a control. Right, sequencing of the urine exRNA PCR products. **f** ddPCR analysis of urine exRNA and urine cell total RNA ($N = 4$ replicates each) from S8 using separate probe sets that are specific for DMD 52 deletion transcripts with exon 51 included (51–53) or skipped (50–53)[35]. **g** Individual data points indicate quantification of *DMD* splice products in urine exRNA and urine cell total RNA ($N = 4$ replicates each) that have exon 51 included (+) or skipped (−) as copies per microliter of cDNA. **h** Calculation of percent exon 51 inclusion in urine exRNA and urine cells using the data in (**g**). All error bars indicate mean ± s.e.m.

of drug target engagement with imaging, compositional, and functional outcome measures to determine whether a drug that is working on a molecular level also is having a therapeutic benefit.

## Methods

**Human subjects.** The Partners Health Service/Massachusetts General Hospital (MGH) IRB approved all studies involving human subjects described here. For this study, Boston Children's Hospital (BCH) ceded review to the Partners/MGH IRB. We recruited study participants from the MGH Neuromuscular Diagnostic Center and the Neuromuscular Clinic at BCH. Three groups were studied: individuals with DM1 ($N = 40$), a muscular dystrophy due to a mutation in the DMD gene ($N = 17$ total; 14 with Duchenne phenotype, 3 with Becker phenotype), and with no known muscular dystrophy (unaffected [UA]; $N = 29$) that were either a parent, spouse, or cousin of a study participant with muscular dystrophy. Inclusion criteria for DM1 subjects were age 5 years or older, a diagnosis of DM1 based on genetic testing that identified a *DMPK*-CTG repeat expansion of ≥50, or clinical diagnosis of DM1 and a first-degree relative with DM1 due to a *DMPK*-CTG repeat expansion of ≥50, and ability to provide informed consent or assent for participation. Inclusion criteria for subjects with *DMD* mutations included age 5 years or older, known diagnosis of DMD or BMD, and ability to provide informed consent or assent. Inclusion criteria for unaffected individuals were age 18 years or older, no known history of any muscular dystrophy, and ability to provide informed consent. Prior to participation in the study, we obtained informed consent for blood and/or urine collection from all subjects; due to severe autism, informed consent for one individual with BMD was obtained from his mother/legal guardian, according to IRB protocol. Subject information is shown in Supplementary Table 3.

**Collection and processing of human urine.** Subjects donated urine (approximate range from 14 to 120 milliliters) in a standard specimen container. This was a novel study and we had no reference point for the optimal urine volume that would enable accurate measurement of our target splice products. Therefore, to maximize total exRNA recovery from each specimen, we processed the entire volume that was collected. To help minimize potential confounding factors, samples from DM1, DMD/BMD, and UA subjects were collected in parallel, often being collected, processed, and analyzed simultaneously. In all but one case, UA samples were processed on the same day at the same time with one or more DM1 and/or DMD/BMD sample. Collection and processing of all samples was identical, and the same lab performed RNA isolation, gene expression studies, and splicing analyses for all samples. To remove urine cells, we centrifuged the entire volume at $2450 \times g$ for 10 min at room temperature, passed the supernatant through a 0.8-µm filter into sterile 50 ml tubes, examined specimens using urinalysis strips (Supplementary Table 4), and placed on wet ice within 2 h of collection. We proceeded with exRNA isolation from cell-free specimens either immediately or after storage at 4 °C overnight. To analyze total RNA in urine cells, we added Trizol (Life Technologies) directly to the urine cell pellets after completion of the low-speed centrifugation step, and proceeded according to the manufacturer's recommendations.

**Collection and processing of human serum.** Blood was collected in two standard red top serum separator tubes (Becton Dickinson), incubated at room temperature for 30–45 min, and centrifuged at $2450 \times g$ for 10 min at room temperature. To remove any remaining cells, we passed the serum through a 0.8-µm filter into a sterile 15 ml tube, placed on wet ice within 2 h of collection, and stored at −80 °C. The volume of serum recovered ranged from ~5.5 to 8.5 ml. The blood sample from one individual with DM1 was unusable due to hemolysis and total volume of less than 3 ml.

**Experimental mice.** The MGH IACUC approved all experiments involving mice. HSA^LR transgenic and *Mbnl1* knockout (Mbnl1^ΔE3/ΔE3) models of DM1 (both FVB background) have been described[2,43]. FVB wild-type mice served as controls. ASOs 445236 (targeting *ACTA1*-CUG^exp transcripts in HSA^LR transgenic mice)[6] and 486178 (targeting *Dmpk* transcripts in wild-type mice)[8] were gifts of Dr. Frank Bennett at Ionis Pharmaceuticals (Carlsbad, CA). ASOs were administered by subcutaneous injection at a dose of 25 mg/kg twice weekly for 4 weeks, as previously described[6,8]. To isolate total RNA, we homogenized tissues in Trizol (Invitrogen) and proceeded according to the manufacturer's instructions.

**Characterization of extracellular particles in biofluids.** To determine nanoparticle size and concentration, we used the Nanosight LM10 system and Nanoparticle Tracking Analysis 2.0 analytical software according to the manufacturer's instructions (Malvern). The system uses a laser beam, light microscope, and CCD camera to visualize and video record particles in liquid suspension moving under Brownian motion. For accurate measurements, we diluted serum samples 1:1000 and urine either 1:10 or 1:20 in saline to stay in the target concentration range of $1.0 \times 10^8$–$2.5 \times 10^9$ particles/ml. We recorded 60-s videos and analyzed data in auto mode.

**Isolation of exRNA from biofluids.** We ultracentrifuged cell-free urine and serum samples at $100,000 \times g$ 2 h at 4 °C, removed the supernatant, and extracted RNA from the translucent ribonucleoprotein pellet using 700 or 1000 µl Trizol (Life Technologies) according to the manufacturer's instructions. To enhance RNA pellet visibility, we added 1.4 or 2.0 µl linear acrylamide (Ambion) to each sample and mixed well prior to isopropanol precipitation. Pellets were re-suspended in molecular grade water.

**exRNA analysis.** We measured optical density spectra using a microvolume spectrophotometer (Nanodrop). To measure exRNA size, quality, and total mass of recovered, we used chip-based capillary gel electrophoresis according to the manufacturer's instructions (2100 Bioanalyzer, Agilent Technologies). Using electropherogram traces, a software algorithm (Agilent) automatically determined the RNA integrity number (RIN) based on using a numbering system of 1 (most degraded) to 10 (fully intact)[49].

**RT-PCR analysis of splicing outcomes.** We generated cDNA using Superscript III reverse transcriptase (Life Technologies) and random primers, and performed PCR using Amplitaq Gold DNA polymerase (Life Technologies) and gene-specific primers (Supplementary Tables 8 and 9). We used previously published primer sets for *INSR* and *APT2A1*[3,50] and designed all other primers using Primer3 software[51,52]. Due to the relatively small size of the exRNA species, we targeted the product size for exon exclusion isoforms to be ~100–200 nucleotides whenever possible. Commercially available total RNA from normal human skeletal muscle (Ambion), kidney (Ambion), bladder (Zyagen) tissues, and normal human urothelial cells (ScienCell) served as controls. We separated PCR products using agarose gels, stained with SYBR I green nucleic acid gel stain, and quantitated band intensities using a transilluminator, CCD camera, XcitaBlue^TM conversion screen, and Image Lab image acquisition and analysis software (Bio-Rad). Uncropped gels are shown in Supplementary Figs. 17, 18.

**Gene expression by droplet digital PCR (ddPCR).** We used ddPCR Supermix for probes (Bio-Rad), an automated droplet generator (Bio-Rad QX200), automated droplet reader (Bio-Rad QX200), and PCR cycling conditions according to the manufacturer's instructions as follows: enzyme activation 95 °C for 10 min (1 cycle), denaturation 94 °C for 30 s followed by annealing/extension at 60 °C for 1 min (40 cycles), enzyme deactivation 98 °C for 10 min (1 cycle), and hold at 4 °C. After PCR was complete, the plate was loaded into the droplet reader, processed/analyzed using QuantaSoft software, and total events quantitated using the mean copy number per microliter of duplicate 20 µl assays from individual samples.

To measure gene expression level, we used previously published primer probe sets for human *ACTA1*, human *DMPK*, and mouse *Dmpk*[6], and standard assays for human *GTF2B* or mouse *Gtf2b* (Applied Biosciences, FAM-MGB; assay IDs Hs00976256_m1 and Mm00663250_m1) as normalization controls. The sequence of each PP set that we used to evaluate alternative splicing of *INSR*, *MBNL2*, *MBNL1*, *CLASP1*, and *MAP3K4* is shown in Supplementary Table 10, and the PP sets to identify DMD exons 45–50 and exon 52 deletions with and without exon 51 skipping were published previously[35].

**Sample size.** Splicing patterns in human urine and serum, or even whether specific splice isoforms are present or detectable in these biofluids, were unknown. Therefore, we were unable to choose a sample size ahead of time to ensure adequate power to detect disease-specific differences. Instead, we chose a sample size based on splicing outcomes in muscle biopsies[4] and a goal of enrolling a similar number of DM1 and UA controls. In mice, we chose sample sizes for splicing analysis in muscle based on previously reported differences in muscle tissue of these models[2,3,5,6]. Mice ranged from 2 to 4 months of age and were chosen randomly by genotype, stratified for sex to allow an approximately equal number of females and males, and examined without blinding.

**Statistics.** For two-group and multigroup comparisons, we used unpaired two-tailed t test or multigroup analysis of variance (ANOVA), respectively (Prism software, GraphPad, Inc.). We used the F test to compare variances between two groups (Supplementary Table 11). In groups with statistically significant difference in variance, we used t test with Welch's correction to determine differences between groups. Group data are presented as mean ± s.e.m. To determine associations of splicing with CTG repeat expansion, and ddPCR splicing with RT-PCR splicing, we used Pearson correlation coefficients. A P value <0.05 was considered significant.

**Composite biomarker score.** A principal component analysis[30,31] score for each subject was calculated using R statistical software[33] with a linear combination of the ten RT-PCR splicing outcomes shown in Fig. 2 (*INSR, MBNL2, SOS1, MBNL1, CLASP1, MAP3K4, NFIX, NCOR2, VPS39,* and *MAPT*) to form an RT-PCR-based composite biomarker score (Fig. 4a). We used the same methods to combine the ddPCR splicing outcomes of the five transcripts in urine exRNA shown in Fig. 5 (*INSR, MBNL2, MBNL1, CLASP1,* and *MAP3K4*), and four transcripts in urine cells shown in Fig. 8 (*INSR, MBNL2, MBNL1,* and *CLASP1*) to form ddPCR-based composite biomarker scores (Fig. 9 and Supplementary Fig. 9).

**Predictive model**. We randomly assigned 75% of the first 45 subjects ($N = 23$ DM1, 22 UA) to a training set that consisted of 17 subjects with DM1 and 17 unaffected controls (34/45)[32]. The remaining 25% of these subjects ($N = 6$ DM1, 5 UA) formed an independent validation set. We used principal component regression[33] to develop a predictive model of DM1. Only the first principal component was used for prediction. The pls package in R, which uses a singular value decomposition algorithm to compute the principal components, was used to implement the model[33]. A threshold of 0.5 was chosen for model fitting, such that values above threshold were DM1 and values below threshold UA. A fivefold cross validation test produced zero false positives and false negatives. After development and validation of the model, we then implemented it to examine composite splicing data from each of the next 17 consecutive subjects that were enrolled in the study. Including the 11 subjects in the validation set ($N = 6$ DM1, 5 UA) that were used for model development and the 19 subjects that were tested after model implementation ($N = 13$ DM1, 6 UA), a total of $N = 30$ subjects were evaluated using the model.

## Data availability

All relevant data are available from the authors.

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

## Acknowledgements

We thank the patients and family members who donated blood and urine samples for this study, the MGH DNA core for sequencing support, Dr. M. Swanson (U. Florida) for providing the sequence of *MBNL2* alternative exon 6, and Dr. S. Rutkove (Harvard) for critical review of the manuscript. The Elaine and Richard Slye Fund (TMW), Myotonic Dystrophy Foundation/Wyck Foundation (TMW), Muscular Dystrophy Association (TMW), and NIH NCI P01 CA069246 (X.O.B. and L.B.).

## Author contributions

L.A., S.G., B.D., and T.M.W. obtained informed consent for all subjects. L.A., N.H., S.G., and T.M.W. collected samples. L.A., N.H., and T.M.W. processed samples and performed experiments. L.B., X.O.B., and T.M.W. designed the study. S.D. performed PCA and designed the predictive model. X.O.B. and T.M.W. wrote the paper. All authors analyzed the data, discussed the results, and commented on the manuscript.

## Additional information

**Competing interests:** Massachusetts General Hospital, T.M.W., X.O.B., and L.B. have filed a patent application on the use of urine exRNA to identify markers of muscular dystrophies. The remaining authors declare no competing interests.

