## [Peer Review File · Nature Communications]

Reviewers' comments:

Reviewer #1 (Remarks to the Author):

- In this manuscript the authors show:

o That systemic diseases affecting alternative RNA splicing can be detected in urine exRNA. To my knowledge this novel, but not too surprising, since systemic (genomic) diseases also affect the cells of the urinary tract, even if these cells are not known to display any signs of the disease. Good to have confirmed and published.

o That splice-patterns in exRNA can be used to separate DM1 patients from UAs. This is novel and convincingly shown, including a Principal Component algorithm, which ensures perfect separation of the DM1s and UAs.

o That RNA from urine cells cannot be shown to contain the same information. This is an important claim, and the authors provide a lot of data in support of this, that unfortunately is not completely convincing (quantitative use of end-point PCR).

o I think they show that urine exRNA splice patterns match those found by tissue biopsy. If I understood this correctly this seems very significant and has important clinical applications, but the point is not very prominently positioned in the manuscript and would need the matching tissue data to be included.

o That urine exRNA splice patterns can be used to show the therapeutic effect of eteplirsen in humans. This is hugely important, but the data presented is not convincing.

o A new cryptic splice site for Becker MD. This seems important in its own right and might deserve its own publication in a targeted journal for Molecular Dystrophy.

- I applaud the authors for their use of UAs that are collected at the same clinic as the patients! This is really the best possible way to avoid the sample collection and pre-analytical biases that are commonly plaguing biomarker studies comparing a case and a control group! Samples collected from subjects and UAs in this study have all been treated and collected the same way and I would invite the authors to take credit for this and highlight the importance of it, for the benefit of educating the readers.

- The RNA quality from the nanodrop measurements in Suppl. Fig. 1 seems to be very poor, with OD 260/280 = 1!? I agree with the authors suspicion that the samples seem to be contaminated with residual Phenol/Trizol. In fact, I suspect this spectrum is primarily phenol and less RNA. For good quality RNA in this OD range I would normally suggest using the NanoDrop to estimate RNA concentration rather than the BioAnalyzer as done in S 1d, but this would clearly give a wrong estimate with this contamination. I am not sure about the Bioanalyzers sensitivity to phenol or the consequences for the concentration measurements reported.

- There are several uses of the unit "per μ L" or "per mL", but it rarely says per volume of what! Is it biofluid? RNA eluate? RT reaction? qPCR reaction? This is especially important in the light of the point the authors make of the concentration of RNA in urine being lower in urine than in serum, but the total recovery being higher from urine due to the higher volumes (that BTW are very variable).

- The study seems to be using a very wide range of urine volumes for analysis – why? I understand that results are always normalized, but would it not be better to always use the same volume? What was the rationale for using a certain volume of biofluid? Was it simply all that was available?

- The tissue/cell type expression data in Suppl. Fig. 7b is important to the interpretation of the data in Fig. 1. I wonder if it should be included as Fig. 1d?

- It seems the "normalizer" (GTF2B) is the transcript that has a higher dynamic range than the "biomarker" (DMPK), which makes me wonder what it is "normalizing" to? Here, especially, it would be important to know what the volume of urine is for each sample. I know it doesn't fit with the biological story presented by the authors, but the data as presented could be interpreted as

GTF2B really being the biomarker, and DMPK being the normalizer. Unlike DMPK, the expression of GTF2B is high in Urothelial cells and (it would appear) apparently different in patients with DM1 than in UAs. It would make more intuitive sense if it was the other way around.

- There is an impressive separation of DM1 and UA in PCA (Fig 5) – what is on PC1? It seems PC2 is not needed at all. Can the number of genes be reduced? Is there a simple “sum of splice events” underlying PC1? A linear regression analysis could reveal which gene contributes most to the separation.

- Were there ANY confounding factors that could lead to the impressive separation of DM1 and UA? A discussion of potential confounding factors would be good.

- How are the %-alternative splicing in Fig. 2-4? By scanning the gel and using end-point RT-PCR for 36 cycles as quantitative read-out!? Expecting this to be the case, I am skeptical of the data in figures 2-3-4, although the correlation to ddPCR of INSR in Figure 6g is very convincing (for this one gene). If the authors have any additional data to suggest that this is representative of all the other genes and splice variants investigated, I would suggest they provide this data. The observed lack of statistically significant differences in %exon inclusion in urine cells between DM1 and UA (Fig. 4) can easily arise from the end-point PCR not being within its dynamic range.

- Also, from Fig. 1, I am wondering how confident the authors are in their conclusion that urine cells do not show the same pattern as exRNA, since the UA DMPK signal looks like there are two groups of patients – high/low.

- I applaud the authors for the effort to collect longitudinal samples from patients and UAs for figure 3. It would be impossible/difficult to collect this type of data from repeated biopsies and this makes a very compelling case for the use of urine samples and liquid biopsies in general. However, I am a little disappointed that nothing happens over time for any of these patients, since this makes the longitudinal aspect rather uninteresting. I hope the authors will continue their efforts as they suggest and publish later on longitudinal monitoring of the effect of eteplirsen on exon skipping in urine exRNA. The current longitudinal data (only) demonstrates that the signal in urine is biologically stable and that the collection and analysis is reproducible (which is important). Caveat: This is also using the quantitative end-point PCR, which needs to be substantiated.

- In such as study I hope they will also be able to investigate the correlation of pre-eteplirsen levels of exon skipping in urine exRNA with the time matched levels of dystrophin protein levels measured by muscle biopsy. That would be a very important finding!

- For someone not already familiar with the diseases being described, I found it confusing that in the data discussing DM1 the absence of exons is a sign of the disease, but in treating patients with DMD with eteplirsen the absence of exon 51 is the desired therapeutic outcome. That may be worth flagging to the reader. It took me a few re-reads to realize.

- It is also confusing that DMD is both a gene DMD and a disease DMD. Guide the uninitiated reader.

- Remember to write out the abbreviations on first mention e.g. DMD (gene and disease) and BMD.

- I fear I may be missing something in regards to Fig. 7: If I understand the data in Fig 7a,b correctly it shows that frame-shifting deletions found in tissue biopsies from DMD patients can be confirmed by urine exRNA!? If this is correct, then the authors should clarify that the status is already known (from tissue biopsy!?) and that this is a good indication that exRNA can replace biopsy as primary diagnostic test. This would be an important finding with clinical utility.

- Figure 7c,d shows presence of exon 51 skipping transcripts in exRNA, but absence of transcripts containing exon 51 and this is taken as indication of the effect of eteplirsen. This would have been much more convincing if the authors had shown the signal from exon 51 transcripts be present in the untreated individual and then go away during treatment. Alternatively, a few UAs to show that the assay is at least working and can detect exon 51. Having near 100% effect on mRNA splicing

of treatment with eteplirsen is intriguing, but should be backed by observations in tissue. The effect on the protein level in the publication the authors reference (40) was only in the 10% range.

- Where do the spliced transcripts in urine exRNA come from? It seems likely that they come from the cells of the urinary tract (or do the authors disagree?!) and that consequently the supposed therapeutic effect seen in patients with DMD is due to the drug effect on the cells along the urinary tract and NOT in the target muscle cells. This may not matter for the use of exRNA as a biomarker of therapeutic effect, but the pharmacokinetics could be different in muscle cells and urinary tract cells and would need to be established. Again, I am looking forward to the follow-up work from these authors.

Reviewer #2 (Remarks to the Author):

In the manuscript "Analysis of extracellular mRNA in human urine reveal splice variant biomarkers of muscular dystrophies" Antoury et. al. assess exRNA from biofluids as a means to identify potential biomarkers for muscular dystrophies. They identify mis-spliced transcripts from the urine of DM1 patients via RT-PCR and digital droplet PCR, and apply this to correctly identify DM1 patients in an independent validation set. Using exRNA from urine, they are also able to identify an abnormal splicing event that is responsible for a BMD phenotype. Finally, they utilize urine exRNA to assess the efficacy of ASOs in downregulation of DMPK-CUG transcripts (by assessing correction of mis-splicing events) in DM1 as well as the success of exon skipping in DMD patients treated with the FDA approved drug, eteplirsen and identify a novel BMD-causing mutation in intron 67 of the DMD gene.

Overall the paper is thorough, well written, and has exciting implications for muscular dystrophy diagnostics. Further the ddPCR and the section on composite splicing biomarker and predictive modeling for DM1 is impressive. I have one major concern, however, and a few minor points to note.

Major concern:

- The potential use of urine exRNAs to demonstrate effective exon skipping in DMD is overstated. Particularly since the authors show that in urine exRNA from patients treated with eteplirsen, there was almost 100% efficiency for skipping of exon 51. This is in stark contrast to the data that has been obtained from muscle biopsies of these same patients where % exon skipping was much less than 100%¹. It has recently been found that the regenerative stage of the muscle is important for ASO uptake, and thus efficient dystrophin production². The extent of exon skipping seen in cells from the urinary tract may differ because of differences in uptake efficiency and therefore may not reflect what is happening in the muscle. This could be extrapolated to the ASO study in DM1 patients – while detecting a correction of splicing defects in the urine could either indicate that the muscle has also been effectively treated, or could also be a false indicator of ASO success. The authors should show a correlation between the splicing changes seen in exRNA from urine vs. RNA from muscle (both for the indicators for ASO effects in DM1 and in DMD exon skipping). If the correlation is lacking between the two, the authors should temper their discussion on the therapeutic implications of urine exRNA biomarkers.

Minor concerns:

- The Results section titled "Gene expression and alternative splicing patterns in urinary tract tissues and cells vs. muscle tissue." is confusing. The main point of this section is that the RNA detected is not coming from the muscle, but from kidney, bladder and urothelial cells. The authors should rewrite this section to clarify.

• If the novelty of the exRNA coming from non-muscle tissue is significant, the authors should include it in a main figure rather than supplement.
The re-referencing of previous figures out of order is confusing. Example: Line 189-193 where Figure 2 is re-referenced.

References Cited:

1. Mendell, J.R. et al. Eteplirsen for the treatment of Duchenne muscular dystrophy. *Annals of neurology* 74, 637-647 (2013).
2. Novak, J.S. et al. Myoblasts and macrophages are required for therapeutic morpholino antisense oligonucleotide delivery to dystrophic muscle. *Nature communications* 8, 941 (2017).

Reviewer #3 (Remarks to the Author):

The authors present in this paper novel and original findings in urine from DM1 and DMD patients. The findings could be interesting for the community, if referred to only one disease.. My major concern is that these findings would be more robust and significant if referred to only one disease as DM1 with a specific pathomolecular mechanism. For this reason, I would suggest to increase the number of patients, extending also to other DM1 phenotypes, as congenital, childhood, juvenile and late onset. I will add also data from patients with DM2. In my opinion the findings from DMD are not appropriated in this paper and could be confusing.

Reviewer #4 (Remarks to the Author):

In the manuscript by Antoury et al., the authors identify 10 transcripts that are spliced differentially in urine exRNA from DM1 patients compared to controls. The composite biomarker showed excellent specificity and sensitivity, and stability of urine exRNA splicing patterns was demonstrated over 2 years. Further, the authors demonstrate exon 51 skipping in a DMD boy treated with eteplirsen by evaluation of urine exRNA. These results pave the way for the use of urine exRNA as a noninvasive biomarker for splicing modulation in DM1 and DMD. This is an excellent manuscript of high potential impact.

Minor comments

1. Supplemental Figure 3. Please include the mean age of the UA subjects.
2. Figure 2a: Please define MDC, is this the DMD urine?
3. In Figure 7d, the authors demonstrate by dd PCR that exon 51 inclusion splice products were undetectable in a boy treated with eteplirsen. However, there are two bands shown in the RT-PCR of 7b. What is the sequence of the larger molecular weight band? Can the authors reconcile the large skipped/upskipped ratio of splice products with the very low (1%) expression of dystrophin in the eteplirsen trials?
4. In reference to Figure 8, the subject had a "clinical picture atypical of DMD." Presumably this means that the subject was more of a Becker phenotype, perhaps still walking? Please clarify.

Reviewer comments:

Reviewer #1 (Remarks to the Author):

- In this manuscript the authors show:

1. That systemic diseases affecting alternative RNA splicing can be detected in urine exRNA. To my knowledge this novel, but not too surprising, since systemic (genomic) diseases also affect the cells of the urinary tract, even if these cells are not known to display any signs of the disease. Good to have confirmed and published.
2. That splice-patterns in exRNA can be used to separate DM1 patients from UAs. This is novel and convincingly shown, including a Principal Component algorithm, which ensures perfect separation of the DM1s and UAs.
3. That RNA from urine cells cannot be shown to contain the same information. This is an important claim, and the authors provide a lot of data in support of this, that unfortunately is not completely convincing (quantitative use of end-point PCR).

Response: In the revised manuscript, we have added quantitative droplet digital PCR (ddPCR) analysis of splicing in urine cells, which confirm that the splicing patterns of all five transcripts tested are different in urine cells as compared to urine exRNA, and that the difference between means of the DM1 and UA groups in urine cells is smaller than that in urine exRNA (Figs. 5 and 6). In urine cells, *MBNL2* was the only transcript for which inclusion percentages correlated well with those urine exRNA, while correlation between urine cells and urine exRNA for the remaining four transcripts was weak (Fig. 6). The wide dynamic range of *MBNL1* splicing in urine cells that was evident in both groups by RT-PCR also was evident by ddPCR, and is in stark contrast to the narrow range and much higher exon 7 inclusion percentages evident in urine exRNA of both groups. ddPCR also confirmed a significant difference of *MBNL2* exon 6 inclusion in DM1 vs. UA groups that was identified by RT-PCR, and revealed smaller statistically significant differences in splicing of *MBNL1* and *CLASP1*, which were non-significant trends by RT-PCR. Differences in splicing of *INSR* and *MAP3K4* in urine cells were non-significant by both RT-PCR and ddPCR. Our ddPCR-based composite biomarker scores for urine cells demonstrated partial separation of DM1 and UA groups, in contrast to the wide separation for urine exRNA (Fig. 7 and Supplementary Fig. 9).

Collectively, these data indicate that the primary source of the splice products that we are measuring in the urine exRNA is not derived from the cells that appear in the urine, but suggest that urine cells could serve as a second distinct biomarker source complementary to urine exRNA. A potential limitation of using urine cells for RNA biomarkers includes the limited survival of urine cells in voided specimens, which may reduce RNA recovery in situations that prevent immediate processing of samples, and may explain why splice products in several of our urine cell samples were undetectable, even by ddPCR. This is in contrast to the long-term stability of exRNA, previously reported to be up to 30 days in urine (Martinez-Fernandez, et al., 2016), owing to its encasement in membrane-bound vesicles that provide protection from RNases.

Reference:

- Martinez-Fernandez, M., Paramio, J.M. & Duenas, M. RNA Detection in Urine: From RNA Extraction to Good Normalizer Molecules. *J Mol Diagn* **18**, 15-22 (2016).

4. I think they show that urine exRNA splice patterns match those found by tissue biopsy. If I understood this correctly this seems very significant and has important clinical applications, but the point is not very prominently positioned in the manuscript and would need the matching tissue data to be included.

Response: In Fig. 2, we use cDNA generated from commercially available total RNA that was isolated from UA skeletal muscle tissue as a control for splicing analysis by RT-PCR. Due to the high sensitivity and clinical availability of DNA testing in the U.S., muscle biopsies are no longer required for diagnosis of DM1, and consequently, are rarely clinically indicated. Diagnosis of DMD also is made primarily by genetic testing of leukocytes, with muscle biopsies typically limited to situations in which genetic testing has failed to identify a specific mutation, such as a base substitution, so that dystrophin protein can be examined and/or an alternative diagnosis considered. Thus, it is virtually impossible to obtain matched muscle and urine samples.

Fortunately, several previous publications have described alternative splicing patterns in muscle biopsies from DM1 and control subjects. In the revised manuscript, Supplementary Table 5

highlights the RT-PCR splice patterns in muscle biopsies and autopsy specimens from two previous studies, including one that used a combination of muscle biopsy and autopsy specimens to identify hundreds of candidate splicing defects in DM (Nakamori, et al., 2013). Although the exact exon inclusion percentages are different in RNA from muscle biopsy tissue as compared to urine exRNA, the significant differences between DM1 and UA subjects in splicing of *INSR*, *MBNL2*, *MBNL1*, *SOS1*, *NFIX*, *NCOR2*, and *VPS39* that we identify in urine exRNA here are similar to those in muscle tissue.

For most transcripts, alternative exon inclusion in muscle biopsies of DM1 individuals displays a wide dynamic range (Supplementary Table 5), and the reproducibility of splicing patterns in subsequent biopsies of the same muscle from the same individual is unknown. As we demonstrate in Fig. 3, urine exRNA provides a novel and renewable source of splice products that enable cost-effective and non-invasive longitudinal analysis to determine reproducibility.

5. That urine exRNA splice patterns can be used to show the therapeutic effect of eteplirsen in humans. This is hugely important, but the data presented is not convincing.

Response: In the initial submission, we used the phrase “therapeutic ASO effects.” In this context, “therapeutic” referred to the type of ASO rather than “therapeutic effects of the ASO.” We now see how this could be confusing to readers. Therefore, in the revised manuscript, we have been careful to avoid stating or implying that urine exRNA splice patterns can be used to show “therapeutic effect” of eteplirsen. Instead, the absence of exon 51 in urine exRNA and urine cells from a patient that has been treated for three years with a drug that induces removal of exon 51 from mRNA is very strong evidence of “drug target engagement.” In other words, the absence of exon 51 indicates that eteplirsen has had the intended pharmacological effect of suppressing exon 51 inclusion in urine exRNA and urine cells. A second urine specimen from this individual collected six months after the first specimen confirmed the RT-PCR results, and provided a more robust signal by ddPCR, enabling determination of the exon 51 inclusion percentages (Fig. 8).

We have found no evidence in the literature that exon 51 is alternatively spliced. To determine whether exon 51 inclusion could be reduced by a previously unknown alternative splicing event that is unique to urine exRNA and urine cells, we examined exon 51 splicing in DM1 and UA

urine samples by RT-PCR and found no evidence of a splice product that excludes exon 51 (Supplementary Fig. 16). Sequencing of PCR products confirmed the presence of exon 51 in UA samples.

Pharmacological evidence of drug target engagement is distinct from a functional therapeutic effect. We view the monitoring of *DMD* exon skipping in urine mRNA as a drug development tool (Ref: Guidance for industry and FDA staff) **primarily for novel ASOs that are designed for improved performance over eteplirsen**, with the possibility to provide an important early indicator of target engagement on a molecular level as a complement to measurements of target engagement in muscle biopsies, and, more importantly, to functional outcome measures of therapeutic efficacy.

Reference:

- Guidance for industry and FDA staff: qualification process for drug development tools. *U.S. Department of Health and Human Services, Food and Drug Administration, Center for Drug Evaluation and Research (CDER)*, 1-32 (2014).
6. A new cryptic splice site for Becker MD. This seems important in its own right and might deserve its own publication in a targeted journal for Molecular Dystrophy.

Response: We agree that it is important, and included it in the initial submission as a demonstration of the broad application of urine splice products in human urine. However, we have removed these data and will publish it elsewhere.

7. I applaud the authors for their use of UAs that are collected at the same clinic as the patients! This is really the best possible way to avoid the sample collection and pre-analytical biases that are commonly plaguing biomarker studies comparing a case and a control group! Samples collected from subjects and UAs in this study have all been treated and collected the same way and I would invite the authors to take credit for this and highlight the importance of it, for the benefit of educating the readers.

Response: Thank you for these comments.

8. The RNA quality from the nanodrop measurements in Suppl. Fig. 1 seems to be very poor, with OD 260/280 = 1!? I agree with the authors suspicion that the samples seem to be contaminated with residual Phenol/Trizol. In fact, I suspect this spectrum is primarily phenol and less RNA. For good quality RNA in this OD range I would normally suggest using the NanoDrop to estimate RNA concentration rather than the BioAnalyzer as done in S 1d, but this would clearly give a wrong estimate with this contamination. I am not sure about the Bioanalyzers sensitivity to phenol or the consequences for the concentration measurements reported.

Response: According to the Nanodrop, typical A260/280 measurements for exRNA were in the 1.5 - 1.6 range. We have included a new graph of these measurements in Supplementary Fig. 1d. We believe that the rightward shift of the RNA spectra (Supplementary Fig. 1c) and low 260/280 ratios probably represent artifact from the phenol that led to an overestimation of the exRNA concentration, in terms of ng exRNA per microliter of RNase-free water that was used to re-suspend the exRNA pellet after isolation. For this reason, we also examined exRNA integrity and concentration by capillary gel electrophoresis (Bioanalyzer). In Supplementary Figure 1g we show that Nanodrop estimates of exRNA concentration (ng/ μ l water) differed dramatically with Bioanalyzer estimates. The Nanodrop estimate of mean exRNA concentrations of DM1 urine exRNA was ~ 3-fold higher than the Bioanalyzer, and Nanodrop estimates for mean exRNA concentrations of UA urine and for serum exRNA in both groups were an order of magnitude higher than the Bioanalyzer estimates. The Nanodrop results also suggest that concentrations of exRNA (ng/ μ l water) obtained from urine and serum were similar, while the Bioanalyzer indicates that mean exRNA concentration was higher in DM1 urine than in UA urine or in serum from either group.

Using the exRNA concentration (ng exRNA/ μ l water) estimated by each method, and the volume of each biofluid specimen (milliliters of either urine or serum), we calculated the total yield of exRNA (ng) per milliliter of biofluid. In Supplementary Figure 1h, we show that the Nanodrop estimates of exRNA yield are higher in serum than in urine, and show no difference between DM1 and UA groups, while the Bioanalyzer data suggest that the exRNA yield is higher in DM1 urine than UA urine, and higher than in serum from either group.

To determine which method provides the most accurate estimation of exRNA concentration and yield for our samples, we correlated exRNA concentration, in terms of ng exRNA per microliter of RNase-free water that was used to re-suspend the exRNA, with ddPCR quantitation of *GTF2B* expression, in terms of transcript copies per microliter of cDNA that was made from the exRNA sample. As shown in Supplementary Fig. 1i, the Nanodrop measurements of exRNA concentration showed no correlation with *GTF2B* expression (Pearson correlation coefficient $r = 0.0001$, $P = 0.99$), while Bioanalyzer estimates showed good correlation ($r = 0.61$, $P < 0.0001$).

Collectively, our results suggest that, 1) the Bioanalyzer provided a more accurate estimation of exRNA concentration than the Nanodrop, 2) DM1 patients tend to excrete higher amounts of exRNA in urine than UA subjects, and 3) the wide dynamic range of *GTF2B* expression (Fig. 1) is due to the variable concentration of the exRNA that was used for cDNA synthesis.

9. There are several uses of the unit “per μL ” or “per mL”, but it rarely says per volume of what! Is it biofluid? RNA eluate? RT reaction? qPCR reaction? This is especially important in the light of the point the authors make of the concentration of RNA in urine being lower in urine than in serum, but the total recovery being higher from urine due to the higher volumes (that BTW are very variable).

Response: We have corrected this oversight by specifying the identity of the volume measured in the figures and figure legends.

10. The study seems to be using a very wide range of urine volumes for analysis – why? I understand that results are always normalized, but would it not be better to always use the same volume? What was the rationale for using a certain volume of biofluid? Was it simply all that was available?

Response: This was a novel study and we had no reference point for how much volume would be required for our purposes. Consequently, to maximize exRNA yield, we processed the entire volume of urine collected. This, in turn, has enabled us to screen splicing of nearly three-dozen transcripts to date. We agree that starting with the same volume of urine for each sample has some appeal. However, in our experience, exRNA yield is variable (as explained above), even

between samples of approximately the same volume. For application to natural history studies, or potentially to future clinical trials, a standardized protocol that specifies a minimum and maximum target volume could be designed to enable sufficient exRNA yield from each sample so that all desired splice products could be measured accurately.

11. The tissue/cell type expression data in Suppl. Fig. 7b is important to the interpretation of the data in Fig. 1. I wonder if it should be included as Fig. 1d?

Response: This is a good suggestion. We have moved the *DMPK* expression data in urinary tract tissues and cells to Fig. 1.

12. It seems the “normalizer” (*GTF2B*) is the transcript that has a higher dynamic range than the “biomarker” (*DMPK*), which makes me wonder what it is “normalizing” to? Here, especially, it would be important to know what the volume of urine is for each sample. I know it doesn’t fit with the biological story presented by the authors, but the data as presented could be interpreted as *GTF2B* really being the biomarker, and *DMPK* being the normalizer. Unlike *DMPK*, the expression of *GTF2B* is high in Urothelial cells and (it would appear) apparently different in patients with DM1 than in UAs. It would make more intuitive sense if it was the other way around.

Response: As explained in Point 8, Supplementary Fig. 1i demonstrates that *GTF2B* expression by ddPCR correlates with the concentration of the exRNA that was used to generate the cDNA for subsequent use in the ddPCR studies. The variability of *GTF2B* expression reflects a variable exRNA input from each individual sample. In this respect, the *GTF2B* expression by ddPCR could be interpreted as a “biomarker” of RNA concentration of each sample. Normalization of *DMPK* expression to reference gene *GTF2B* reduces the impact of this important source of biologic variability. Using a second quantitative PCR method, Taqman qPCR, *DMPK* mRNA expression normalized to *GTF2B*, or to a second reference gene, *GAPDH*, also appeared significantly lower in urine exRNA of DM1 patients than UA subjects (Supplementary Fig. 2). Without normalization, ddPCR and Taqman qPCR overestimate the *DMPK* content in DM1 exRNA samples.

In UA subjects, both *DMPK* alleles are normal and should be contributing equally to the exRNA pool. In DM1 subjects, *DMPK* transcripts that arise from the mutant allele contain an expanded CUG repeat and form ribonuclear inclusions. While trapped in nuclei, mutant *DMPK* transcripts are presumably less available for export from cells as exRNA, potentially reducing the overall *DMPK* content in exRNA of DM1 patients by up to 50%, with most or all of the extracellular transcripts contributed by the non-expanded allele. By contrast, isolation of total RNA from urine cells of DM1 patients would include *DMPK* transcripts from both the expanded and non-expanded alleles, and expression is expected to be similar as in UA subjects.

13. There is an impressive separation of DM1 and UA in PCA (Fig 5) – what is on PC1? It seems PC2 is not needed at all. Can the number of genes be reduced? Is there a simple “sum of splice events” underlying PC1? A linear regression analysis could reveal which gene contributes most to the separation.

Response: PC1 refers to the first principal component, which is the variable that best separates the data points into two groups, DM1 and non-DM1. In Supplementary Table 6, we have added the PCA weights for the ten-transcript RT-PCR-based composite biomarker, which shows that most transcripts contribute fairly evenly to the separation along PC1. The second principal component, PC2, further separates the data points within each group. Using the new ddPCR splicing data, we also demonstrate that a reduction from ten transcripts to five is sufficient to create a composite biomarker capable of separating the groups effectively (Fig 7).

To develop the predictive model, we used only PC1 because it accounts for most of the variance between DM1 and non-DM. The root mean square error of prediction (RMSEP) shown in Fig. 4 (previously Fig. 5) demonstrates that principal components 2 through 10 contribute little to the model. Also in Fig. 4, we have added a graph of regression coefficients, calculated as a weighted sum, which demonstrate the relative contribution of each individual transcript to the model. *MBNL2* is the transcript that is most responsible for prediction, followed by *CLASP1*, *SOS1*, and *MBNL1*.

14. Were there ANY confounding factors that could lead to the impressive separation of DM1 and UA? A discussion of potential confounding factors would be good.

Response: To help minimize potential confounding factors, samples from DM1, DMD/BMD, and UA subjects were collected in parallel, often being collected, processed, and analyzed simultaneously. In all but one case, UA samples were processed on the same day at the same time with one or more DM1 and/or DMD/BMD sample. Inclusion and exclusion criteria were identical for both study sites, and the study was conducted under a single IRB protocol (see Methods). Collection and processing of all samples was identical, and the same lab performed RNA isolation, gene expression studies, and splicing analyses for all samples. The splicing differences that we observed in urine exRNA are consistent with the known disease mechanism in clinically affected cells and tissues of DM1 patients.

15. How are the %-alternative splicing in Fig. 2-4? By scanning the gel and using end-point RT-PCR for 36 cycles as quantitative read-out!? Expecting this to be the case, I am skeptical of the data in figures 2-3-4, although the correlation to ddPCR of *INSR* in Figure 6g is very convincing (for this one gene). If the authors have any additional data to suggest that this is representative of all the other genes and splice variants investigated, I would suggest they provide this data.

Response: In the initial submission, we also included ddPCR data for the transcript *CLASP1* in Supplementary Fig. 6, which were equally impressive to the *INSR* data. To increase visibility, we have moved the graphs showing *CLASP1* inclusion percentage and the correlation with RT-PCR to Fig. 5, alongside the *INSR* data, while the remaining *CLASP1* data are now shown in Supplementary Figs. 6 and 7. In addition, we have added new ddPCR data for three additional transcripts, *MBNL2*, *MBNL1*, and *MAP3K4*, to Fig. 5 and Supplementary Figs. 6 and 7, all of which confirm a robust difference in exon inclusion between DM1 and UA groups, and correlate strongly with RT-PCR data.

For RT-PCR analysis of splicing in urine exRNA, we used 36 cycles because band intensity at 40 cycles appeared overamplified, while 34 cycles appeared under-amplified for most samples. The ddPCR validation of the RT-PCR results (Fig. 5) for all five transcripts tested argues against 36 cycles being end-point PCR for urine exRNA samples.

16. The observed lack of statistically significant differences in % exon inclusion in urine cells between DM1 and UA (Fig. 4) can easily arise from the end-point PCR not being within its dynamic range.

Response: We agree. In some urine cell samples, band intensity after 36 cycles of PCR was lower than with exRNA from the same sample, and sometimes required re-analysis at 38 or 40 cycles. Samples that required 40 cycles of PCR probably were end-point and the results expected to be less quantitative. In some other urine samples, band intensity after 36 cycles was greater than with urine exRNA from the same sample. To improve the accuracy of our measurements, we examined splicing by ddPCR in urine cells and added these data to Fig. 6, and Supplementary Figs. 9 and 10.

17. Also, from Fig. 1, I am wondering how confident the authors are in their conclusion that urine cells do not show the same pattern as exRNA, since the UA DMPK signal looks like there are two groups of patients – high/low.

Response: The range of *DMPK* expression in urine cells shown in Fig. 1 appears similar in the DM1 and UA groups. As we explain in Point 12, we expect *DMPK* expression in urine cells to be similar in DM1 and UA individuals.

18. I applaud the authors for the effort to collect longitudinal samples from patients and UAs for figure 3. It would be impossible/difficult to collect this type of data from repeated biopsies and this makes a very compelling case for the use of urine samples and liquid biopsies in general. However, I am a little disappointed that nothing happens over time for any of these patients, since this makes the longitudinal aspect rather uninteresting. I hope the authors will continue their efforts as they suggest and publish later on longitudinal monitoring of the effect of eteplirsen on exon skipping in urine exRNA. The current longitudinal data (only) demonstrates that the signal in urine is biologically stable and that the collection and analysis is reproducible (which is important). Caveat: This is also using the quantitative end-point PCR, which needs to be substantiated.

Response: DM1 progresses slowly over several decades. Evidence of significant changes in splicing patterns over a 6 - 24 month period would be concerning that the measurements are unreliable. In addition to the ddPCR confirmation of RT-PCR results, the stability of splicing patterns that we observed further substantiates the reliability of RT-PCR for urine exRNA in our study. Moving forward, longitudinal monitoring of urine exRNA splicing by ddPCR will provide the basis for important natural history studies ahead of upcoming clinical trials.

As we mention above, we believe that the real value of urine RNA splicing for *DMD* exon skipping will be for monitoring target engagement of upcoming drugs that are in development, drugs that are designed to improve uptake into skeletal muscle, either through use of a different chemical backbone or addition of a molecular conjugate.

19. In such a study I hope they will also be able to investigate the correlation of pre-etepirlisen levels of exon skipping in urine exRNA with the time matched levels of dystrophin protein levels measured by muscle biopsy. That would be a very important finding!

Response: We agree that it will be important to correlate pre-treatment exRNA splicing in urine with time-matched dystrophin protein in muscle biopsies, and look forward to the inclusion of urine exRNA splicing measurements in future clinical trials of etepirlisen, and of novel ASOs as they become available.

Now that etepirlisen has received accelerated approval from the U.S. Food and Drug Administration, patients that begin treatment with the drug outside of a clinical trial are unlikely ever to have a muscle biopsy because it is an invasive and painful procedure that offers no personal clinical benefit. However, urine RNA may provide a convenient biomarker source to demonstrate the pharmacological activity of etepirlisen, and potentially other drugs, in a post-marketing setting.

20. For someone not already familiar with the diseases being described, I found it confusing that in the data discussing DM1 the absence of exons is a sign of the disease, but in treating patients with DMD with etepirlisen the absence of exon 51 is the desired

therapeutic outcome. That may be worth flagging to the reader. It took me a few re-reads to realize.

Response: In DM1, mis-regulation of alternative splicing results from pathogenic effects of mutant *DMPK* transcripts on splicing regulator proteins. For some mis-spliced transcripts, exon inclusion is unusually high (e.g., *MBNL2*, *MBNL1*, and *MAP3K4* in our study), while exon inclusion of other transcripts is unusually low (e.g., *INSR* and *CLASP1* in our study). Therefore, DM1 is associated with higher exon inclusion of some transcripts, and lower exon inclusion of other transcripts, depending on the baseline activity of the alternative splicing regulator proteins for each transcript.

In DMD, exon skipping involves direct manipulation of splicing by the ASO, resulting in exclusion of the target exon to produce a shorter, unique mRNA product that is absent in the treated individual without drug intervention. We have stressed this distinction in the revised manuscript.

21. It is also confusing that DMD is both a gene DMD and a disease DMD. Guide the uninitiated reader.

Response: We agree it is confusing. The official symbol for the dystrophin gene is “*DMD*,” in italics. Duchenne muscular dystrophy, which results from mutations in the *DMD* gene, is abbreviated “DMD” (no italics) throughout the literature. A Reviewer of a recent paper from a different group (Bengtsson, et al., 2017) recommended adherence to this convention after, in the initial manuscript draft, the authors preferred using “dystrophin gene” instead of *DMD* to avoid confusion. We adhere to this convention, although are open to suggestions from the Editors.

Reference:

- Bengtsson, N.E. et al. Muscle-specific CRISPR/Cas9 dystrophin gene editing ameliorates pathophysiology in a mouse model for Duchenne muscular dystrophy. *Nat Commun* **8**, 14454 (2017).

22. Remember to write out the abbreviations on first mention e.g. DMD (gene and disease) and BMD.

Response: We have corrected this oversight. *DMD* gene and DMD disease abbreviations are defined on page 8 of the manuscript.

23. I fear I may be missing something in regards to Fig. 7: If I understand the data in Fig 7a,b correctly it shows that frame-shifting deletions found in tissue biopsies from DMD patients can be confirmed by urine exRNA!? If this is correct, then the authors should clarify that the status is already known (from tissue biopsy!?) and that this is a good indication that exRNA can replace biopsy as primary diagnostic test. This would be an important finding with clinical utility.

Response: In Fig. 8 (previously Fig. 7), we show that urine contains frame-shifting mRNA deletions that correspond to the *DMD* gene deletions in the DNA that were identified by genetic testing of leukocytes. By knowing the location of the DNA deletions, we were able to target the correct exons to identify the deletions in urine RNA.

24. Figure 7c,d shows presence of exon 51 skipping transcripts in exRNA, but absence of transcripts containing exon 51 and this is taken as indication of the effect of eteplirsen. This would have been much more convincing if the authors had shown the signal from exon 51 transcripts be present in the untreated individual and then go away during treatment. Alternatively, a few UAs to show that the assay is at least working and can detect exon 51. Having near 100% effect on mRNA splicing of treatment with eteplirsen is intriguing, but should be backed by observations in tissue. The effect on the protein level in the publication the authors reference (40) was only in the 10% range.

Response: We agree that examining exon 51 inclusion in urine exRNA and urine cells pre- and post-treatment, and correlation with splicing in time-matched muscle biopsy tissue would be ideal. However, this individual has never had a muscle biopsy, and now that he is being treated outside of a clinical trial, is very unlikely ever to have one.

As we explain in Point 5, Supplementary Fig. 16 shows exon 51-containing transcripts in urine of DM1 and UA subjects using RT-PCR and three separate primer sets, without evidence of an alternative splice event involving exon 51.

In the clinical trial that led to accelerated approval of eteplirsen, immunofluorescence analysis of muscle biopsies demonstrated an increase in the number of muscle fibers expressing detectable dystrophin protein from a pre-treatment baseline of 1.12% to 17.39% after 180 weeks of treatment (Ref: FDA briefing document, April 25, 2016). Exon skipping at the mRNA level was examined in these muscle biopsies using qualitative end-point RT-PCR and sequencing of PCR products; therefore, precise quantification of exon skipping activity in these biopsies is unavailable.

Long-term effects of morpholino ASO target engagement in human urinary tract tissues are unknown. Our finding that exon 51 inclusion in urine exRNA is only ~ 10% (or ~ 90% skipped) by ddPCR was surprising, and suggests that eteplirsen has greater uptake in cells and tissues lining the urinary tract, and in urine cells, than in deltoid or biceps muscle tissue (the two muscles that were examined in the clinical trials that led to accelerated approval of eteplirsen). This may be explained by pharmacokinetic properties of uncharged phosphorodiamidate morpholino ASOs like eteplirsen, which are excreted rapidly after systemic delivery, with little or negligible muscle tissue uptake (Bennett, et al., 2017). We discuss the pharmacokinetic properties of ASOs further in Point 25.

Reference:

- Bennett, C.F., Baker, B.F., Pham, N., Swayze, E. & Geary, R.S. Pharmacology of Antisense Drugs. *Annu Rev Pharmacol Toxicol* **57**, 81-105 (2017).
- www.fda.gov/downloads/advisorycommittees/committeesmeetingmaterials/drugs/peripheralandcentralnervoussystemdrugsadvisorycommittee/ucm497064.pdf

25. Where do the spliced transcripts in urine exRNA come from? It seems likely that they come from the cells of the urinary tract (or do the authors disagree?!) and that consequently the supposed therapeutic effect seen in patients with DMD is due to the drug effect on the cells along the urinary tract and NOT in the target muscle cells. This may not matter for the use of exRNA as a biomarker of therapeutic effect, but the

pharmacokinetics could be different in muscle cells and urinary tract cells and would need to be established. Again, I am looking forward to the follow-up work from these authors.

Response: Based on the similar expression of *DMPK* in urine exRNA and urinary tract tissues (Fig. 1), the similar expression of skeletal actin, *ACTA1*, in urine exRNA and urinary tract tissues (Supplementary Fig. 11), splicing patterns in urine exRNA that appear different than in control UA muscle tissue (Fig. 2 and Supplementary Table 5), and the similarity of urine exRNA splice patterns with those in urinary tract cells and tissues (Supplementary Fig. 12), we believe that splice products in urine exRNA represent a pool released from cells lining the urinary tract.

The pharmacokinetic properties of ASOs are determined mostly by the chemistry of its backbone linkage, and are largely independent of sequence within a chemical class (Bennett, et al., 2017). An ASO with an uncharged linkage, including a morpholino ASO such as eteplirsen, exhibits rapid clearance from the blood, resulting in minimal muscle tissue uptake (Bennett, et al., 2017). By contrast, ASOs with a phosphorothioate linkage demonstrate activity in normal muscle that is similar to activity in kidney of non-human primates (Pandey, et al., 2015), and in wild-type mouse muscle, bladder, and kidney (Supplementary Fig. 15). We agree that pharmacokinetic properties of either charged or uncharged ASOs could be different in urine than in muscle or urinary tract tissues, and look forward to examination of urine and muscle biopsy splice products together in future clinical trials.

References:

- Bennett, C.F., Baker, B.F., Pham, N., Swayze, E. & Geary, R.S. Pharmacology of Antisense Drugs. *Annu Rev Pharmacol Toxicol* **57**, 81-105 (2017).
- Pandey, S.K. et al. Identification and characterization of modified antisense oligonucleotides targeting *DMPK* in mice and nonhuman primates for the treatment of myotonic dystrophy type 1. *J Pharmacol Exp Ther* **355**, 329-340 (2015).

Reviewer #2 (Remarks to the Author):

In the manuscript “Analysis of extracellular mRNA in human urine reveal splice variant biomarkers of muscular dystrophies” Antoury et. al. assess exRNA from biofluids as a means to identify potential biomarkers for muscular dystrophies. They identify mis-spliced transcripts

from the urine of DM1 patients via RT-PCR and digital droplet PCR, and apply this to correctly identify DM1 patients in an independent validation set. Using exRNA from urine, they are also able to identify an abnormal splicing event that is responsible for a BMD phenotype. Finally, they utilize urine exRNA to assess the efficacy of ASOs in downregulation of DMPK-CUG transcripts (by assessing correction of mis-splicing events) in DM1 as well as the success of exon skipping in DMD patients treated with the FDA approved drug, eteplirsen and identify a novel BMD-causing mutation in intron 67 of the DMD gene. Overall the paper is thorough, well written, and has exciting implications for muscular dystrophy diagnostics. Further the ddPCR and the section on composite splicing biomarker and predictive modeling for DM1 is impressive. I have one major concern, however, and a few minor points to note.

References Cited:

1. Mendell, J.R. et al. Eteplirsen for the treatment of Duchenne muscular dystrophy. *Annals of neurology* 74, 637-647 (2013).
2. Novak, J.S. et al. Myoblasts and macrophages are required for therapeutic morpholino antisense oligonucleotide delivery to dystrophic muscle. *Nature communications* 8, 941 (2017).

Major concern:

1. The potential use of urine exRNAs to demonstrate effective exon skipping in DMD is overstated. Particularly since the authors show that in urine exRNA from patients treated with eteplirsen, there was almost 100% efficiency for skipping of exon 51. This is in stark contrast to the data that has been obtained from muscle biopsies of these same patients where % exon skipping was much less than 100%

Response: As we indicate above in Point 24 for Reviewer 1, the % exon skipping in muscle biopsies of clinical trial participants is unknown, because it was evaluated using qualitative end-point RT-PCR and sequencing of PCR products (Ref: FDA briefing document, April 25, 2016). Droplet digital PCR (ddPCR) is a state-of-the-art technology that enables precise quantification of mRNA exon inclusion percentages, and its implementation has been suggested for quantification of exon skipping efficiencies in drug development programs for DMD as a complement to dystrophin protein measurements (Verheul, et al., 2016). If RNA from these biopsies remains available, ddPCR presumably could be used to determine the precise % exon skipping. In addition, it's unclear whether any urine samples were collected from these patients at the time the biopsies were performed, or, if so, whether any of these urine specimens remain

available for RNA isolation and splicing analysis. We would be eager to examine splice products in muscle biopsies and urine liquid biopsies from the same individual as soon as they are made available to us (see Point 3 below).

The individual in our study who is being treated with eteplirsen is non-ambulatory and began receiving the ASO in December 2014 as a participant in an open label safety and tolerability clinical trial of eteplirsen in advanced DMD. Due to accelerated approval of eteplirsen by the FDA, he has continued receiving weekly infusions of the drug after the clinical trial primary completion date in April 2017. This individual has never had a muscle biopsy, and is unlikely ever to have one because it is an invasive and painful procedure that offers no personal clinical benefit. In lieu of a muscle biopsy, we collected a urine “liquid biopsy” to determine whether splice products in urine mRNA could detect pharmacological activity of eteplirsen. Quantification of exon 51 skipping by droplet digital PCR (ddPCR) and sequencing of RT-PCR products in urine exRNA and urine cells (Fig. 8) are the only measurements of pharmacological activity of eteplirsen that are available in this patient.

The controversy surrounding the FDA accelerated approval of eteplirsen (Aartsma-Rus and Krieg, 2017) indicates that realization of the full potential of exon skipping for DMD will require the development of new ASOs that outperform eteplirsen. Novel ASOs that demonstrate similar activity in muscle tissue and urinary tract tissue, such as conjugated morpholinos (Morcos, et al., 2008) or ASOs that have phosphorothioate backbones (Pandey, et al., 2015), may enable monitoring of pharmacological activity in urine RNA. In Supplementary Fig. 15, we show that systemic delivery of a *Dmpk*-targeting phosphorothioate ASO demonstrates similar activity in wild-type mouse kidney, bladder, and skeletal muscle after a four-week course of therapy. This confirms a previous study that also showed similar activity in kidney and muscle tissues of non-human primates after systemic delivery of the same *DMPK*-targeting ASO over a 12-week period (Pandey, et al., 2015). If an ASO designed for treatment of either DMD or DM1 demonstrates pharmacological similar activity in skeletal muscle and urinary tissues over a several week course of therapy, and the urinary tract tissues produce urine exRNA, then we believe that it is reasonable to conclude that drug effects on splice products in urine exRNA have the **potential** to correlate with those in muscle biopsy tissue, although the exact exon inclusion/exclusion pattern may be different between these two RNA sources. Please also see our response to Point 5 for Reviewer 1.

In the revised manuscript, we have re-worded the *DMD* exon-skipping portion of the Discussion as follows:

“For DMD, the urine splice products are more than traditional biomarkers: they are personalized genetic markers that can be designed specifically for each individual patient and enable the possibility to monitor splice-shifting activity of ASOs. Our finding of ASO-dependent exon-skipping activity in urine exRNA and urine cell total RNA provides the first non-invasive measurement of eteplirsen target engagement, and suggest that our urine biomarkers have the potential to facilitate development of novel ASOs that can outperform eteplirsen or target new exons in DMD. At this point, it is unlikely that an RNA-based assay will eliminate the need for muscle biopsies, as dystrophin protein measurement was used as a surrogate marker of drug effect that led to the accelerated approval of eteplirsen by the U.S. Food and Drug Administration (Aartsma-Rus and Krieg, 2017). However, ddPCR-based monitoring of skipped/unskipped *DMD* splice products in urine during the course of treatment may complement splicing analysis of muscle biopsies (Verhuel, et al. 2016) as newer and better splice-shifting drugs are developed. Long-term, it will be important to correlate these early indicators of drug target engagement with imaging, compositional, and functional outcome measures to determine whether a drug that is working on a molecular level also is having a therapeutic benefit.”

References:

- Aartsma-Rus, A. & Krieg, A.M. FDA Approves Eteplirsen for Duchenne Muscular Dystrophy: The Next Chapter in the Eteplirsen Saga. *Nucleic Acid Ther* **27**, 1-3 (2017).
- Morcos, P.A., Li, Y. & Jiang, S. Vivo-Morpholinos: a non-peptide transporter delivers Morpholinos into a wide array of mouse tissues. *Biotechniques* **45**, 613-614, 616, 618 passim (2008).
- Pandey, S.K. et al. Identification and characterization of modified antisense oligonucleotides targeting DMPK in mice and nonhuman primates for the treatment of myotonic dystrophy type 1. *J Pharmacol Exp Ther* **355**, 329-340 (2015).
- Verheul, R.C., van Deutekom, J.C. & Datson, N.A. Digital Droplet PCR for the Absolute Quantification of Exon Skipping Induced by Antisense Oligonucleotides in (Pre-)Clinical Development for Duchenne Muscular Dystrophy. *PLoS One* **11**, e0162467 (2016).
- www.fda.gov/downloads/advisorycommittees/committeesmeetingmaterials/drugs/peripheralandcentralnervoussystemdrugsadvisorycommittee/ucm497064.pdf

2. It has recently been found that the regenerative stage of the muscle is important for ASO uptake, and thus efficient dystrophin production

Response: The Novak, et al., reference you cite highlights an important limitation of unconjugated phosphorodiamidate morpholino ASOs, like eteplirsen, for targeting skeletal muscle tissue. In addition, two previous studies found that uptake of unconjugated morpholino ASOs after local intramuscular injection of DM1 mouse models, which feature a non-necrotizing myopathy similar to DM1 patients, was limited to fibers that were injured by the needle that injected the ASO, requiring concomitant electroporation to achieve sufficient ASO activity in treated muscles (Wheeler, et al., 2007; Wheeler, et al., 2009). To achieve sufficient activity of systemically delivered morpholino ASOs in muscle tissue of DM1 mice required the addition of a peptide conjugate to the morpholino ASO that facilitates muscle tissue uptake of the drug (Leger, et al., 2013). Low activity of systemically delivered unconjugated morpholinos in DM1 mouse muscle and unacceptable toxicity of peptide-linked morpholinos thus far have limited their clinical application for treatment of DM1.

The importance of active muscle regeneration for ASO uptake may be unique to phosphorodiamidate morpholinos, and perhaps other ASOs that also have an uncharged backbone. For example, ASOs that have a charged phosphorothioate backbone display a substantial pharmacokinetic benefit over ASOs like morpholinos that have uncharged linkages because of increased binding to plasma proteins, which facilitates delivery to target tissues and prevents rapid excretion via the kidney (Bennett, et al., 2017). Systemic delivery of phosphorothioate ASOs has shown strong activity in skeletal muscle of wild-type mice, DM1 mice, and healthy non-human primates, and in urinary tract tissues of wild-type mice and healthy non-human primates (Wheeler, et al., 2012; Pandey, et al., 2015; Supplementary Fig. 15), demonstrating that tissue regeneration is unnecessary for efficient uptake and activity of these ASOs.

References:

- Bennett, C.F., Baker, B.F., Pham, N., Swayze, E. & Geary, R.S. Pharmacology of Antisense Drugs. *Annu Rev Pharmacol Toxicol* **57**, 81-105 (2017).
- Leger, A.J. et al. Systemic delivery of a Peptide-linked morpholino oligonucleotide neutralizes mutant RNA toxicity in a mouse model of myotonic dystrophy. *Nucleic Acid Ther* **23**, 109-117 (2013).

- Pandey, S.K. et al. Identification and characterization of modified antisense oligonucleotides targeting DMPK in mice and nonhuman primates for the treatment of myotonic dystrophy type 1. *J Pharmacol Exp Ther* **355**, 329-340 (2015).
 - Wheeler, T.M., Lueck, J.D., Swanson, M.S., Dirksen, R.T. & Thornton, C.A. Correction of CIC-1 splicing eliminates chloride channelopathy and myotonia in mouse models of myotonic dystrophy. *J Clin Invest* **117**, 3952-3957 (2007).
 - Wheeler, T.M. et al. Reversal of RNA dominance by displacement of protein sequestered on triplet repeat RNA. *Science* **325**, 336-339 (2009).
 - Wheeler, T.M. et al. Targeting nuclear RNA for in vivo correction of myotonic dystrophy. *Nature* **488**, 111-115 (2012).
3. The extent of exon skipping seen in cells from the urinary tract may differ because of differences in uptake efficiency and therefore may not reflect what is happening in the muscle. This could be extrapolated to the ASO study in DM1 patients – while detecting a correction of splicing defects in the urine could either indicate that the muscle has also been effectively treated, or could also be a false indicator of ASO success. The authors should show a correlation between the splicing changes seen in exRNA from urine vs. RNA from muscle (both for the indicators for ASO effects in DM1 and in DMD exon skipping). If the correlation is lacking between the two, the authors should temper their discussion on the therapeutic implications of urine exRNA biomarkers.

Response: The pharmacokinetic properties of phosphorodiamidate or phosphorothioate ASOs in urine exRNA and urine cells are unknown, and we agree that it will be important to correlate them with ASO pharmacokinetic properties in muscle biopsy tissue (see Point 25 for Reviewer 1). However, patients receiving eteplirsen outside of a clinical trial are unlikely ever to have a muscle biopsy because it is an invasive and painful procedure that offers no personal clinical benefit. Therefore, correlation of drug target engagement in urine exRNA and time-matched muscle biopsies almost certainly will occur only in the context of a clinical trial that requires muscle biopsies for all participants. We have no access to any biopsy material from any clinical trials. On numerous occasions over the past 20 months, we have contacted Sarepta Pharmaceuticals, the sponsor of the only clinical trials for eteplirsen and newer morpholino ASOs targeting exons 45 and 53, in hopes of obtaining urine samples from patients enrolled in an active clinical trial. Although they have yet to demonstrate an interest in collaboration, or to

provide urine samples from trial participants for our studies, we look forward to being able to evaluate the correlation of muscle biopsy splicing with urine splicing results, and are hopeful that matched samples will be made available to us in the future.

It's worth noting that quantification of dystrophin protein in muscle biopsies, which was used by the FDA as a "surrogate endpoint that is reasonably likely to predict clinical benefit," also is a measurement of pharmacological activity (successful skipping of exon 51 at the RNA level produces an internally truncated, partially functional protein [Aartsma-Rus and van Ommen, 2010]), but is not a measurement of clinical benefit. Long-term, it will be important to correlate pharmacological activity in urine "liquid biopsies" and muscle biopsies with functional outcome measures to determine whether a drug that is working on a molecular level also is having a therapeutic benefit.

The U.S. Food and Drug Administration's decision to grant eteplirsen accelerated approval was controversial, largely due to disagreement over whether the degree of pharmacological activity, measured as dystrophin protein production in muscle biopsies of treated patients, was "reasonably likely" to predict eventual therapeutic benefit (Aartsma-Rus and Krieg 2017). A better exon skipping drug for DMD is needed, one that demonstrates pharmacological activity and functional therapeutic benefits that are obvious to all. Novel, non-invasive measurements of pharmacological activity that complement dystrophin protein measurements in muscle tissue, may facilitate the development new exon-skipping ASOs for DMD that outperform eteplirsen.

ASOs in development for DM1 have a phosphorothioate linkage and show similar activity in skeletal muscle and urinary tract tissues (Pandey, et al., 2015; Supplementary Fig. 15). For novel ASOs developed for treatment of DMD or DM1 that engage its mRNA target in muscle and urinary tract tissues with similar efficacy, we believe that the downstream effects of splicing modulation in muscle tissue RNA have the potential to correlate with splicing modulation evident in urine RNA. Similarly, the absence of detectable pharmacological activity in urine RNA also could serve as an early indicator that the dose may be too low, or that the candidate drug eventually will fail, and thereby could help save valuable resources. These are important and potentially overlooked advantages of non-invasive monitoring of drug target engagement.

References:

- Aartsma-Rus, A. & van Ommen, G.J. Progress in therapeutic antisense applications for neuromuscular disorders. *Eur J Hum Genet* **18**, 146-153 (2010).
- Aartsma-Rus, A. & Krieg, A.M. FDA Approves Eteplirsen for Duchenne Muscular Dystrophy: The Next Chapter in the Eteplirsen Saga. *Nucleic Acid Ther* **27**, 1-3 (2017).
- Bennett, C.F., Baker, B.F., Pham, N., Swayze, E. & Geary, R.S. Pharmacology of Antisense Drugs. *Annu Rev Pharmacol Toxicol* **57**, 81-105 (2017).
- Pandey, S.K. et al. Identification and characterization of modified antisense oligonucleotides targeting DMPK in mice and nonhuman primates for the treatment of myotonic dystrophy type 1. *J Pharmacol Exp Ther* **355**, 329-340 (2015).

Minor concerns:

4. The Results section titled “Gene expression and alternative splicing patterns in urinary tract tissues and cells vs. muscle tissue.” is confusing. The main point of this section is that the RNA detected is not coming from the muscle, but from kidney, bladder and urothelial cells. The authors should rewrite this section to clarify.

Response: Yes, it was confusing. We have re-titled the section, “Gene expression and alternative splicing patterns in the human urinary tract.”

5. If the novelty of the exRNA coming from non-muscle tissue is significant, the authors should include it in a main figure rather than supplement.

Response: This is a good suggestion, one that Reviewer 1 also raised (Point 11). We have moved the data showing *DMPK* expression in urinary tract tissues to Fig. 1.

6. The re-referencing of previous figures out of order is confusing. Example: Line 189-193 where Figure 2 is re-referenced.

Response: We re-referenced Fig. 2 in this position because it provided context near the end of the Discussion section that helped us during preparation of the manuscript, but agree that it could be confusing. Here we have omitted the Fig. 2 re-reference.

Reviewer #3 (Remarks to the Author):

The authors present in this paper novel and original findings in urine from DM1 and DMD patients. The findings could be interesting for the community, if referred to only one disease.

My major concern is that these findings would be more robust and significant if referred to only one disease as DM1 with a specific pathomolecular mechanism. For this reason, I would suggest to increase the number of patients, extending also to other DM1 phenotypes, as congenital, childhood, juvenile and late onset. I will add also data from patients with DM2. In my opinion the findings from DMD are not appropriated in this paper and could be confusing.

Response: The use of urine exRNA as biomarkers of muscular dystrophies is counterintuitive. We include the DM and DMD data together to demonstrate and highlight the broad application of our novel approach to biomarker development that will be of interest beyond the muscular dystrophy community.

The suggestion to extend analysis to the full spectrum of DM1 phenotypes, and to include data from DM2 patients, is a good one. In the revised manuscript, we add four DM1 and four DM2 patients, for an overall number of 42 DM patients. Our control group includes 28 UA and 15 DMD/BMD, for an overall total of 43 non-DM subjects. DM1 subgroups in our study includes N = 4 congenital, 9 childhood/juvenile onset, 22 adult-onset, and 3 asymptomatic DM1, and N = 4 DM2, which is a good representation of the overall DM population. Congenital DM1 is a rare and severe form of DM1, while DM2 patients may be under-diagnosed due to the symptoms that typically are milder than DM1 (Suominen, et al., 2011), accounting for the lower numbers for these two groups.

The quantitation of splicing outcomes by ddPCR that we include in the revised manuscript enabled us to analyze splicing outcomes relative to DM1 phenotype and symptoms, finding that asymptomatic DM1 individuals feature an early change of splicing patterns of *MBNL2*, *MBNL1*, and *CLASP1*, while splicing patterns of *INSR* and *MAP3K4* appear correlated with greater symptom severity (Fig. 7, Supplementary Fig. 9). By combining the splicing outcomes for all five transcripts together to form a single composite biomarker, the statistical power of the

difference between the symptomatic DM1, asymptomatic DM1, and UA groups is overwhelmingly robust.

Reference:

- Suominen, T. et al. Population frequency of myotonic dystrophy: higher than expected frequency of myotonic dystrophy type 2 (DM2) mutation in Finland. *Eur J Hum Genet* **19**, 776-782 (2011).

Reviewer #4 (Remarks to the Author):

In the manuscript by Antoury et al., the authors identify 10 transcripts that are spliced differentially in urine exRNA from DM1 patients compared to controls. The composite biomarker showed excellent specificity and sensitivity, and stability of urine exRNA splicing patterns was demonstrated over 2 years. Further, the authors demonstrate exon 51 skipping in a DMD boy treated with eteplirsen by evaluation of urine exRNA. These results pave the way for the use of urine exRNA as a noninvasive biomarker for splicing modulation in DM1 and DMD. This is an excellent manuscript of high potential impact.

Minor comments

1. Supplemental Figure 3. Please include the mean age of the UA subjects.

Response: Due to privacy considerations, we have no mean age data for the UA subjects. However, all UA subjects were adults, either a spouse of a DM1 or DM2 participant, or the parent of a DMD, BMD, or young DM1 participant. The mean age for the UA subjects is likely similar to, or perhaps slightly older, than that for the DM1 group.

2. Figure 2a: Please define MDC, is this the DMD urine?

Response: Yes, this should have read DMD, and has been corrected.

3. In Figure 7d, the authors demonstrate by dd PCR that exon 51 inclusion splice products were undetectable in a boy treated with eteplirsen. However, there are two bands shown in the RT-PCR of 7b. What is the sequence of the larger molecular weight band? Can the authors reconcile the large skipped/upskipped ratio of splice products with the very low (1%) expression of dystrophin in the eteplirsen trials?

Response: We were unable to isolate enough DNA from the upper band to sequence it. However, the upper band is the expected size for a PCR product that was generated using primers targeting exons 44 and 52 in the context of an exon 45 - 50 deletion. The second sample from this individual resulted in stronger ddPCR signal that enabled quantitation of exon 51 inclusion (Fig. 8c-e). The primer probe set that we used in the ddPCR assay consists of a left primer targeting exon 51, a right primer targeting exon 52, and a fluorescent probe that targets the exon 51 - 52 splice site (binding to the 3' end of exon 51 and the 5' end of exon 52) and, therefore, is highly specific for the exon 51 inclusion splice product (Verheul, 2016).

The large percentage of exon skipping that we found in urine RNA suggests that eteplirsen has greater uptake into cells and tissues lining the urinary tract than in deltoid or biceps muscle tissue, which were the muscles biopsied during the clinical trial. We discuss this further in Points 24 and 25 for Reviewer 1.

4. In reference to Figure 8, the subject had a “clinical picture atypical of DMD.” Presumably this means that the subject was more of a Becker phenotype, perhaps still walking? Please clarify.

Response: We removed these data and will publish it separately.

Reviewers' comments:

Reviewer #1 (Remarks to the Author):

This version is a significant improvement over the previous and the entire manuscript appears much more cohesive and reads much better.

The addition of ddPCR improves confidence in the data throughout.

I appreciate everything the authors do to characterize the RNA preparations by OD and Bioanalyzer, but I think there is no question the RNA preparations are contaminated with phenol. This makes all the conclusions drawn based on these measurements in the entire first section of results "Characterization of exRNA..." rather questionable and I would suggest to omit these results. They are not required for the remainder of the paper since the PCR/amplification analysis appear to work well, despite the phenol contamination. Drawing conclusions on the quantity of RNA from the various samples or which technical method is best for measuring the RNA will only distract from the main message. The RNA quality can be discussed as a limitation, but does not appear to impair the PCR results.

I appreciate the addition of an additional reference gene (GAPDH), which confirms the previous one (GTF2B), but still find it puzzling that absolute expression level of either of the reference genes seem to be able to correctly separate the patients from the UAs since the reference genes are the ones that display differential expression between the groups. What was the amount of sample used in these PCRs? Were they all performed on the same volume of exRNA or cDNA? Or were they normalized for RNA amount (based on the flawed RNA concentration measurements with phenol contamination)?

The authors examined 33 mRNA candidates and found 10 that were different between UA and DM1/DMD, but the DMD mRNA is not among them. Why? Later in the manuscript a lot of focus is on splicing of DMD-mRNA and it seems strange that DMD-mRNA did not show up in the original screen. Why is the data in Fig 8 not part of Fig 1? The authors should discuss this (or even better add the data for DMD-mRNA in Figure 1ff).

And why is DMD the only gene where every patient has a different aberrant splice pattern, whereas the other genes (in Fig1) seem to be conserved between patients. Discuss.

Since DMD is a genomic disease (mutations in the DMD gene), but severity / phenotype depends on the degree of aberrant splicing of key mRNAs. The authors should briefly describe/mention the potential utility of a urine based test beyond the use as a therapy monitoring tool.

The manuscript now makes a convincing claim that the mRNA splice patterns in urine EVs, urine cells and serum EVs are different and that only the urine EV splice patterns correlate with the disease. The conclusion seems to be that the urine EVs to a significant extent come from cells that are affected by the disease, e.g. because expression of the relevant disease genes is high. This does not seem to be the case for the cells found in urine and also not for the majority of EVs found in serum. I think the authors have enough circumstantial evidence to speculate about which cells the urine EVs might come from, but it is never clearly called out. Which tissue is the main contributor to EVs in urine? The authors could be a little more clear about whether they think EVs from muscle cells (where the disease is having its main manifestation) is not present in serum or whether they are simply diluted out by other cells.

Where does the ASO go in the body? Is it administered IV? Then what? Where does the drug go in the body? Maybe that could suggest where it might have the biggest effect.

All the PCR data used to separate UAs from DM1/DMD is on %Splice-variant, which works nice. However, it would be good to mention (discussion?) whether simple expression levels of the relevant genes had any diagnostic performance at all.

It is really too bad that there is nothing at all to compare the data for DMD mRNA in Figure 8b to. All of the gels in Figure 8a are supposed to be evidence of DMD aberrant splicing that might be helped by ASO therapy, but no therapy is given so there is no evidence of any change in urine exRNA as a result of therapy. Similarly, the data in Fig 8b is during treatment with ASO, but there is no evidence of what the splice pattern was before therapy, so again, no evidence of an effect. I acknowledge that this data is not easy to get to and that longitudinal monitoring of patients during ASO treatment is a tall order. Unfortunately, the very impressive collection of

longitudinal collections in Figure 3 does not show any sign of change (and it is not discussed whether patients are on ASO treatment during collection, but I suspect not). Again, I wonder why the DMD gene is not included in this gene set. Did none of these subjects have frame shifting deletions in DMD?

What are the different lanes in the gels in Fig8b? Why are there bands in two of 5 lanes in the 2nd panel?

The observation that the urine cells from subject S7 shows almost the same %exon11 inclusion runs counter to the observation from previously in the manuscript that urine EVs and urine cells show different splice patterns. Maybe analysis of serum EVs from S7 would show the 44-51-52 variant more clearly, which would support the claim that urine EVs are the better source for measuring these aberrant transcripts, than serum EVs or urine cells.

Without any kind of comparison I think it is a stretch to call it "evidence of ASO activity"... it is very likely the case, but...

Reviewer #2 (Remarks to the Author):

The authors have addressed the reviewer points sufficiently. I have no additional points to raise.

Reviewer #3 (Remarks to the Author):

The paper is very much improved and I am satisfied.

Reviewer #4 (Remarks to the Author):

The authors have addressed my concerns

Reviewers' comments:

Reviewer #1 (Remarks to the Author):

1. This version is a significant improvement over the previous and the entire manuscript appears much more cohesive and reads much better. The addition of ddPCR improves confidence in the data throughout.
2. I appreciate everything the authors do to characterize the RNA preparations by OD and Bioanalyzer, but I think there is no question the RNA preparations are contaminated with phenol. This makes all the conclusions drawn based on these measurements in the entire first section of results “Characterization of exRNA...” rather questionable and I would suggest to omit these results. They are not required for the remainder of the paper since the PCR/amplification analysis appear to work well, despite the phenol contamination. Drawing conclusions on the quantity of RNA from the various samples or which technical method is best for measuring the RNA will only distract from the main message. The RNA quality can be discussed as a limitation, but does not appear to impair the PCR results.

Response: We think it is important to include these results precisely to alert readers that nanospectrophotometry is inaccurate for characterization and quantification of exRNA using our methods. This was confusing to us early in our studies, and we want to help others avoid the problems that we had interpreting the exRNA readings. We also demonstrate that capillary gel electrophoresis provides an accurate estimation of exRNA concentration that, in stark contrast to nanospectrophotometry, correlates well with ddPCR quantification of reference gene *GTF2B* expression (Supplementary Fig. 1i).

3. I appreciate the addition of an additional reference gene (GAPDH), which confirms the previous one (GTF2B), but still find it puzzling that absolute expression level of either of the reference genes seem to be able to correctly separate the patients from the UAs since the reference genes are the ones that display differential expression between the groups. What was the amount of sample used in these PCRs? Were they all performed on the

same volume of exRNA or cDNA? Or were they normalized for RNA amount (based on the flawed RNA concentration measurements with phenol contamination)?

Response: We addressed this issue in Point 12 of the first Response. exRNA concentration correlates well with *GTF2B* expression by ddPCR (Supplementary Fig. 1), indicating that the differential expression of reference genes *GTF2B* and *GAPDH* is due to higher exRNA concentration in DM1 urine. In addition to these two reference genes, the total expression level of *MBNL2*, *MBNL1*, *MAP3K4*, *CLASP1*, and *INSR* by ddPCR, based on copies per microliter of the exon inclusion splice product plus the copies per microliter of the exon exclusion splice product, is higher in DM1 individuals than UA controls (Supplementary Fig. 6e). In all, six transcripts that we examined by ddPCR, plus *GAPDH* by qPCR, for a total of seven transcripts, are expressed at higher levels in DM1 individuals than UA controls, consistent with a higher total exRNA content in DM1 urine. This is an interesting disease manifestation that was previously unknown.

DM1 is autosomal dominant: the mutant polyadenylated *DMPK*-CUG^{exp} transcripts (50% of the total) form intranuclear inclusions (Taneja et al., 1995), while the *DMPK* transcripts that arise from the normal non-expanded allele (50% of the total) are transported to the cytoplasm. In individuals without DM1, *DMPK* transcripts from both alleles contain non-expanded CUG repeats, none of these transcripts form nuclear inclusions, and all are transported to the cytoplasm. While the mutant *DMPK*-CUG^{exp} transcripts are trapped in nuclear inclusions, they are unavailable for release from cells as exRNA. This could result in up to 50% fewer *DMPK* transcripts released as exRNA in DM1 individuals as compared to non-DM1 controls, which would explain the lower expression of *DMPK* mRNA in urine exRNA as compared to UA subjects that we observed (Fig. 1c).

All of the ddPCR results are shown as per microliter of cDNA; all qPCR results were determined using one microliter of cDNA from each sample.

Reference:

- Taneja, K.L., McCurrach, M., Schalling, M., Housman, D. & Singer, R.H. Foci of trinucleotide repeat transcripts in nuclei of myotonic dystrophy cells and tissues. *J Cell Biol* **128**, 995-1002 (1995).

4. The authors examined 33 mRNA candidates and found 10 that were different between UA and DM1/DMD, but the DMD mRNA is not among them. Why?

Response: We found ten alternative splice events that were significantly different in urine exRNA of DM1 patients as compared to DMD or UA controls. As we show in Figure 4a and Figure 5d, alternative splicing in DMD patients is similar to UA controls. Alternative splicing of *DMD* exon 71 and exon 78 is reported to be abnormal in muscle biopsy tissue of DM1 patients (Nakamori, et al., 2013). The similar splicing pattern of these alternative *DMD* exons in DM1 and UA exRNA (Supplementary Fig. 3) suggests that the regulation of these exons is different in urine exRNA than in muscle tissue. Almost all of the remaining 79 *DMD* exons are constitutively spliced and, therefore, are expected to be identical in DM1, DMD, and UA groups (Bouge, et al., 2017).

Reference:

- Bouge, A.L. et al. Targeted RNA-Seq profiling of splicing pattern in the DMD gene: exons are mostly constitutively spliced in human skeletal muscle. *Sci Rep* **7**, 39094 (2017).
 - Nakamori, M. et al. Splicing biomarkers of disease severity in myotonic dystrophy. *Ann Neurol* **74**, 862-872 (2013).
5. Later in the manuscript a lot of focus is on splicing of DMD-mRNA and it seems strange that DMD-mRNA did not show up in the original screen. Why is the data in Fig 8 not part of Fig 1? The authors should discuss this (or even better add the data for DMD-mRNA in Figure 1ff).

Response: It's unclear what you mean by "original screen" in the context of *DMD* mRNA. In Figure 8, we used RT-PCR to demonstrate that *DMD* deletion transcripts are routinely detectable in urine exRNA, and can serve as personalized genetic markers (mRNA copies of the DNA gene deletion for each individual) (see Point 6 below). Personalized genetic markers of individual DMD patients have no relationship to the aberrant alternative splicing in exRNA of DM1 individuals, other than urine exRNA can be used to identify both.

In Figure 1, we used ddPCR to determine the expression of *DMPK*, the mutated gene in DM1, and reference gene *GTF2B* in urine exRNA, urine cells, serum exRNA of DM1, bladder tissue total RNA, urothelial cell total RNA, kidney tissue total RNA, and muscle tissue total RNA. *DMPK* expression is unrelated to *DMD* gene deletions. There is no Figure 1f. It would make no sense to include data demonstrating *DMD* deletion transcripts in urine exRNA of DMD patients in the same figure with *DMPK* expression level in DM1 patients.

6. And why is DMD the only gene where every patient has a different aberrant splice pattern, whereas the other genes (in Fig1) seem to be conserved between patients. Discuss.

Response: Duchenne muscular dystrophy (DMD; no italics) patients have frame-shifting mutations in the *DMD* (italics) gene. Many different mutations in the *DMD* gene cause DMD. In Figure 4a and Figure 5d, we demonstrate that alternative splicing patterns in urine exRNA of DMD patients is similar to UA subjects. As we explained in Point 23 of the first Response, the *DMD* transcripts that we show in Figure 8 are mRNA copies of the genetic mutations in the DNA from each individual, effectively serving as personalized genetic markers in urine. None of these *DMD* transcripts have “an aberrant splice pattern;” instead, they are spliced correctly in the context of the underlying DNA deletion. The deletions examined in Figure 8 are as follows:

- Subject 1 (S1) has an exon 18 - 22 deletion in the *DMD* gene. Using urine exRNA, RT-PCR, and primers targeting exons 17 and 23, we identified a PCR product that contains the exon 18 - 22 deletion, and confirmed it by sequencing.
- S2 has an exon 51 - 53 deletion in the *DMD* gene. This deletion is amenable to a therapeutic strategy that involves skipping of exon 50. Using urine exRNA, RT-PCR, and primers targeting exons 49 and 54, we identified a PCR product that contains the exon 51 - 53 deletion, and confirmed it by sequencing.
- S3 has an exon 49 - 52 deletion in the *DMD* gene. This deletion is amenable to a therapeutic strategy that involves skipping of exon 53. Using urine exRNA, RT-PCR, and primers targeting exons 48 and 53, we identified a PCR product containing an exon 49 - 52 deletion, and confirmed it with sequencing.

- S4 and S5 have an exon 46 - 52 deletion in the *DMD* gene. This deletion is amenable to a therapeutic strategy that involves skipping of both exons 45 and 53 together. Using urine exRNA, RT-PCR, and primers targeting exons 45 and 53, we identified a PCR product containing an exon 46 - 52 deletion, and confirmed it by sequencing.
 - S6 has an exon 24 - 43 deletion in the *DMD* gene. Using urine exRNA, RT-PCR, and primers targeting exons 23 and 44, we identified a PCR product containing an exon 24 - 43 deletion, and confirmed it by sequencing.
 - S7 has an exon 45 - 50 deletion in the *DMD* gene. This deletion is amenable to a therapeutic strategy that involves skipping of exon 51, and this individual has been receiving treatment with the exon 51-skipping ASO eteplirsén. Using urine exRNA, urine cell total RNA, RT-PCR, and primers targeting exons 44 and 52, we identified a PCR product that contains an exon 45 - 51 deletion, indicating ASO activity to induce skipping of exon 51 that was confirmed by sequencing of the PCR product. A larger faint gel band that corresponds to an exon 45 - 50 deletion (exon 51 still present) is evident in some of the PCR reactions. Using ddPCR, we quantified the percent exon 51 inclusion in urine exRNA and urine cells.
7. Since DMD is a genomic disease (mutations in the *DMD* gene), but severity / phenotype depends on the degree of aberrant splicing of key mRNAs. The authors should briefly describe/mention the potential utility of a urine based test beyond the use as a therapy monitoring tool.

Response: The statement that DMD disease severity depends on the degree of aberrant splicing of key mRNAs is incorrect. As we show in Figure 4a and Figure 5d, alternative splicing patterns in DMD individuals are similar to UA individuals. You may be confusing myotonic dystrophy (DM1) with Duchenne muscular dystrophy (DMD; no italics). Both are muscular dystrophies, but they are caused by mutations of different genes and feature markedly different phenotypes. DMD results from loss-of-function mutations of the *DMD* (italics) gene. In DMD patients, symptoms result from absence of dystrophin protein, which leads to progressive muscle weakness and degeneration. In Figure 8, we demonstrate that *DMD* deletion transcripts

are routinely detectable in urine exRNA, and can serve as personalized genetic markers (mRNA copies of the DNA gene deletion for each individual). It seems possible that the expression of *DMD* deletion transcripts in the cells that produce urine exRNA may have downstream effects on other genes, and that these effects could be measured in urine exRNA.

In the BMD patient data that you asked us to remove, we demonstrate that *DMD* transcripts in urine exRNA also can be used to identify a novel cryptic splice site in an intron of an individual with dystrophinopathy but who has a normal *DMD* coding sequence, suggesting the capacity of urine exRNA to substitute for muscle biopsies as a means to determine disease mechanism of specific *DMD* mutations. It seems likely that the capability of urine exRNA to identify novel splice variants could be extended to other genes.

8. The manuscript now makes a convincing claim that the mRNA splice patterns in urine EVs, urine cells and serum EVs are different and that only the urine EV splice patterns correlate with the disease. The conclusion seems to be that the urine EVs to a significant extent come from cells that are affected by the disease, e.g. because expression of the relevant disease genes is high. This does not seem to be the case for the cells found in urine and also not for the majority of EVs found in serum. I think the authors have enough circumstantial evidence to speculate about which cells the urine EVs might come from, but it is never clearly called out. Which tissue is the main contributor to EVs in urine? The authors could be a little more clear about whether they think EVs from muscle cells (where the disease is having its main manifestation) is not present in serum or whether they are simply diluted out by other cells.

Response: We discussed this in Point 25 of the first response. In addition, the Discussion reads as follows:

“Because DM1 is primarily a disease of skeletal muscle, heart, and the central nervous system (CNS), and these tissues release EVs, it is counter-intuitive that exRNA reflecting the characteristic mis-regulated splicing events in DM1 would appear in urine rather than in blood, as exRNA has not been shown to pass from the blood through the proximal tubules of the kidney (Erdbrügger and Le, 2016). This suggests that the source of exRNA in these biofluids is likely to be different, and that the primary source in serum is unlikely to be muscle tissue. The fact that the presence of *DMPK* transcripts is an order of magnitude lower in serum than in urine

exRNA suggests that the cells contributing to the serum exRNA pool may be primarily those that are unaffected in DM1 patients due to low expression of *DMPK*, thereby explaining the similarity of serum splicing patterns in DM1 and UA subjects.”

“Here we find that splice patterns of several transcripts in urine exRNA are more similar to those in kidney tissue and urothelial cells as compared to muscle tissue, suggesting that the exRNA found in urine may represent a pool from multiple different cell types along this urinary route.”

“In this study, we also found an important difference of splice patterns in the urine exRNA pool, as compared to those in total RNA from urine cells, indicating that the source of these two RNA populations is distinct, but also suggesting that urine cells have the potential to serve as a second biomarker source complementary to urine exRNA.”

9. Where does the ASO go in the body? Is it administered IV? Then what? Where does the drug go in the body? Maybe that could suggest where it might have the biggest effect.

Response: We discussed the pharmacokinetic properties of ASOs in Point 25 of the first Response. The pharmacokinetic properties of ASOs are determined mostly by the chemistry of its backbone linkage, and are largely independent of sequence within a chemical class (Bennett, et al., 2017). An ASO with an uncharged linkage, including a morpholino ASO such as eteplirsén, exhibits rapid clearance from the blood via the kidney, resulting in minimal muscle tissue uptake (Bennett, et al., 2017). By contrast, ASOs with a phosphorothioate linkage demonstrate activity in normal muscle that is similar to activity in kidney of non-human primates (Pandey, et al., 2015), and in wild-type mouse muscle, bladder, and kidney (Supplementary Fig. 15). For additional information about the medicinal chemistry, pharmacokinetics, toxicology, and pharmacology of ASOs, please see the excellent review by Bennett et al.

Reference:

- Bennett, C.F., Baker, B.F., Pham, N., Swayze, E. & Geary, R.S. Pharmacology of Antisense Drugs. *Annu Rev Pharmacol Toxicol* **57**, 81-105 (2017).

10. All the PCR data used to separate UAs from DM1/DMD is on %Splice-variant, which works nice. However, it would be good to mention (discussion?) whether simple expression levels of the relevant genes had any diagnostic performance at all.

Response: As we mention in the Introduction section of the manuscript, pre-mRNA splicing outcomes in muscle biopsies are used as biomarkers of disease severity (Nakamori, et al., 2013), while in DM1 mice they also have served as sensitive indicators of therapeutic drug activity (Wheeler, et al., 2009; Wheeler, et al., 2012). In urine exRNA, analysis of splicing is straightforward and our splicing data are overwhelmingly robust, which combine to form an ideal measurement of disease activity. By measuring both the exon inclusion and exon exclusion splice products generated from the same gene, we eliminate the effect of overall gene expression level, which is an important source of biological variation.

However, application of urine exRNA to other disorders may involve determination of overall expression levels of individual genes, which would require measurement of the target transcript relative to a reference gene. Similar to determination of the exon inclusion percentage, this involves quantification of two transcripts, followed by normalization.

References:

- Nakamori, M. et al. Splicing biomarkers of disease severity in myotonic dystrophy. *Ann Neurol* **74**, 862-872 (2013).
- Wheeler, T.M. et al. Reversal of RNA dominance by displacement of protein sequestered on triplet repeat RNA. *Science* **325**, 336-339 (2009).
- Wheeler, T.M. et al. Targeting nuclear RNA for in vivo correction of myotonic dystrophy. *Nature* **488**, 111-115 (2012).

11. It is really too bad that there is nothing at all to compare the data for DMD mRNA in Figure 8b to. All of the gels in Figure 8a are supposed to be evidence of DMD aberrant splicing that might be helped by ASO therapy, but not therapy is given so there is no evidence of any change in urine exRNA as a result of therapy. Similarly, the data in Fig 8b is during treatment with ASO, but there is no evidence of what the splice pattern was before therapy, so again, no evidence of an effect. I acknowledge that this data is not easy to get to and that longitudinal monitoring of patients during ASO treatment is a

tall order. Unfortunately, the very impressive collection of longitudinal collections in Figure 3 does not show any sign of change (and it is not discussed whether patients are on ASO treatment during collection, but I suspect not). Again, I wonder why the DMD gene is not included in this gene set. Did none of these subjects have frame shifting deletions in DMD? What are the different lanes in the gels in Fig8b? Why are there bands in two of 5 lanes in the 2nd panel?

Response: None of the gels in Figure 8 show aberrant *DMD* splicing. Instead, they show PCR products that accurately represent an mRNA copy of the DNA gene deletion for each individual (see Points 5, 6, and 7 above). This is important because the presence of *DMD* deletion transcripts in urine RNA suggests the possibility to measure pharmacological activity of exon-skipping ASOs in urine, which we show in Figure 8b-e. In the absence of treatment, *DMD* deletion transcripts will remain unchanged throughout an individual's lifetime.

Figure 3 has no relationship to Figure 8. In Figure 3, we demonstrate longitudinal analysis of splicing outcomes of the ten transcripts that are mis-regulated in urine exRNA of DM1 individuals. As we show in Figure 4a and Figure 5d, splicing in DMD patients is similar to that in UA controls. In Figure 8, we show that *DMD* deletion transcripts are routinely detectable in the urine. The *DMD* deletion transcripts in DMD patients have no relationship to the mis-regulated alternative splicing patterns that are evident in urine exRNA of DM1 subjects (Fig. 4a, 5d).

It is important to note that the vast majority of the 79 *DMD* exons, including exon 51, are constitutively spliced (Bouge, et al., 2017). As we explained in Point 5 of the first response, we found no evidence in the scientific literature that *DMD* exon 51 is alternatively spliced. To determine whether exon 51 inclusion could be reduced by a previously unknown alternative splicing event that is unique to urine exRNA and urine cells, we examined exon 51 splicing in DM1 and UA urine samples by RT-PCR and found that exon 51 inclusion is > 99% (Supplementary Fig. 16). Sequencing of PCR products confirmed the presence of exon 51. In a previous study, ddPCR analysis of cultured DMD patient cells that have an exon 45 - 50 deletion, the same deletion as in S7, found that exon 51 inclusion was 99.3% (Verheul, et al., 2016). In cultured DMD patient cells with an exon 48 - 50 deletion, ddPCR analysis found that exon 51 inclusion was 99.9%, and in cultured cells with an exon 52 deletion, exon 51 inclusion was > 99.9% (Verheul, et al., 2016). ***DMD* exon 51 is constitutively spliced, and in the absence of ASO treatment, its inclusion is > 99% in muscle, urine exRNA, and urine cells.**

As we explained in Point 5 of the first Response, longitudinal analysis of two urine samples from S7 collected six months apart showed the same result. However, RNA recovery was slightly better in the second sample, enabling ddPCR quantification of exon 51 inclusion (Fig 8e) in the presence of ASO treatment.

In Figure 8b, the different lanes on the gels represent five separate PCR reactions, indicating that the results are reproducible. The faint upper bands correspond to the exon 51 inclusion PCR product. The black boxes labeled 44 and 52 adjacent to the gel images indicate the *DMD* exons targeted by the left and right PCR primers, respectively. The blue box labeled 51 represents *DMD* exon 51. The size of the PCR products with and without exon 51 is shown in Supplementary Table 7.

References:

- Bouge, A.L. et al. Targeted RNA-Seq profiling of splicing pattern in the DMD gene: exons are mostly constitutively spliced in human skeletal muscle. *Sci Rep* **7**, 39094 (2017).
- Verheul, R.C., van Deutekom, J.C. & Datson, N.A. Digital Droplet PCR for the Absolute Quantification of Exon Skipping Induced by Antisense Oligonucleotides in (Pre-)Clinical Development for Duchenne Muscular Dystrophy. *PLoS One* **11**, e0162467 (2016).

12. The observation that the urine cells from subject S7 shows almost the same %exon11 inclusion runs counter to the observation from previously in the manuscript that urine EVs and urine cells show different splice patterns. Maybe analysis of serum EVs from S7 would show the 44-51-52 variant more clearly, which would support the claim that urine EVs are the better source for measuring these aberrant transcripts, than serum EVs or urine cells. Without any kind of comparison I think it is a stretch to call it “evidence of ASO activity”... it is very likely the case, but...

Response: The statement, “The observation that urine cells from subject S7 shows almost the same % exon 11 inclusion runs counter to the observation from previously in the manuscript that urine EVs and urine cells show different splice patterns,” is incorrect. You appear to be confusing alternative splicing outcomes, which are biomarkers of disease activity in DM, with

eteplirsen-induced exon 51 skipping for treatment of DMD. In Figure 6, we demonstrate that alternative splicing in urine exRNA is different than in urine cells. As we explain above in Points 4 and 11, **DMD exon 51 is constitutively spliced** (Bouge, et al., 2017; Verheul et al, 2016; Supplementary Fig. 16). In the absence of ASO treatment, *DMD* exon 51 inclusion is > 99% in muscle, urine exRNA, and urine cells. The similarity of exon 51 skipping in urine exRNA and urine cells of S7 (Fig. 8e) indicates that pharmacological activity of the ASO is similar in the RNA from these two sources. Constitutive splicing of *DMD* exon 51 in urine exRNA and urine cells, with or without ASO treatment, has no relationship to alternative splicing of unrelated transcripts in these two RNA sources.

S7 declined blood draw, which in our experience is common in pediatric patients, and even in many adults. This further highlights two additional important advantages of urine biomarkers: collection of urine samples is painless and truly non-invasive.

According to the genetic testing results in the medical record, S7 has an exon 45 - 50 deletion in the *DMD* gene. This is a frame-shifting deletion that results in absence of dystrophin protein at the muscle cell membrane, leading to progressive muscle weakness and degeneration. An exon 45 - 50 deletion is amenable to a therapeutic strategy that involves skipping of exon 51, which would produce an in-frame exon 45 - 51 deletion and an internally truncated, but partially functional, dystrophin protein that is localized to the muscle cell membrane (Aartsma-Rus and van Ommen, 2010). This mutation meets inclusion criteria for treatment with eteplirsen. If urine exRNA, and/or urine cell RNA from an individual with an exon 45 - 50 deletion who is receiving a drug that induces skipping of exon 51 is confirmed to have an exon 51 inclusion percentage that is significantly less than the baseline 99+%, then it is reasonable to conclude that it is due to activity of the ASO. Indeed, we are unaware of a scientifically rational alternative explanation.

References:

- Aartsma-Rus, A. & van Ommen, G.J. Progress in therapeutic antisense applications for neuromuscular disorders. *Eur J Hum Genet* **18**, 146-153 (2010).
- Bouge, A.L. et al. Targeted RNA-Seq profiling of splicing pattern in the DMD gene: exons are mostly constitutively spliced in human skeletal muscle. *Sci Rep* **7**, 39094 (2017).

- Verheul, R.C., van Deutekom, J.C. & Datson, N.A. Digital Droplet PCR for the Absolute Quantification of Exon Skipping Induced by Antisense Oligonucleotides in (Pre-)Clinical Development for Duchenne Muscular Dystrophy. *PLoS One* **11**, e0162467 (2016).

Reviewer #2 (Remarks to the Author):

- The authors have addressed the reviewer points sufficiently. I have no additional points to raise.

Reviewer #3 (Remarks to the Author):

- The paper is very much improved and I am satisfied.

Reviewer #4 (Remarks to the Author):

- The authors have addressed my concerns.

Reviewers' comments:

Reviewer #1 (Remarks to the Author):

1. This version is a significant improvement over the previous and the entire manuscript appears much more cohesive and reads much better. □The addition of ddPCR improves confidence in the data throughout.
2. I appreciate everything the authors do to characterize the RNA preparations by OD and Bioanalyzer, but I think there is no question the RNA preparations are contaminated with phenol. This makes all the conclusions drawn based on these measurements in the entire first section of results “Characterization of exRNA...” rather questionable and I would suggest to omit these results. They are not required for the remainder of the paper since the PCR/amplification analysis appear to work well, despite the phenol contamination. Drawing conclusions on the quantity of RNA from the various samples or which technical method is best for measuring the RNA will only distract from the main message. The RNA quality can be discussed as a limitation, but does not appear to impair the PCR results.

Response: We think it is important to include these results precisely to alert readers that nanospectrophotometry is inaccurate for characterization and quantification of exRNA using our methods. This was confusing to us early in our studies, and we want to help others avoid the problems that we had interpreting the exRNA readings. We also demonstrate that capillary gel electrophoresis provides an accurate estimation of exRNA concentration that, in stark contrast to nanospectrophotometry, correlates well with ddPCR quantification of reference gene *GTF2B* expression (Supplementary Fig. 1i).

As you please, but IMO the main (and interesting) message of using urine microvesicle RNA to detect alternative splicing in patients with molecular dystrophy should not be distracted with technical musing over how best to measure RNA concentration when RNA quality is bad.

3. I appreciate the addition of an additional reference gene (GAPDH), which confirms the previous one (GTF2B), but still find it puzzling that absolute expression level of either of

the reference genes seem to be able to correctly separate the patients from the UAs since the reference genes are the ones that display differential expression between the groups. What was the amount of sample used in these PCRs? Were they all performed on the same volume of exRNA or cDNA? Or were they normalized for RNA amount (based on the flawed RNA concentration measurements with phenol contamination)?

Response: We addressed this issue in Point 12 of the first Response. exRNA concentration correlates well with *GTF2B* expression by ddPCR (Supplementary Fig. 1), indicating that the differential expression of reference genes *GTF2B* and *GAPDH* is due to higher exRNA concentration in DM1 urine. In addition to these two reference genes, the total expression level of *MBNL2*, *MBNL1*, *MAP3K4*, *CLASP1*, and *INSR* by ddPCR, based on copies per microliter of the exon inclusion splice product plus the copies per microliter of the exon exclusion splice product, is higher in DM1 individuals than UA controls (Supplementary Fig. 6e). In all, six transcripts that we examined by ddPCR, plus *GAPDH* by qPCR, for a total of seven transcripts, are expressed at higher levels in DM1 individuals than UA controls, consistent with a higher total exRNA content in DM1 urine. This is an interesting disease manifestation that was previously unknown.

I agree, but I do not think this is well described in the manuscript – “amount of RNA” is never(?) discussed as a biomarker, but that seems to be the case. Caution, intentionally provocative: If the RNA quality was good, then an alternative test for identification of DM1 individuals relative to UA would be simple OD measurement!?

DM1 is autosomal dominant: the mutant polyadenylated *DMPK-CUG^{exp}* transcripts (50% of the total) form intranuclear inclusions (Taneja et al., 1995), while the *DMPK* transcripts that arise from the normal non-expanded allele (50% of the total) are transported to the cytoplasm. In individuals without DM1, *DMPK* transcripts from both alleles contain non-expanded CUG repeats, none of these transcripts form nuclear inclusions, and all are transported to the cytoplasm. While the mutant *DMPK-CUG^{exp}* transcripts are trapped in nuclear inclusions, they are unavailable for release from cells as exRNA. This could result in up to 50% fewer *DMPK* transcripts released as exRNA in DM1 individuals as compared to non-DM1 controls, which would explain the lower expression of *DMPK* mRNA in urine exRNA as compared to UA subjects that we observed (Fig. 1c).

All of the ddPCR results are shown as per microliter of cDNA; all qPCR results were determined using one microliter of cDNA from each sample.

OK, that's a nice explanation... but that does not explain why there is "a higher total exRNA content in urine"

Reference:

- Taneja, K.L., McCurrach, M., Schalling, M., Housman, D. & Singer, R.H. Foci of trinucleotide repeat transcripts in nuclei of myotonic dystrophy cells and tissues. *J Cell Biol* **128**, 995-1002 (1995).
4. The authors examined 33 mRNA candidates and found 10 that were different between UA and DM1/DMD, but the DMD mRNA is not among them. Why?

Response: We found ten alternative splice events that were significantly different in urine exRNA of DM1 patients as compared to DMD or UA controls. As we show in Figure 4a and Figure 5d, alternative splicing in DMD patients is similar to UA controls. Alternative splicing of *DMD* exon 71 and exon 78 is reported to be abnormal in muscle biopsy tissue of DM1 patients (Nakamori, et al., 2013). The similar splicing pattern of these alternative *DMD* exons in DM1 and UA exRNA (Supplementary Fig. 3) suggests that the regulation of these exons is different in urine exRNA than in muscle tissue. Almost all of the remaining 79 *DMD* exons are constitutively spliced and, therefore, are expected to be identical in DM1, DMD, and UA groups (Bouge, et al., 2017).

Reference:

- Bouge, A.L. et al. Targeted RNA-Seq profiling of splicing pattern in the DMD gene: exons are mostly constitutively spliced in human skeletal muscle. *Sci Rep* **7**, 39094 (2017).
 - Nakamori, M. et al. Splicing biomarkers of disease severity in myotonic dystrophy. *Ann Neurol* **74**, 862-872 (2013).
5. Later in the manuscript a lot of focus is on splicing of DMD-mRNA and it seems strange that DMD-mRNA did not show up in the original screen. Why is the data in Fig 8 not

part of Fig 1? The authors should discuss this (or even better add the data for DMD-mRNA in Figure 1ff).

Response: It's unclear what you mean by "original screen" in the context of *DMD* mRNA. In Figure 8, we used RT-PCR to demonstrate that *DMD* deletion transcripts are routinely detectable in urine exRNA, and can serve as personalized genetic markers (mRNA copies of the DNA gene deletion for each individual) (see Point 6 below). **Personalized genetic markers of individual DMD patients have no relationship to the aberrant alternative splicing in exRNA of DM1 individuals,** other than urine exRNA can be used to identify both.

OK, I clearly did not understand this completely.

In Figure 1, we used ddPCR to determine the expression of *DMPK*, the mutated gene in DM1, and reference gene *GTF2B* in urine exRNA, urine cells, serum exRNA of DM1, bladder tissue total RNA, urothelial cell total RNA, kidney tissue total RNA, and muscle tissue total RNA. *DMPK* expression is unrelated to *DMD* gene deletions. There is no Figure 1f. It would make no sense to include data demonstrating *DMD* deletion transcripts in urine exRNA of DMD patients in the same figure with *DMPK* expression level in DM1 patients.

Apologies, I meant Figure 2 (where alternative splicing is first shown). "ff" was supposed to mean that figure and all following instances of discussing alternative splicing.

I still don't understand why % exon 51 inclusion in *DMD* is completely separate from % exon 7 inclusion in *MBNL1*! (NB: See next comment :-)

6. And why is DMD the only gene where every patient has a different aberrant splice pattern, whereas the other genes (in Fig1) seem to be conserved between patients. Discuss.

Response: Duchenne muscular dystrophy (DMD; no italics) patients have frame-shifting mutations in the *DMD* (italics) gene. Many different mutations in the *DMD* gene cause DMD. In Figure 4a and Figure 5d, we demonstrate that alternative splicing patterns in urine exRNA of DMD patients is similar to UA subjects. As we explained in Point 23 of the first Response, the *DMD* transcripts that we show in Figure 8 are mRNA copies of the genetic mutations in the DNA

from each individual, effectively serving as personalized genetic markers in urine. None of these *DMD* transcripts have “an aberrant splice pattern;” instead, they are spliced correctly in the context of the underlying DNA deletion. The deletions examined in Figure 8 are as follows:

Ahh... that worked! Got it. Now I understand the difference. Thanks! That may be obvious to every other reader of the manuscript, in which case no edits are required.

- Subject 1 (S1) has an exon 18 - 22 deletion in the *DMD* gene. Using urine exRNA, RT-PCR, and primers targeting exons 17 and 23, we identified a PCR product that contains the exon 18 - 22 deletion, and confirmed it by sequencing.
- S2 has an exon 51 - 53 deletion in the *DMD* gene. This deletion is amenable to a therapeutic strategy that involves skipping of exon 50. Using urine exRNA, RT-PCR, and primers targeting exons 49 and 54, we identified a PCR product that contains the exon 51 - 53 deletion, and confirmed it by sequencing.
- S3 has an exon 49 - 52 deletion in the *DMD* gene. This deletion is amenable to a therapeutic strategy that involves skipping of exon 53. Using urine exRNA, RT-PCR, and primers targeting exons 48 and 53, we identified a PCR product containing an exon 49 - 52 deletion, and confirmed it with sequencing.
- S4 and S5 have an exon 46 - 52 deletion in the *DMD* gene. This deletion is amenable to a therapeutic strategy that involves skipping of both exons 45 and 53 together. Using urine exRNA, RT-PCR, and primers targeting exons 45 and 53, we identified a PCR product containing an exon 46 - 52 deletion, and confirmed it by sequencing.
- S6 has an exon 24 - 43 deletion in the *DMD* gene. Using urine exRNA, RT-PCR, and primers targeting exons 23 and 44, we identified a PCR product containing an exon 24 - 43 deletion, and confirmed it by sequencing.
- S7 has an exon 45 - 50 deletion in the *DMD* gene. This deletion is amenable to a therapeutic strategy that involves skipping of exon 51, and this individual has been receiving treatment with the exon 51-skipping ASO eteplirsén. Using urine exRNA, urine cell total RNA, RT-PCR, and primers targeting exons 44 and 52, we identified a PCR

product that contains an exon 45 - 51 deletion, indicating ASO activity to induce skipping of exon 51 that was confirmed by sequencing of the PCR product. A larger faint gel band that corresponds to an exon 45 - 50 deletion (exon 51 still present) is evident in some of the PCR reactions. Using ddPCR, we quantified the percent exon 51 inclusion in urine exRNA and urine cells.

7. Since DMD is a genomic disease (mutations in the DMD gene), but severity / phenotype depends on the degree of aberrant splicing of key mRNAs. The authors should briefly describe/mention the potential utility of a urine based test beyond the use as a therapy monitoring tool. □

Response: The statement that DMD disease severity depends on the degree of aberrant splicing of key mRNAs is incorrect. As we show in Figure 4a and Figure 5d, alternative splicing patterns in DMD individuals are similar to UA individuals. You may be confusing myotonic dystrophy (DM1) with Duchenne muscular dystrophy (DMD; no italics). Both are muscular dystrophies, but they are caused by mutations of different genes and feature markedly different phenotypes. DMD results from loss-of-function mutations of the *DMD* (italics) gene. In DMD patients, symptoms result from absence of dystrophin protein, which leads to progressive muscle weakness and degeneration. In Figure 8, we demonstrate that *DMD* deletion transcripts are routinely detectable in urine exRNA, and can serve as personalized genetic markers (mRNA copies of the DNA gene deletion for each individual). It seems possible that the expression of *DMD* deletion transcripts in the cells that produce urine exRNA may have downstream effects on other genes, and that these effects could be measured in urine exRNA.

In the BMD patient data that you asked us to remove, we demonstrate that *DMD* transcripts in urine exRNA also can be used to identify a novel cryptic splice site in an intron of an individual with dystrophinopathy but who has a normal *DMD* coding sequence, suggesting the capacity of urine exRNA to substitute for muscle biopsies as a means to determine disease mechanism of specific *DMD* mutations. It seems likely that the capability of urine exRNA to identify novel splice variants could be extended to other genes.

8. The manuscript now makes a convincing claim that the mRNA splice patterns in urine EVs, urine cells and serum EVs are different and that only the urine EV splice patterns correlate with the disease. The conclusion seems to be that the urine EVs to a significant extent come from cells that are affected by the disease, e.g. because expression of the relevant disease genes is high. This does not seem to be the case for the cells found in urine and also not for the majority of EVs found in serum. I think the authors have enough circumstantial evidence to speculate about which cells the urine EVs might come from, but it is never clearly called out. Which tissue is the main contributor to EVs in urine? The authors could be a little more clear about whether they think EVs from muscle cells (where the disease is having its main manifestation) is not present in serum or whether they are simply diluted out by other cells. □

Response: We discussed this in Point 25 of the first response. In addition, the Discussion reads as follows:

“Because DM1 is primarily a disease of skeletal muscle, heart, and the central nervous system (CNS), and these tissues release EVs, it is counter-intuitive that exRNA reflecting the characteristic mis-regulated splicing events in DM1 would appear in urine rather than in blood, as exRNA has not been shown to pass from the blood through the proximal tubules of the kidney (Erdbrügger and Le, 2016). This suggests that the source of exRNA in these biofluids is likely to be different, and that the primary source in serum is unlikely to be muscle tissue. The fact that the presence of *DMPK* transcripts is an order of magnitude lower in serum than in urine exRNA suggests that the cells contributing to the serum exRNA pool may be primarily those that are unaffected in DM1 patients due to low expression of *DMPK*, thereby explaining the similarity of serum splicing patterns in DM1 and UA subjects.”

“Here we find that splice patterns of several transcripts in urine exRNA are more similar to those in kidney tissue and urothelial cells as compared to muscle tissue, suggesting that the exRNA found in urine may represent a pool from multiple different cell types along this urinary route.”

“In this study, we also found an important difference of splice patterns in the urine exRNA pool, as compared to those in total RNA from urine cells, indicating that the source of these two RNA populations is distinct, but also suggesting that urine cells have the potential to serve as a second biomarker source complementary to urine exRNA.”

9. Where does the ASO go in the body? Is it administered IV? Then what? Where does the drug go in the body? Maybe that could suggest where it might have the biggest effect.

Response: We discussed the pharmacokinetic properties of ASOs in Point 25 of the first Response. The pharmacokinetic properties of ASOs are determined mostly by the chemistry of its backbone linkage, and are largely independent of sequence within a chemical class (Bennett, et al., 2017). An ASO with an uncharged linkage, including a morpholino ASO such as eteplirsen, exhibits rapid clearance from the blood via the kidney, resulting in minimal muscle tissue uptake (Bennett, et al., 2017). By contrast, ASOs with a phosphorothioate linkage demonstrate activity in normal muscle that is similar to activity in kidney of non-human primates (Pandey, et al., 2015), and in wild-type mouse muscle, bladder, and kidney (Supplementary Fig. 15). For additional information about the medicinal chemistry, pharmacokinetics, toxicology, and pharmacology of ASOs, please see the excellent review by Bennett et al.

So, the ASO is quickly going to the kidney, where it likely affects the kidney cells to induce exon 51 exclusion much more rapidly/effectively than in the muscles where it is poorly taken up. The signal seen in the urine is that from the kidney cells and/or possibly the muscle cells along the urinary tract(!?) rather than the peripheral muscles that are the main therapeutic target. This would be important to consider in evaluating a urine-based test for therapy monitoring.

Reference:

- Bennett, C.F., Baker, B.F., Pham, N., Swayze, E. & Geary, R.S. Pharmacology of Antisense Drugs. *Annu Rev Pharmacol Toxicol* **57**, 81-105 (2017).

10. All the PCR data used to separate UAs from DM1/DMD is on %Splice-variant, which works nice. However, it would be good to mention (discussion?) whether simple expression levels of the relevant genes had any diagnostic performance at all.

Response: As we mention in the Introduction section of the manuscript, pre-mRNA splicing outcomes in muscle biopsies are used as biomarkers of disease severity (Nakamori, et al., 2013), while in DM1 mice they also have served as sensitive indicators of therapeutic drug activity (Wheeler, et al., 2009; Wheeler, et al., 2012). In urine exRNA, analysis of splicing is

straightforward and our splicing data are overwhelmingly robust, which combine to form an ideal measurement of disease activity. By measuring both the exon inclusion and exon exclusion splice products generated from the same gene, we eliminate the effect of overall gene expression level, which is an important source of biological variation.

However, application of urine exRNA to other disorders may involve determination of overall expression levels of individual genes, which would require measurement of the target transcript relative to a reference gene. Similar to determination of the exon inclusion percentage, this involves quantification of two transcripts, followed by normalization.

OK

References:

- Nakamori, M. et al. Splicing biomarkers of disease severity in myotonic dystrophy. *Ann Neurol* **74**, 862-872 (2013).
- Wheeler, T.M. et al. Reversal of RNA dominance by displacement of protein sequestered on triplet repeat RNA. *Science* **325**, 336-339 (2009).
- Wheeler, T.M. et al. Targeting nuclear RNA for in vivo correction of myotonic dystrophy. *Nature* **488**, 111-115 (2012).

11. It is really too bad that there is nothing at all to compare the data for DMD mRNA in Figure 8b to. All of the gels in Figure 8a are supposed to be evidence of DMD aberrant splicing that might be helped by ASO therapy, but not therapy is given so there is no evidence of any change in urine exRNA as a result of therapy. Similarly, the data in Fig 8b is during treatment with ASO, but there is no evidence of what the splice pattern was before therapy, so again, no evidence of an effect. I acknowledge that this data is not easy to get to and that longitudinal monitoring of patients during ASO treatment is a tall order. Unfortunately, the very impressive collection of longitudinal collections in Figure 3 does not show any sign of change (and it is not discussed whether patients are on ASO treatment during collection, but I suspect not). Again, I wonder why the DMD gene is not included in this gene set. Did none of these subjects have frame shifting deletions in DMD? □ What are the different lanes in the gels in Fig8b? Why are there bands in two of 5 lanes in the 2nd panel?

Response: None of the gels in Figure 8 show aberrant *DMD* splicing. Instead, they show PCR products that accurately represent an mRNA copy of the DNA gene deletion for each individual (see Points 5, 6, and 7 above). This is important because the presence of *DMD* deletion transcripts in urine RNA suggests the possibility to measure pharmacological activity of exon-skipping ASOs in urine, which we show in Figure 8b-e. In the absence of treatment, *DMD* deletion transcripts will remain unchanged throughout an individual's lifetime.

Figure 3 has no relationship to Figure 8. In Figure 3, we demonstrate longitudinal analysis of splicing outcomes of the ten transcripts that are mis-regulated in urine exRNA of DM1 individuals. As we show in Figure 4a and Figure 5d, splicing in DMD patients is similar to that in UA controls. In Figure 8, we show that *DMD* deletion transcripts are routinely detectable in the urine. The *DMD* deletion transcripts in DMD patients have no relationship to the mis-regulated alternative splicing patterns that are evident in urine exRNA of DM1 subjects (Fig. 4a, 5d).

It is important to note that the vast majority of the 79 *DMD* exons, including exon 51, are constitutively spliced (Bouge, et al., 2017). As we explained in Point 5 of the first response, we found no evidence in the scientific literature that *DMD* exon 51 is alternatively spliced. To determine whether exon 51 inclusion could be reduced by a previously unknown alternative splicing event that is unique to urine exRNA and urine cells, we examined exon 51 splicing in DM1 and UA urine samples by RT-PCR and found that exon 51 inclusion is > 99% (Supplementary Fig. 16). Sequencing of PCR products confirmed the presence of exon 51. In a previous study, ddPCR analysis of cultured DMD patient cells that have an exon 45 - 50 deletion, the same deletion as in S7, found that exon 51 inclusion was 99.3% (Verheul, et al., 2016). In cultured DMD patient cells with an exon 48 - 50 deletion, ddPCR analysis found that exon 51 inclusion was 99.9%, and in cultured cells with an exon 52 deletion, exon 51 inclusion was > 99.9% (Verheul, et al., 2016). ***DMD* exon 51 is constitutively spliced, and in the absence of ASO treatment, its inclusion is > 99% in muscle, urine exRNA, and urine cells.**

As we explained in Point 5 of the first Response, longitudinal analysis of two urine samples from S7 collected six months apart showed the same result. However, RNA recovery was slightly better in the second sample, enabling ddPCR quantification of exon 51 inclusion (Fig 8e) in the presence of ASO treatment.

In Figure 8b, the different lanes on the gels represent five separate PCR reactions, indicating that the results are reproducible. The faint upper bands correspond to the exon 51 inclusion PCR product. The black boxes labeled 44 and 52 adjacent to the gel images indicate the *DMD* exons targeted by the left and right PCR primers, respectively. The blue box labeled 51 represents *DMD* exon 51. The size of the PCR products with and without exon 51 is shown in Supplementary Table 7.

References:

- Bouge, A.L. et al. Targeted RNA-Seq profiling of splicing pattern in the *DMD* gene: exons are mostly constitutively spliced in human skeletal muscle. *Sci Rep* **7**, 39094 (2017).
- Verheul, R.C., van Deutekom, J.C. & Datson, N.A. Digital Droplet PCR for the Absolute Quantification of Exon Skipping Induced by Antisense Oligonucleotides in (Pre-)Clinical Development for Duchenne Muscular Dystrophy. *PLoS One* **11**, e0162467 (2016).

I clearly do not understand this completely, and it **appears to be important!** I simply sought to point out that having a signal (no exon 51) under one condition (ASO excreting its effect) is not quite as convincing as also having no signal in the absence of that condition (control)! As a reader I feel left with “trust me, that’s what the control condition would show”!

12. The observation that the urine cells from subject S7 shows almost the same %exon11 inclusion runs counter to the observation from previously in the manuscript that urine EVs and urine cells show different splice patterns. Maybe analysis of serum EVs from S7 would show the 44-51-52 variant more clearly, which would support the claim that urine EVs are the better source for measuring these aberrant transcripts, than serum EVs or urine cells. Without any kind of comparison I think it is a stretch to call it “evidence of ASO activity”... it is very likely the case, but...□□□□□

Response: The statement, “The observation that urine cells from subject S7 shows almost the same % exon 11 inclusion runs counter to the observation from previously in the manuscript that urine EVs and urine cells show different splice patterns,” is incorrect. **You appear to be confusing alternative splicing outcomes, which are biomarkers of disease activity in DM, with eteplirsen-induced exon 51 skipping for treatment of DMD.**

Yes, I definitely was! Thanks! It is still not obvious to me why one would not also measure the “biomarkers of disease” and the impact on these during ASO therapy!? When the exon 51 is excluded and the therapy is working is it not expected to have some downstream impact on the biomarkers of disease!?

In Figure 6, we demonstrate that alternative splicing in urine exRNA is different than in urine cells. As we explain above in Points 4 and 11, **DMD exon 51 is constitutively spliced** (Bouge, et al., 2017; Verheul et al, 2016; Supplementary Fig. 16). In the absence of ASO treatment, *DMD* exon 51 inclusion is > 99% in muscle, urine exRNA, and urine cells. The similarity of exon 51 skipping in urine exRNA and urine cells of S7 (Fig. 8e) indicates that pharmacological activity of the ASO is similar in the RNA from these two sources. Constitutive splicing of *DMD* exon 51 in urine exRNA and urine cells, with or without ASO treatment, has no relationship to alternative splicing of unrelated transcripts in these two RNA sources.

S7 declined blood draw, which in our experience is common in pediatric patients, and even in many adults. This further highlights two additional important advantages of urine biomarkers: collection of urine samples is painless and truly non-invasive.

According to the genetic testing results in the medical record, S7 has an exon 45 - 50 deletion in the *DMD* gene. This is a frame-shifting deletion that results in absence of dystrophin protein at the muscle cell membrane, leading to progressive muscle weakness and degeneration. An exon 45 - 50 deletion is amenable to a therapeutic strategy that involves skipping of exon 51, which would produce an in-frame exon 45 - 51 deletion and an internally truncated, but partially functional, dystrophin protein that is localized to the muscle cell membrane (Aartsma-Rus and van Ommen, 2010). This mutation meets inclusion criteria for treatment with eteplirsen. If urine exRNA, and/or urine cell RNA from an individual with an exon 45 - 50 deletion who is receiving a drug that induces skipping of exon 51 is confirmed to have an exon 51 inclusion percentage that is significantly less than the baseline 99+%, then it is reasonable to conclude that it is due to activity of the ASO. Indeed, we are unaware of a scientifically rational alternative explanation.

Mmm... controls are run all the time to prove the obvious... e.g. assay specificity. A few patients that are NOT under treatment with ASO and show substantial inclusion of exon 51 would have been comforting.

References:

- Aartsma-Rus, A. & van Ommen, G.J. Progress in therapeutic antisense applications for neuromuscular disorders. *Eur J Hum Genet* **18**, 146-153 (2010).
- Bouge, A.L. et al. Targeted RNA-Seq profiling of splicing pattern in the DMD gene: exons are mostly constitutively spliced in human skeletal muscle. *Sci Rep* **7**, 39094 (2017).
- Verheul, R.C., van Deutekom, J.C. & Datson, N.A. Digital Droplet PCR for the Absolute Quantification of Exon Skipping Induced by Antisense Oligonucleotides in (Pre-)Clinical Development for Duchenne Muscular Dystrophy. *PLoS One* **11**, e0162467 (2016).

Reviewer #2 (Remarks to the Author):

- The authors have addressed the reviewer points sufficiently. I have no additional points to raise.

Reviewer #3 (Remarks to the Author):

- The paper is very much improved and I am satisfied.

Reviewer #4 (Remarks to the Author):

- The authors have addressed my concerns.

Reviewers' comments:

Reviewer #1 (Remarks to the Author):

1. This version is a significant improvement over the previous and the entire manuscript appears much more cohesive and reads much better. _The addition of ddPCR improves confidence in the data throughout.

2. I appreciate everything the authors do to characterize the RNA preparations by OD and Bioanalyzer, but I think there is no question the RNA preparations are contaminated with phenol. This makes all the conclusions drawn based on these measurements in the entire first section of results “Characterization of exRNA...” rather questionable and I would suggest to omit these results. They are not required for the remainder of the paper since the PCR/amplification analysis appear to work well, despite the phenol contamination. Drawing conclusions on the quantity of RNA from the various samples or which technical method is best for measuring the RNA will only distract from the main message. The RNA quality can be discussed as a limitation, but does not appear to impair the PCR results.

Response: We think it is important to include these results precisely to alert readers that nanospectrophotometry is inaccurate for characterization and quantification of exRNA using our methods. This was confusing to us early in our studies, and we want to help others avoid the problems that we had interpreting the exRNA readings. We also demonstrate that capillary gel electrophoresis provides an accurate estimation of exRNA concentration that, in stark contrast to nanospectrophotometry, correlates well with ddPCR quantification of reference gene *GTF2B* expression (Supplementary Fig. 1i).

- **As you please, but IMO the main (and interesting) message of using urine microvesicle RNA to detect alternative splicing in patients with molecular dystrophy should not be distracted with technical musing over how best to measure RNA concentration when RNA quality is bad.**

Response: We agree, which is why the only “technical musing” over whether measurement of RNA concentration is more accurate by capillary gel electrophoresis than microvolume spectrophotometry has been in response to your comments.

The differences in alternative splicing between the DM1 and non-DM groups by both RT-PCR and ddPCR in our study are as robust as the differences between these groups in muscle tissue samples by RT-PCR (Supplementary Fig. 5), which argues strongly against the exRNA quality in our study as “bad.”

Also, we are unsure what you mean by “molecular dystrophy.” You used this term in Point 6 of the first set of comments, appearing to reference muscular dystrophy in a general sense. Here, the context suggests that you may be referring to myotonic dystrophy.

3. I appreciate the addition of an additional reference gene (*GAPDH*), which confirms the previous one (*GTF2B*), but still find it puzzling that absolute expression level of either of the reference genes seem to be able to correctly separate the patients from the UAs since the reference genes are the ones that display differential expression between the groups. What was the amount of sample used in these PCRs? Were they all performed on the same volume of exRNA or cDNA? Or were they normalized for RNA amount (based on the flawed RNA concentration measurements with phenol contamination)?

Response: We addressed this issue in Point 12 of the first Response. exRNA concentration correlates well with *GTF2B* expression by ddPCR (Supplementary Fig. 1), indicating that the differential expression of reference genes *GTF2B* and *GAPDH* is due to higher exRNA concentration in DM1 urine. In addition to these two reference genes, the total expression level of *MBNL2*, *MBNL1*, *MAP3K4*, *CLASP1*, and *INSR* by ddPCR, based on copies per microliter of the exon inclusion splice product plus the copies per microliter of the exon exclusion splice product, is higher in DM1 individuals than UA controls (Supplementary Fig. 6e). In all, six transcripts that we examined by ddPCR, plus *GAPDH* by qPCR, for a total of seven transcripts, are expressed at higher levels in DM1 individuals than UA controls, consistent with a higher total exRNA content in DM1 urine. This is an interesting disease manifestation that was previously unknown.

- I agree, but I do not think this is well described in the manuscript – “amount of RNA” is never(?) discussed as a biomarker, but that seems to be the case. Caution, intentionally provocative: If the RNA quality was good, then an alternative test for identification of

DM1 individuals relative to UA would be simple OD measurement!?

Response: In this study, we are reporting splice variant biomarkers of muscular dystrophies in human urine. We have made no recommendation for the use RNA amount as a biomarker of muscular dystrophies, although, as we stated in Point 3 of the second response, the apparently higher quantity of splice products that we have noted in urine exRNA of DM1 individuals is an interesting disease manifestation.

In Supplementary Fig. 1, we show that the optical density measurements are inaccurate due to an artifact from phenol contamination that has been described previously (Krebs, et al., 2009). You seem to be misinterpreting an optical density measurement artifact as evidence of “bad” RNA. In fact, Krebs, et al., show very nicely that removal of residual phenol from minute RNA samples corrects the optical density artifact.

Reference:

- Krebs, S., Fischaleck, M. & Blum, H. A simple and loss-free method to remove TRIzol contaminations from minute RNA samples. *Anal Biochem* **387**, 136-138 (2009).

DM1 is autosomal dominant: the mutant polyadenylated *DMPK-CUGexp* transcripts (50% of the total) form intranuclear inclusions (Taneja et al., 1995), while the *DMPK* transcripts that arise from the normal non-expanded allele (50% of the total) are transported to the cytoplasm. In individuals without DM1, *DMPK* transcripts from both alleles contain non-expanded CUG repeats, none of these transcripts form nuclear inclusions, and all are transported to the cytoplasm. While the mutant *DMPK-CUGexp* transcripts are trapped in nuclear inclusions, they are unavailable for release from cells as exRNA. This could result in up to 50% fewer *DMPK* transcripts released as exRNA in DM1 individuals as compared to non-DM1 controls, which would explain the lower expression of *DMPK* mRNA in urine exRNA as compared to UA subjects that we observed (Fig. 1c).

All of the ddPCR results are shown as per microliter of cDNA; all qPCR results were determined using one microliter of cDNA from each sample.

- **OK, that's a nice explanation... but that does not explain why there is “a higher total**

exRNA content in urine”

Response: We agree. We have no explanation for why DM1 patients appear, as a group, to excrete more exRNA in urine than individuals without DM1, although this is an interesting question that deserves further study.

Reference:

- Taneja, K.L., McCurrach, M., Schalling, M., Housman, D. & Singer, R.H. Foci of trinucleotide repeat transcripts in nuclei of myotonic dystrophy cells and tissues. *J Cell Biol* **128**, 995-1002 (1995).

4. The authors examined 33 mRNA candidates and found 10 that were different between UA and DM1/DMD, but the DMD mRNA is not among them. Why?

Response: We found ten alternative splice events that were significantly different in urine exRNA of DM1 patients as compared to DMD or UA controls. As we show in Figure 4a and Figure 5d, alternative splicing in DMD patients is similar to UA controls. Alternative splicing of *DMD* exon 71 and exon 78 is reported to be abnormal in muscle biopsy tissue of DM1 patients (Nakamori, et al., 2013). The similar splicing pattern of these alternative *DMD* exons in DM1 and UA exRNA (Supplementary Fig. 3) suggests that the regulation of these exons is different in urine exRNA than in muscle tissue. Almost all of the remaining 79 *DMD* exons are constitutively spliced and, therefore, are expected to be identical in DM1, DMD, and UA groups (Bouge, et al., 2017).

References:

- Bouge, A.L. et al. Targeted RNA-Seq profiling of splicing pattern in the DMD gene: exons are mostly constitutively spliced in human skeletal muscle. *Sci Rep* **7**, 39094 (2017).
- Nakamori, M. et al. Splicing biomarkers of disease severity in myotonic dystrophy. *Ann Neurol* **74**, 862-872 (2013).

5. Later in the manuscript a lot of focus is on splicing of DMD-mRNA and it seems

strange that *DMD*-mRNA did not show up in the original screen. Why is the data in Fig 8 not part of Fig 1? The authors should discuss this (or even better add the data for *DMD*mRNA in Figure 1ff).

Response: It's unclear what you mean by "original screen" in the context of *DMD* mRNA. In Figure 8, we used RT-PCR to demonstrate that *DMD* deletion transcripts are routinely detectable in urine exRNA, and can serve as personalized genetic markers (mRNA copies of the DNA gene deletion for each individual) (see Point 6 below). **Personalized genetic markers of individual *DMD* patients have no relationship to the aberrant alternative splicing in exRNA of *DM1* individuals**, other than urine exRNA can be used to identify both.

- **OK, I clearly did not understand this completely.**

In Figure 1, we used ddPCR to determine the expression of *DMPK*, the mutated gene in *DM1*, and reference gene *GTF2B* in urine exRNA, urine cells, serum exRNA of *DM1*, bladder tissue total RNA, urothelial cell total RNA, kidney tissue total RNA, and muscle tissue total RNA. *DMPK* expression is unrelated to *DMD* gene deletions. There is no Figure 1f. It would make no sense to include data demonstrating *DMD* deletion transcripts in urine exRNA of *DMD* patients in the same figure with *DMPK* expression level in *DM1* patients.

- **Apologies, I meant Figure 2 (where alternative splicing is first shown). "ff" was supposed to mean that figure and all following instances of discussing alternative splicing. I still don't understand why % exon 51 inclusion in *DMD* is completely separate from % exon 7 inclusion in *MBNL1*!**

And why is *DMD* the only gene where every patient has a different aberrant splice pattern, whereas the other genes (in Fig1) seem to be conserved between patients. Discuss.

Response: Duchenne muscular dystrophy (*DMD*; no italics) patients have frame-shifting mutations in the *DMD* (italics) gene. Many different mutations in the *DMD* gene cause *DMD*. In Figure 4a and Figure 5d, we demonstrate that alternative splicing patterns in

urine exRNA of DMD patients is similar to UA subjects. As we explained in Point 23 of the first Response, the *DMD* transcripts that we show in Figure 8 are mRNA copies of the genetic mutations in the DNA from each individual, effectively serving as personalized genetic markers in urine. **None of these *DMD* transcripts have "an aberrant splice pattern;" instead, they are spliced correctly in the context of the underlying DNA deletion.** The deletions examined in Figure 8 are as follows:

Ahh... that worked! Got it. Now I understand the difference. Thanks! That may be obvious to every other reader of the manuscript, in which case no edits are required.

- Subject 1 (S1) has an exon 18 - 22 deletion in the *DMD* gene. Using urine exRNA, RTPCR, and primers targeting exons 17 and 23, we identified a PCR product that contains the exon 18 - 22 deletion, and confirmed it by sequencing.
- S2 has an exon 51 - 53 deletion in the *DMD* gene. This deletion is amenable to a therapeutic strategy that involves skipping of exon 50. Using urine exRNA, RT-PCR, and primers targeting exons 49 and 54, we identified a PCR product that contains the exon 51 - 53 deletion, and confirmed it by sequencing.
- S3 has an exon 49 - 52 deletion in the *DMD* gene. This deletion is amenable to a therapeutic strategy that involves skipping of exon 53. Using urine exRNA, RT-PCR, and primers targeting exons 48 and 53, we identified a PCR product containing an exon 49 - 52 deletion, and confirmed it with sequencing.
- S4 and S5 have an exon 46 - 52 deletion in the *DMD* gene. This deletion is amenable to a therapeutic strategy that involves skipping of both exons 45 and 53 together. Using urine exRNA, RT-PCR, and primers targeting exons 45 and 53, we identified a PCR product containing an exon 46 - 52 deletion, and confirmed it by sequencing.
- S6 has an exon 24 - 43 deletion in the *DMD* gene. Using urine exRNA, RT-PCR, and primers targeting exons 23 and 44, we identified a PCR product containing an exon 24 - 43 deletion, and confirmed it by sequencing.
- S7 has an exon 45 - 50 deletion in the *DMD* gene. This deletion is amenable to a therapeutic strategy that involves skipping of exon 51, and this individual has been receiving treatment with the exon 51-skipping ASO eteplirsen. Using urine exRNA, urine cell total RNA, RT-PCR, and primers targeting exons 44 and 52, we identified a PCR product that contains an exon 45 - 51 deletion, indicating ASO activity to induce skipping of exon 51 that was confirmed by sequencing of the PCR product. A larger faint gel band

that corresponds to an exon 45 - 50 deletion (exon 51 still present) is evident in some of the PCR reactions. Using ddPCR, we quantified the percent exon 51 inclusion in urine exRNA and urine cells.

7. Since DMD is a genomic disease (mutations in the DMD gene), but severity / phenotype depends on the degree of aberrant splicing of key mRNAs. The authors should briefly describe/mention the potential utility of a urine based test beyond the use as a therapy monitoring tool.

Response: The statement that DMD disease severity depends on the degree of aberrant splicing of key mRNAs is incorrect. As we show in Figure 4a and Figure 5d, alternative splicing patterns in DMD individuals are similar to UA individuals. You may be confusing myotonic dystrophy (DM1) with Duchenne muscular dystrophy (DMD; no italics). Both are muscular dystrophies, but they are caused by mutations of different genes and feature markedly different phenotypes. DMD results from loss-of-function mutations of the *DMD* (italics) gene. In DMD patients, symptoms result from absence of dystrophin protein, which leads to progressive muscle weakness and degeneration. In Figure 8, we demonstrate that *DMD* deletion transcripts are routinely detectable in urine exRNA, and can serve as personalized genetic markers (mRNA copies of the DNA gene deletion for each individual). It seems possible that the expression of *DMD* deletion transcripts in the cells that produce urine exRNA may have downstream effects on other genes, and that these effects could be measured in urine exRNA.

In the BMD patient data that you asked us to remove, we demonstrate that *DMD* transcripts in urine exRNA also can be used to identify a novel cryptic splice site in an intron of an individual with dystrophinopathy but who has a normal *DMD* coding sequence, suggesting the capacity of urine exRNA to substitute for muscle biopsies as a means to determine disease mechanism of specific *DMD* mutations. It seems likely that the capability of urine exRNA to identify novel splice variants could be extended to other genes.

8. The manuscript now makes a convincing claim that the mRNA splice patterns in urine

EVs, urine cells and serum EVs are different and that only the urine EV splice patterns correlate with the disease. The conclusion seems to be that the urine EVs to a significant extent come from cells that are affected by the disease, e.g. because expression of the relevant disease genes is high. This does not seem to be the case for the cells found in urine and also not for the majority of EVs found in serum. I think the authors have enough circumstantial evidence to speculate about which cells the urine EVs might come from, but it is never clearly called out. Which tissue is the main contributor to EVs in urine? The authors could be a little more clear about whether they think EVs from muscle cells (where the disease is having its main manifestation) is not present in serum or whether they are simply diluted out by other cells.

Response: We discussed this in Point 25 of the first response. In addition, the Discussion reads as follows:

“Because DM1 is primarily a disease of skeletal muscle, heart, and the central nervous system (CNS), and these tissues release EVs, it is counter-intuitive that exRNA reflecting the characteristic mis-regulated splicing events in DM1 would appear in urine rather than in blood, as exRNA has not been shown to pass from the blood through the proximal tubules of the kidney (Erdburger and Le, 2016). This suggests that the source of exRNA in these biofluids is likely to be different, and that the primary source in serum is unlikely to be muscle tissue. The fact that the presence of *DMPK* transcripts is an order of magnitude lower in serum than in urine exRNA suggests that the cells contributing to the serum exRNA pool may be primarily those that are unaffected in DM1 patients due to low expression of *DMPK*, thereby explaining the similarity of serum splicing patterns in DM1 and UA subjects.”

“Here we find that splice patterns of several transcripts in urine exRNA are more similar to those in kidney tissue and urothelial cells as compared to muscle tissue, suggesting that the exRNA found in urine may represent a pool from multiple different cell types along this urinary route.”

“In this study, we also found an important difference of splice patterns in the urine exRNA pool, as compared to those in total RNA from urine cells, indicating that the source of these two RNA populations is distinct, but also suggesting that urine cells have the potential to serve as a second biomarker source complementary to urine exRNA.”

9. Where does the ASO go in the body? Is it administered IV? Then what? Where does the drug go in the body? Maybe that could suggest where it might have the biggest effect.

Response: We discussed the pharmacokinetic properties of ASOs in Point 25 of the first Response. The pharmacokinetic properties of ASOs are determined mostly by the chemistry of its backbone linkage, and are largely independent of sequence within a chemical class (Bennett, et al., 2017). An ASO with an uncharged linkage, including a morpholino ASO such as eteplirsen, exhibits rapid clearance from the blood via the kidney, resulting in minimal muscle tissue uptake (Bennett, et al., 2017). By contrast, ASOs with a phosphorothioate linkage demonstrate activity in normal muscle that is similar to activity in kidney of non-human primates (Pandey, et al., 2015), and in wild-type mouse muscle, bladder, and kidney (Supplementary Fig. 15). For additional information about the medicinal chemistry, pharmacokinetics, toxicology, and pharmacology of ASOs, please see the excellent review by Bennett et al.

- **So, the ASO is quickly going to the kidney, where it likely affects the kidney cells to induce exon 51 exclusion much more rapidly/effectively than in the muscles where it is poorly taken up. The signal seen in the urine is that from the kidney cells and/or possibly the muscle cells along the urinary tract(!?) rather than the peripheral muscles that are the main therapeutic target. This would be important to consider in evaluating a urine-based test for therapy monitoring.**

Response: We agree. Please see Point 25 and Point 3 (Reviewer 2) of the first response.

Reference:

- Bennett, C.F., Baker, B.F., Pham, N., Swayze, E. & Geary, R.S. Pharmacology of Antisense Drugs. *Annu Rev Pharmacol Toxicol* **57**, 81-105 (2017).

10. All the PCR data used to separate UAs from DM1/DMD is on %Splice-variant, which works nice. However, it would be good to mention (discussion?) whether simple expression levels of the relevant genes had any diagnostic performance at all.

Response: As we mention in the Introduction section of the manuscript, pre-mRNA splicing outcomes in muscle biopsies are used as biomarkers of disease severity (Nakamori, et al., 2013), while in DM1 mice they also have served as sensitive indicators of therapeutic drug activity (Wheeler, et al., 2009; Wheeler, et al., 2012). In urine exRNA, analysis of splicing is straightforward and our splicing data are overwhelmingly robust, which combine to form an ideal measurement of disease activity. By measuring both the exon inclusion and exon exclusion splice products generated from the same gene, we eliminate the effect of overall gene expression level, which is an important source of biological variation. However, application of urine exRNA to other disorders may involve determination of overall expression levels of individual genes, which would require measurement of the target transcript relative to a reference gene. Similar to determination of the exon inclusion percentage, this involves quantification of two transcripts, followed by normalization.

- **OK**

References:

- Nakamori, M. et al. Splicing biomarkers of disease severity in myotonic dystrophy. *Ann Neurol* **74**, 862-872 (2013).
- Wheeler, T.M. et al. Reversal of RNA dominance by displacement of protein sequestered on triplet repeat RNA. *Science* **325**, 336-339 (2009).
- Wheeler, T.M. et al. Targeting nuclear RNA for in vivo correction of myotonic dystrophy. *Nature* **488**, 111-115 (2012).

11. It is really too bad that there is nothing at all to compare the data for DMD mRNA in Figure 8b to. All of the gels in Figure 8a are supposed to be evidence of DMD aberrant splicing that might be helped by ASO therapy, but not therapy is given so there is no evidence of any change in urine exRNA as a result of therapy. Similarly, the data in Fig 8b is during treatment with ASO, but there is no evidence of what the splice pattern was before therapy, so again, no evidence of an effect. I acknowledge that this data is not easy to get to and that longitudinal monitoring of patients during ASO treatment is a tall order. Unfortunately, the very impressive collection of longitudinal collections in

Figure 3 does not show any sign of change (and it is not discussed whether patients are on ASO treatment during collection, but I suspect not). Again, I wonder why the *DMD* gene is not included in this gene set. Did none of these subjects have frame shifting deletions in *DMD*?_What are the different lanes in the gels in Fig8b? Why are there bands in two of 5 lanes in the 2nd panel?

Response: None of the gels in Figure 8 show aberrant *DMD* splicing. Instead, they show PCR products that accurately represent an mRNA copy of the DNA gene deletion for each individual (see Points 5, 6, and 7 above). This is important because the presence of *DMD* deletion transcripts in urine RNA suggests the possibility to measure pharmacological activity of exon skipping ASOs in urine, which we show in Figure 8b-e. In the absence of treatment, *DMD* deletion transcripts will remain unchanged throughout an individual's lifetime.

Figure 3 has no relationship to Figure 8. In Figure 3, we demonstrate longitudinal analysis of splicing outcomes of the ten transcripts that are mis-regulated in urine exRNA of DM1 individuals. As we show in Figure 4a and Figure 5d, splicing in *DMD* patients is similar to that in UA controls. In Figure 8, we show that *DMD* deletion transcripts are routinely detectable in the urine. The *DMD* deletion transcripts in *DMD* patients have no relationship to the mis-regulated alternative splicing patterns that are evident in urine exRNA of DM1 subjects (Fig. 4a, 5d).

It is important to note that the vast majority of the 79 *DMD* exons, including exon 51, are constitutively spliced (Bouge, et al., 2017). As we explained in Point 5 of the first response, we found no evidence in the scientific literature that *DMD* exon 51 is alternatively spliced. To determine whether exon 51 inclusion could be reduced by a previously unknown alternative splicing event that is unique to urine exRNA and urine cells, we examined exon 51 splicing in DM1 and UA urine samples by RT-PCR and found that exon 51 inclusion is > 99% (Supplementary Fig. 16). Sequencing of PCR products confirmed the presence of exon 51. In a previous study, ddPCR analysis of cultured *DMD* patient cells that have an exon 45 - 50 deletion, the same deletion as in S7, found that exon 51 inclusion was 99.3% (Verheul, et al., 2016). In cultured *DMD* patient cells with an exon 48 - 50 deletion, ddPCR analysis found that exon 51 inclusion was 99.9%, and in cultured cells with an exon 52 deletion, exon 51 inclusion was >

99.9% (Verheul, et al., 2016). **DMD exon 51 is constitutively spliced, and in the absence of ASO treatment, its inclusion is > 99% in muscle, urine exRNA, and urine cells.**

As we explained in Point 5 of the first Response, longitudinal analysis of two urine samples from S7 collected six months apart showed the same result. However, RNA recovery was slightly better in the second sample, enabling ddPCR quantification of exon 51 inclusion (Fig 8e) in the presence of ASO treatment.

In Figure 8b, the different lanes on the gels represent five separate PCR reactions, indicating that the results are reproducible. The faint upper bands correspond to the exon 51 inclusion PCR product. The black boxes labeled 44 and 52 adjacent to the gel images indicate the *DMD* exons targeted by the left and right PCR primers, respectively. The blue box labeled 51 represents *DMD* exon 51. The size of the PCR products with and without exon 51 is shown in Supplementary Table 7.

References:

- Bouge, A.L. et al. Targeted RNA-Seq profiling of splicing pattern in the DMD gene: exons are mostly constitutively spliced in human skeletal muscle. *Sci Rep* **7**, 39094 (2017).
- Verheul, R.C., van Deutekom, J.C. & Datson, N.A. Digital Droplet PCR for the Absolute Quantification of Exon Skipping Induced by Antisense Oligonucleotides in (Pre-)Clinical Development for Duchenne Muscular Dystrophy. *PLoS One* **11**, e0162467 (2016).
- **I clearly do not understand this completely, and it appears to be important! I simply sought to point out that having a signal (no exon 51) under one condition (ASO excreting its effect) is not quite as convincing as also having no signal in the absence of that condition (control)! As a reader I feel left with “trust me, that’s what the control condition would show”!**

Response: Of course we agree that it would be ideal to have a pre-treatment sample. However, as we explained in Point 1 for Reviewer 2 in the first Response and in the manuscript text, the individual S7 had been on the ASO for nearly three years at the time we collected the sample. In Fig. 10, we have added data from a second individual, S8, who also has been the

ASO for more than three years. Obviously, there is no way to go back in time to collect a pre-treatment sample from these individuals, and we have no access to pre-treatment samples from individuals who are being enrolled in ongoing clinical trials.

Research on the dystrophin gene as the cause of Duchenne muscular dystrophy has been going on for more than thirty years (Hoffman, et al., 1987). During this time, studies on splicing of the dystrophin gene, including recent quantitative studies by RNA sequencing (Bouge, et al., 2017) and ddPCR (Verheul, et al., 2016) have determined that exon 51 is constitutively spliced (inclusion > 99%). A thirty-year body of research in this area is fairly compelling.

References:

- Bouge, A.L. et al. Targeted RNA-Seq profiling of splicing pattern in the DMD gene: exons are mostly constitutively spliced in human skeletal muscle. *Sci Rep* **7**, 39094 (2017).
- Hoffman, E.P., Brown, R.H., Jr. & Kunkel, L.M. Dystrophin: the protein product of the Duchenne muscular dystrophy locus. *Cell* **51**, 919-928 (1987).
- Verheul, R.C., van Deutekom, J.C. & Datson, N.A. Digital Droplet PCR for the Absolute Quantification of Exon Skipping Induced by Antisense Oligonucleotides in (Pre-)Clinical Development for Duchenne Muscular Dystrophy. *PLoS One* **11**, e0162467 (2016).

12. The observation that the urine cells from subject S7 shows almost the same %exon11 inclusion runs counter to the observation from previously in the manuscript that urine EVs and urine cells show different splice patterns. Maybe analysis of serum EVs from S7 would show the 44-51-52 variant more clearly, which would support the claim that urine EVs are the better source for measuring these aberrant transcripts, than serum EVs or urine cells. Without any kind of comparison I think it is a stretch to call it “evidence of ASO activity”... it is very likely the case, but...

Response: The statement, “The observation that urine cells from subject S7 shows almost the same % exon 11 inclusion runs counter to the observation from previously in the manuscript that urine EVs and urine cells show different splice patterns,” is incorrect. You appear to be confusing alternative splicing outcomes, which are biomarkers of disease activity in DM, with eteplirsen-induced exon 51 skipping for treatment of DMD.

- Yes, I definitely was! Thanks! It is still not obvious to me why one would not also measure the “biomarkers of disease” and the impact on these during ASO therapy!?! When the exon 51 is excluded and the therapy is working is it not expected to have some downstream impact on the biomarkers of disease!?!?

Response: Yes, we believe it is possible that restoration of the *DMD* reading frame may induce a downstream impact that could be measured in urine, and are investigating this possibility.

In Figure 6, we demonstrate that alternative splicing in urine exRNA is different than in urine cells. As we explain above in Points 4 and 11, ***DMD* exon 51 is constitutively spliced** (Bouge, et al., 2017; Verheul et al, 2016; Supplementary Fig. 16). In the absence of ASO treatment, *DMD* exon 51 inclusion is > 99% in muscle, urine exRNA, and urine cells. The similarity of exon 51 skipping in urine exRNA and urine cells of S7 (Fig. 8e) indicates that pharmacological activity of the ASO is similar in the RNA from these two sources. Constitutive splicing of *DMD* exon 51 in urine exRNA and urine cells, with or without ASO treatment, has no relationship to alternative splicing of unrelated transcripts in these two RNA sources.

S7 declined blood draw, which in our experience is common in pediatric patients, and even in many adults. This further highlights two additional important advantages of urine biomarkers: collection of urine samples is painless and truly non-invasive.

According the genetic testing results in the medical record, S7 has an exon 45 - 50 deletion in the *DMD* gene. This is a frame-shifting deletion that results in absence of dystrophin protein at the muscle cell membrane, leading to progressive muscle weakness and degeneration. An exon 45 - 50 deletion is amenable to a therapeutic strategy that involves skipping of exon 51, which would produce an in-frame exon 45 - 51 deletion and an internally truncated, but partially functional, dystrophin protein that is localized to the muscle cell membrane (Aartsma-Rus and van Ommen, 2010). This mutation meets inclusion criteria for treatment with eteplirsen. If urine exRNA, and/or urine cell RNA from an individual with an exon 45 - 50 deletion who is receiving a drug that induces skipping of exon 51 is confirmed to have an exon 51 inclusion percentage that is significantly less than the baseline 99+%, then it is reasonable to conclude that it

is due to activity of the ASO. Indeed, we are unaware of a scientifically rational alternative explanation.

- **Mmm... controls are run all the time to prove the obvious... e.g. assay specificity. A few patients that are NOT under treatment with ASO and show substantial inclusion of exon 51 would have been comforting.**

Response: In Supplementary Fig. 15, we show that inclusion of exon 51 is > 99% in urine exRNA of several untreated individuals. Also, please see the Response to Point 11 above.

References:

- Aartsma-Rus, A. & van Ommen, G.J. Progress in therapeutic antisense applications for neuromuscular disorders. *Eur J Hum Genet* **18**, 146-153 (2010).
- Bouge, A.L. et al. Targeted RNA-Seq profiling of splicing pattern in the DMD gene: exons are mostly constitutively spliced in human skeletal muscle. *Sci Rep* **7**, 39094 (2017).
- Verheul, R.C., van Deutekom, J.C. & Datson, N.A. Digital Droplet PCR for the Absolute Quantification of Exon Skipping Induced by Antisense Oligonucleotides in (Pre-)Clinical Development for Duchenne Muscular Dystrophy. *PLoS One* **11**, e0162467 (2016).

Reviewer #2 (Remarks to the Author):

- The authors have addressed the reviewer points sufficiently. I have no additional points to raise.

Reviewer #3 (Remarks to the Author):

- The paper is very much improved and I am satisfied.

Reviewer #4 (Remarks to the Author):

- The authors have addressed my concerns.